# DeCoRe: Decoding by Contrasting Retrieval Heads to Mitigate Hallucinations

## Abstract

Large Language Models (LLMs) often hallucinate, producing unfaithful or factually incorrect outputs by misrepresenting the provided context or incorrectly recalling internal knowledge. Recent studies have identified specific attention heads within the Transformer architecture, known as *retrieval heads*, responsible for extracting relevant contextual information. We hypothesise that masking these retrieval heads can induce hallucinations and that contrasting the outputs of the base LLM and the masked LLM can reduce hallucinations. To this end, we propose **De**coding by **Co**ntrasting **Re**trieval Heads (DeCoRe), a novel training-free decoding strategy that amplifies information found in the context and model parameters. DeCoRe mitigates potentially hallucinated responses by dynamically contrasting the outputs of the base LLM and the masked LLM, using conditional entropy as a guide. Our extensive experiments confirm that DeCoRe improves performance on tasks requiring high contextual faithfulness, such as summarisation (XSum by 18.6%), instruction following (MemoTrap by 10.9%), and open-book question answering (NQ-Open by 2.4% and NQ-Swap by 5.5%).[1]

## 1 Introduction

Large Language Models (LLMs) have emerged as powerful natural language generators, demonstrating remarkable capabilities across a range of tasks (Radford et al., 2019; Brown et al., 2020; Wei et al., 2022a; Ouyang et al., 2022). However, LLMs are prone to *hallucinations*, where the model generates content that is not grounded in reality or misrepresents the facts (Ji et al., 2023; Rawte et al., 2023; Zhang et al., 2023c; Li et al., 2024a). The tendency of LLMs to hallucinate undermines their reliability, especially when applied in high-stakes domains such as clinical decision-making or legal reasoning (Ahmad et al., 2023; Dahl et al., 2024).

Understanding the underlying mechanisms responsible for hallucinations in LLMs remains challenging. Wu et al. (2024) found that there are special attention heads responsible for retrieving relevant information from a given context, which they called *"retrieval heads"*. While identifying these mechanisms is key to understanding LLMs, little research has explored how to use these insights to effectively mitigate hallucinations, which is the focus of our work.

We propose a novel decoding method termed **De**coding by **Co**ntrasting **Re**trieval Heads (**DeCoRe**), as illustrated in Figure 1. This method builds on the assumption that masking retrieval heads can induce hallucination by impairing the ability of the model to retrieve relevant information from the context. DeCoRe leverages Contrastive Decoding (Li et al., 2023) to amplify the differences between the original and the hallucinating outputs, leading to more accurate final responses. Furthermore, we propose using the conditional entropy of the model's next-token distribution to control the contrastive decoding mechanism.

Our findings show that DeCoRe significantly improves accuracy in tasks that require contextual faithfulness, such as XSum (Narayan et al., 2018), MemoTrap (Liu & Liu, 2023), Open Book Natural Questions (NQ; Kwiatkowski et al., 2019), and NQ-Swap (Longpre et al., 2021). Furthermore, our experiments show that DeCoRe enhances the model's accuracy in factual recall tasks. For example, in the TriviaQA (Joshi et al., 2017) and PopQA (Mallen et al., 2023) dataset, DeCoRe shows an accuracy gain compared to other hallucination mitigation methods. Similarly, when applied to

---

[1]Code is available at `https://anonymous.4open.science/r/decore-4FB7`.

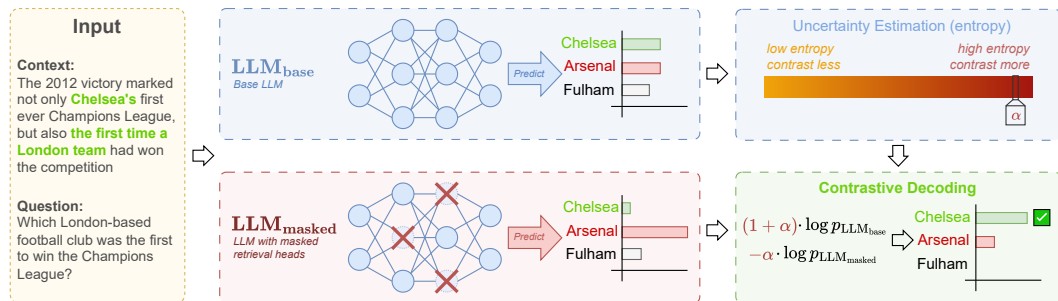

Figure 1: Overview of the DeCoRe workflow. Given the same input, the base LLM ($\text{LLM}_{\text{base}}$) and the variant with masked retrieval heads ($\text{LLM}_{\text{masked}}$) predict the next token. An uncertainty estimation is applied to the base model's output using conditional entropy: higher conditional entropy increases the contrastive factor ($\alpha$), penalising predictions that align with the $\text{LLM}_{\text{masked}}$. The final prediction is selected based on weighted contrastive decoding of the outputs from both models, leading to a more grounded response.

TruthfulQA (Lin et al., 2022), the Llama3-8b-Instruct model (Dubey et al., 2024), when combined with DeCoRe, generates more truthful and informative responses than other comparable hallucination mitigation methods. Finally, our experiments on MuSiQue (Trivedi et al., 2022) show that DeCoRe can significantly improve the accuracy of the model in long-form generation and reasoning tasks, for example, when used jointly with Chain of Thought (CoT; Wei et al., 2022b) prompting.

## 2    DeCoRe: Decoding by Contrasting Retrieval Heads

DeCoRe operates by masking specific retrieval heads to trigger hallucinations and then employs a contrastive mechanism that penalises outputs resembling those from the hallucinating model, thereby amplifying the more accurate predictions of the base model. We further enhance this approach with a dynamic entropy-controlled mechanism to adjust the contrastive effect based on the entropy of the next token distribution of the model. Figure 1 illustrates this process over time.

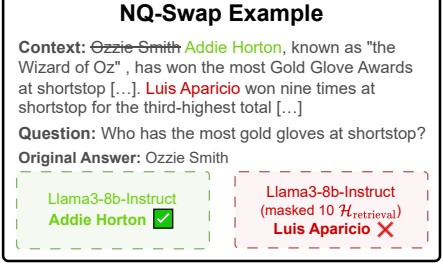

Figure 2: Example of hallucination induced by masking retrieval heads in the NQ-Swap task. The base model retrieves the correct answer from the substituted context, while the masked model generates an incorrect answer.

### 2.1    Masking Retrieval Heads

In this section, we describe how we mask retrieval heads in our base LLM to induce hallucinations, following the notation from Vaswani et al. (2017).

Given a base LLM $f_{\text{base}}$, let $x_{<t} = (x_1, x_2, \ldots, x_{t-1})$ be a sequence of previous tokens, where $x_i \in \mathcal{X}$ and $\mathcal{X}$ denotes the vocabulary of the model. The logits for the next token distribution at time step $t$ are given by $f_{\text{base}}(x_{<t}) \in \mathbb{R}^{|\mathcal{X}|}$, and the probability of the next token $x_t$ is:

$$p_{\text{base}}\left(x_t \mid x_{<t}\right) \propto \exp\left(f_{\text{base}}(x_{<t})\right) \tag{1}$$

In our approach, we derive a variant of the base LLM by masking a set of *retrieval heads*. We identify these heads using the method proposed by Wu et al. (2024), which involves analysing attention patterns on the Needle-in-a-Haystack (NitH; Kamradt, 2023) dataset. The NitH dataset is designed to evaluate the ability of a model to retrieve specific information from a long context, making it suitable for identifying attention heads that contribute to information retrieval. We compute a *retrieval score*, defining it as the ratio of successful copy-paste operations by each attention head, following Wu et al. (2024). Further details are provided in Appendix C.1. We then rank the attention heads according to their retrieval scores and select the top $N$ heads as retrieval heads. Let $\mathcal{H}_{\text{retrieval}} = \{(l_1, h_1), \ldots, (l_N, h_N)\}$ denote the set of retrieval heads to be masked, where $l_i$ is the layer index and $h_i$ is the head index within that layer. In a Transformer architecture, the output of

the multi-head attention mechanism at layer $l$ is given by:

$$\text{MultiHead}^{(l)}\left(Q^{(l)}, K^{(l)}, V^{(l)}\right) = \text{Concat}\left(\text{head}_1^{(l)}, \ldots, \text{head}_H^{(l)}\right) W_O^{(l)}, \quad (2)$$

where $H \in \mathbb{Z}_+$ is the number of attention heads; $Q^{(l)}$, $K^{(l)}$, and $V^{(l)} \in \mathbb{R}^{d \times d_k}$ respectively denote the query, key, and value matrices at layer $l$; $d$ denotes the model hidden dimension and $d_k$ is the key dimension (where $d_k = d/H$); $W_O^{(l)} \in \mathbb{R}^{H d_k \times d}$ denotes output projection layer; and each head is computed as:

$$\text{head}_h^{(l)} = \text{Attention}\left(Q^{(l)} W_{Q,h}^{(l)}, K^{(l)} W_{K,h}^{(l)}, V^{(l)} W_{V,h}^{(l)}\right), \quad (3)$$

where $W_{Q,h}^{(l)}$, $W_{K,h}^{(l)}$, $W_{V,h}^{(l)} \in \mathbb{R}^{d \times d_k}$ respectively denote query, key, value weight matrices at layer $l$. To mask each head $\text{head}_h^{(l)}$ such that $(l, h) \in \mathcal{H}_{\text{retrieval}}$, we define a mask $m_h^{(l)} \in \{0, 1\}$ such that:

$$m_h^{(l)} = \begin{cases} 0 & \text{if } (l, h) \in \mathcal{H}_{\text{retrieval}}, \\ 1 & \text{otherwise.} \end{cases} \quad (4)$$

Then, the masked multi-head attention output at layer $l$ becomes:

$$\text{MultiHead}_{\text{masked}}^{(l)}\left(Q^{(l)}, K^{(l)}, V^{(l)}\right) = \text{Concat}\left(m_1^{(l)} \cdot \text{head}_1^{(l)}, \ldots, m_H^{(l)} \cdot \text{head}_H^{(l)}\right) W_O^{(l)}, \quad (5)$$

where $\cdot$ denotes the scalar multiplication. The token logits of the masked model is then given by $f_{\text{masked}}(x_{<t})$, and the next-token distribution is:

$$p_{\text{masked}}(x_t \mid x_{<t}) \propto \exp\left(f_{\theta_{\text{masked}}}(x_{<t})\right). \quad (6)$$

We hypothesise that masking retrieval heads reduces the model's ability to retrieve relevant information from the context, leading to a higher likelihood of generating hallucinations. We empirically validate this hypothesis in Appendix D, where we evaluate the model on factuality and faithfulness evaluation tasks before and after masking retrieval heads. Figure 2 shows an example of the induced hallucination after masking 10 retrieval heads in Llama3-8b-Instruct (Dubey et al., 2024).

## 2.2 CONTRASTING BASE AND MASKED LLMS

Given the base and masked LLMs from Section 2.1, our goal is to improve the faithfulness of the generated output. To achieve this, we propose contrasting the next-token distributions of the base and masked models, effectively increasing the likelihood of the tokens selected by the former while decreasing the likelihood of the tokens selected by the latter. More formally, DeCoRe uses the following next-token distribution $p(x_t \mid x_{<t})$:

$$p(x_t \mid x_{<t}) \propto \exp\left[(1 + \alpha) \log p_{\text{base}}(x_t \mid x_{<t}) - \alpha \log p_{\text{masked}}(x_t \mid x_{<t})\right]. \quad (7)$$

In Equation (7), the new next-token distribution $p(x_t \mid x_{<t})$ is defined by contrasting the next-token distributions of the base model $p_{\text{base}}(x_t \mid x_{<t})$ and the masked model $p_{\text{masked}}(x_t \mid x_{<t})$, introduced in Section 2.1; and $\alpha \in \mathbb{R}$ is a scaling factor that controls the weight of the next-token distribution induced by the base model $p_{\text{base}}(x_t \mid x_{<t})$ and the one induced by the masked model $p_{\text{masked}}(x_t \mid x_{<t})$. The term $(1 + \alpha) \log p_{\text{base}}(x_t \mid x_{<t})$ in Equation (7) encourages the base LLM to predict a highly probably token under its distribution, while $\alpha \log p_{\text{masked}}(x_t \mid x_{<t})$ penalises predictions that are also likely under the masked model's distribution.

## 2.3 DYNAMIC CONTRASTIVE DECODING

We propose a method to dynamically select the hyper-parameter $\alpha$ using an uncertainty quantification, namely the *conditional entropy*, which is a reliable predictor for whether a model might generate hallucinations (Malinin & Gales, 2021; Kadavath et al., 2022).[2] For a given context $x_{<t}$, the conditional entropy $H(x_t)$ of the next-token distribution of a model $p(x_t \mid x_{<t})$ is defined as:

$$H(x_t) = -\sum_{x_t \in \mathcal{V}} p(x_t \mid x_{<t}) \log p(x_t \mid x_{<t}), \quad (8)$$

where $\mathcal{V}$ denotes the vocabulary of the model; a high conditional entropy indicates that the model is uncertain about its prediction. We propose dynamically tuning the contrastive decoding process (Equation (7)) using the conditional entropy of the base model (Equation (8)). Specifically, we set the hyperparameter $\alpha$ in Equation (7) to $\alpha = H(x_t)$, where higher next-token distribution entropy increases $\alpha$, reducing the likelihood of selecting potentially hallucinated generations.

---

[2]As we also validate in our experiments in Appendix F.

## 3 EXPERIMENT SETUP

Hallucinations in LLMs can generally be categorised into two types – *factuality* and *faithfulness hallucinations* (Huang et al., 2023; Hong et al., 2024). Factuality hallucinations refer to instances where the generated content is factually incorrect with respect to world knowledge. Faithfulness hallucinations refer to instances where the generated content fails to accurately adhere to the given source of information. Moreover, hallucinations may "snowball" in longer generation tasks such as multi-hop reasoning, compounding errors across generation steps due to the inherently sequential behaviour of auto-regressive decoding in LLMs (Merrill & Sabharwal, 2023; Zhang et al., 2023a).

In this section, we describe our experimental setup to evaluate DeCoRe. We employ a diverse set of benchmarks to assess contextual faithfulness, factual accuracy, and multi-hop reasoning capability. Given that retrieval heads are important in correctly retrieving contextual information (Wu et al., 2024) and looking back over *long reasoning processes* (Wu et al., 2024), while attention heads play a significant role in information transfer between tokens (Elhage et al., 2021), our experimental setup is designed to answer the following key research questions: 1) Can DeCoRe improve *contextual faithfulness*? 2) Can DeCoRe maintain or enhance the *factual recall* capabilities of LLMs? 3) Does coupling DeCoRe with CoT improve the multi-hop reasoning capability of the LLM?

### 3.1 DATASETS AND EVALUATION METRICS

**Faithfulness.** We evaluate faithfulness on summarisation, instruction-following, and reading comprehension datasets. XSum (Narayan et al., 2018) is an abstractive summarisation dataset developed from BBC articles. We sub-sample 1,000 examples, following Chuang et al. (2024), and evaluate summaries using ROUGE-L (Lin, 2004), BERTScore (Zhang et al., 2020), and factKB (Feng et al., 2023) for factual consistency. MemoTrap (Liu & Liu, 2023) tests whether models can avoid *memorisation traps* and adhere to the given instructions, with performance reported using macro- and micro-averaged accuracy. Instruction-Following Eval (IFEval; Zhou et al., 2023) evaluates the ability of the models to follow instructions on a set of verifiable instructions such as "write in more than 400 words". The performance is reported using Prompt-level and Instruction-level strict accuracies, which measure the percentage of prompts where all verifiable instructions are followed, and the percentage of verifiable instructions followed overall. Open-Domain Natural Questions (NQ-Open; Lee et al., 2019) a QA dataset where we use an open-book configuration with one supporting document per question as described by Liu et al. (2024). NQ-Swap (Longpre et al., 2021) is a version of NQ where the answer entity in the context was replaced with another entity and is used to evaluate the faithfulness of the model to the modified context. We evaluate the models with the Exact Match (EM) metric and, following Kandpal et al. (2023) and Liu et al. (2024), we consider a prediction as correct if any sub-string of the prediction exactly matches any of the ground truth answers.

**Factuality.** For factuality evaluation, we use four datasets—TruthfulQA, TriviaQA, PopQA, and NQ-Open. TruthfulQA (Lin et al., 2022) (MC1, MC2, MC3, and Gen) is used to evaluate whether models can avoid common human falsehoods; MC1, MC2, and MC3 are multi-label classification tasks, and Gen is a generation task where evaluations use fine-tuned GPT models to assess the correctness and informativeness of the generated outputs. TriviaQA (Joshi et al., 2017), PopQA (Mallen et al., 2023), and NQ-Open are open-domain QA datasets used to evaluate the ability of a model to answer questions about trivia, long-tail entities, and Google searches, respectively. We use closed-book configuration on these datasets to evaluate factual recall.

**Chain of Thought Reasoning.** We evaluate DeCoRe in CoT-style reasoning tasks on both closed- and open-book setups; we use MuSiQue (Trivedi et al., 2022), a multi-hop QA dataset that requires models to answer questions by reasoning using multiple and disconnected pieces of information.

More details on the evaluation protocol are available in Table 31.

### 3.2 MODELS AND BASELINES

We evaluate two models from the Llama3 family Dubey et al. (2024), namely Llama3-8B-Instruct and Llama3-70B-Instruct. In Appendix I, we also report results from other model families, such as Mistral (Jiang et al., 2023) and Qwen2 (Yang et al., 2024).

We compare DeCoRe against six baselines: 1) Greedy decoding; 2) Contrastive Decoding (CD; Li et al., 2023), where LLaMA3-8B-Instruct serves as the amateur model and LLaMA3-70B-Instruct act as the expert model; 3) Context-Aware Decoding (CAD; Shi et al., 2024), a variant of CD where the amateur model is the same as the expert model but is not presented with the additional context; 4) Decoding by Contrasting Layers (DoLa; Chuang et al., 2023) that subtracts the logits in early layers to calibrate the final-layer logits. We evaluate two versions: DoLa-low (*i.e.,* contrasting the first half of the layers with the final layer) and DoLa-high (*i.e.,* contrasting the second half with the final layer); 5) Activation Decoding (AD; Chen et al., 2024) which uses the sharpness of context activations within intermediate layers to calibrate the next token prediction; 6) ITI (Li et al., 2024b) that trains linear classifiers on TruthfulQA data to obtain "factual" heads and layers with corresponding "factual" direction vectors and then apply intervention during the decoding process. Note that ITI requires a training process on labelled data , whereas other baselines and DeCoRe are training-free. Also note that CAD is only applicable on tasks with additional context (*i.e.,* XSum, open book NQ-Open, NQ-Swap, and open book MuSiQue). All implementation details are available in Appendix K.

### 3.3 DECORE VARIANTS

We evaluate three variants of DeCoRe: 1) DeCoRe$_{static}$, which employs a static scaling factor $\alpha$ throughout generation; 2) DeCoRe$_{entropy}$, which entropy to dynamically adjust the strength of the contrastive decoding; 3) DeCoRe$_{entropy-lite}$, which is similar to DeCoRe$_{entropy}$, except that it employs a smaller LLM with the same vocabulary space as the masked LLM. We use LLama3-70B-Instruct and LLama3-8B-Instruct as the base and masked LLMs, respectively.

## 4 RESULTS

In the following, we present the evaluation results of DeCoRe across faithfulness, factuality, and multi-hop reasoning tasks. We show that DeCoRe mitigates faithfulness and factuality hallucinations, and improves the accuracy of the model when combined with CoT prompting. These effectively answer our research questions stated in Section 3. Additionally, we examine the impact of the number of masked retrieval heads on task performance. Finally, we demonstrate that DeCoRe reduces conditional entropy over time in long-generation tasks, contributing to more accurate outputs. These results highlight DeCoRe's broad effectiveness in enhancing LLM performance.

Table 1: Performance of different models and decoding methods on faithfulness evaluation tasks. For each base model, the best performance is indicated in **bold**, and the second-best is underlined.

| Model | XSum | | | | MemoTrap | | IFEval | | | NQ-Open | NQ-Swap | Avg ↑ |
|---|---|---|---|---|---|---|---|---|---|---|---|---|
| | ROUGE-L ↑ | BERTScore-F1 ↑ | factKB ↑ | Avg ↑ | Macro Acc ↑ | Micro Acc ↑ | Prompt Acc ↑ | Instruct Acc ↑ | Avg ↑ | EM ↑ | EM ↑ | |
| **Llama3-8b-Instruct** | 19.90 | 67.23 | 47.61 | 44.91 | 65.86 | 64.40 | **70.24** | 78.30 | 74.27 | 69.68 | 60.62 | 60.43 |
| + ITI (Li et al., 2024b) | 13.25 | 59.96 | 34.35 | 35.85 | 62.65 | 58.96 | 52.31 | 63.19 | 57.75 | 56.16 | 51.08 | 50.21 |
| + CAD (Shi et al., 2024) | 18.82 | 67.20 | **67.16** | **51.06** | **76.58** | **76.76** | - | - | - | 69.83 | **74.21** | **66.57** |
| + DoLA (low) (Chuang et al., 2023) | 19.82 | 67.19 | 47.21 | 44.74 | 65.27 | 63.69 | 69.69 | 78.18 | 73.94 | 69.68 | 60.77 | 60.17 |
| + DoLA (high) (Chuang et al., 2023) | 19.92 | 67.34 | 48.49 | 45.25 | 64.85 | 63.17 | **70.24** | **78.66** | **74.45** | 69.49 | 60.98 | 60.35 |
| + AD (Chen et al., 2024) | 19.79 | 67.31 | 48.49 | 45.20 | 65.38 | 64.28 | 67.65 | 76.26 | 71.96 | 68.93 | 60.51 | 59.84 |
| + DeCoRe$_{static}$ | 19.87 | **67.83** | 64.07 | 50.59 | 69.53 | 69.20 | 69.13 | 78.06 | 73.60 | 70.62 | 64.43 | 63.64 |
| + DeCoRe$_{entropy}$ | 19.45 | 67.69 | 66.10 | **51.08** | 74.14 | 74.87 | 68.39 | 76.38 | 72.39 | **70.66** | 66.08 | 64.86 |
| **Llama3-70b-Instruct** | 22.41 | 69.77 | 61.32 | 51.17 | 68.47 | 66.52 | 77.45 | 84.41 | 80.93 | 71.07 | 76.11 | 66.39 |
| + ITI (Li et al., 2024b) | 21.64 | 69.46 | 61.33 | 50.81 | 71.24 | 68.73 | 76.71 | 83.69 | 80.20 | 71.90 | 74.76 | 66.60 |
| + CD (Li et al., 2023) | **22.71** | **69.99** | 54.73 | 49.14 | 69.27 | 67.55 | 71.72 | 79.74 | 75.73 | 65.80 | 68.37 | 63.66 |
| + CAD (Shi et al., 2024) | 21.45 | 69.28 | **65.61** | 52.11 | **83.58** | **83.89** | - | - | - | 71.83 | **84.70** | **71.36** |
| + DoLA (low) (Chuang et al., 2023) | 22.46 | 69.80 | 61.11 | 51.12 | 67.99 | 65.93 | 77.08 | 84.29 | 80.69 | 71.07 | 75.98 | 66.23 |
| + DoLA (high) (Chuang et al., 2023) | 22.43 | 69.93 | 59.99 | 50.78 | 67.92 | 65.81 | 78.00 | 84.65 | 81.33 | 70.40 | 75.26 | 66.04 |
| + AD (Chen et al., 2024) | 22.49 | 69.91 | 60.57 | 50.99 | 67.51 | 66.44 | 76.89 | 84.41 | 80.65 | 71.15 | 74.02 | 65.93 |
| + DeCoRe$_{static}$ | 21.94 | 69.35 | 64.88 | 52.06 | 71.96 | 71.41 | **78.56** | **84.89** | **81.73** | 72.51 | 79.06 | 68.29 |
| + DeCoRe$_{entropy}$ | 21.93 | 69.40 | 65.49 | **52.27** | 74.07 | 73.65 | **78.56** | **84.89** | **81.73** | 72.66 | 79.79 | 68.94 |
| + DeCoRe$_{entropy-lite}$ | 22.28 | 69.34 | 59.57 | 50.40 | 72.11 | 70.58 | 61.37 | 71.46 | 66.42 | 71.26 | 75.90 | 63.76 |

**DeCoRe Mitigates Faithfulness Hallucinations.** Table 1 shows the performance of various models and decoding methods on faithfulness evaluation tasks. The results show that DeCoRe$_{static}$, DeCoRe$_{entropy}$, and DeCoRe$_{entropy-lite}$ consistently improve the base models across all tasks and model sizes. Specifically, DeCoRe$_{entropy}$ yields the best or very competitive results in several faithfulness-related tasks. For instance, with Llama3-8b-Instruct, DeCoRe$_{entropy}$ attains a Macro Accuracy of 74.14% and a Micro Accuracy of 74.87% on MemoTrap, producing significantly more accurate results than all baselines. DeCoRe$_{entropy}$ also achieves the highest EM scores on open-book NQ-Open and competitive results on NQ-Swap. Similarly, with Llama3-70b-Instruct, DeCoRe$_{entropy}$ achieves the highest EM score on NQ-Open and competitive results on NQ-Swap. In instruction-

following tasks, DeCoRe$_{entropy}$ also achieves competitive scores in the IFEval benchmark with Llama3-8b-Instruct, yielding Instruct and Prompt Strict Accuracy values of 68.39% and 76.38%, respectively. With Llama3-70b-Instruct, DeCoRe$_{static}$ and DeCoRe$_{entropy}$ achieve the joint-highest Instruct and Prompt Strict Accuracy values of 78.56% and 84.89%, respectively.

While CAD yields accurate results on tasks like XSum and NQ-Swap, its applicability remains limited to datasets that provide additional contexts, making it not trivial to adapt to tasks such as MemoTrap and IFEval. On the other hand, DeCoRe$_{static}$ and DeCoRe$_{entropy}$ both improve the base models in all tasks. In Appendix H.1, we provide pairwise statistical significance analyses, indicating the statistically significant improvement yielded by DeCoRe$_{entropy}$ compared to other baselines.

**DeCoRe Mitigates Faithfulness Hallucinations amidst Distractor Documents.** Table 2 presents the performance of various models and decoding methods on NQ-Open under the Lost-in-the-Middle (LitM) setup, where the context contains one gold document and nine distractor documents. The Oracle setup indicates that the model is given only the single gold document without any distractors, providing an upper bound for the accuracy of the model. The results show that DeCoRe$_{static}$, DeCoRe$_{entropy}$, and DeCoRe$_{entropy-lite}$ consistently produce more accurate results than the base models across different positions of the gold document and model sizes. Specifically, DeCoRe$_{entropy}$ achieves the highest EM scores in several configurations. For instance, with Llama3-8b-Instruct, DeCoRe$_{entropy}$ attains the highest

Table 2: Performance of different models and decoding methods on NQ-Open with Lost-in-the-Middle Setup (one gold document with nine distractor documents). The Average column represents the mean of the Gold 1st, Gold 4th, and Gold 9th EMs. The best performance for each base model is indicated in **bold**, and the second-best is underlined.

| Model | NQ-Open | | | | |
|---|---|---|---|---|---|
| | Oracle ↑ | Gold 1st ↑ | Gold 4th ↑ | Gold 9th ↑ | Avg ↑ |
| Llama3-8b-Instruct | 69.68 | 52.92 | 45.61 | 44.48 | 47.34 |
| + ITI (Li et al., 2024b) | 56.16 | 16.61 | 13.45 | 11.45 | 13.84 |
| + CAD (Shi et al., 2024) | 69.83 | 40.57 | 31.53 | 29.30 | 33.80 |
| + DoLA (low) (Chuang et al., 2023) | 69.68 | 52.88 | 45.76 | 44.37 | 47.34 |
| + DoLA (high) (Chuang et al., 2023) | 69.49 | 52.28 | 45.39 | 44.14 | 47.27 |
| + AD (Chen et al., 2024) | 68.93 | 52.96 | 45.46 | 43.96 | 47.46 |
| + DeCoRe$_{static}$ | 70.62 | 54.58 | 47.42 | 44.90 | 48.97 |
| + DeCoRe$_{entropy}$ | 70.66 | 54.39 | 47.50 | 45.42 | 49.10 |
| Llama3-70b-Instruct | 71.07 | 60.49 | 52.99 | 49.00 | 54.16 |
| + ITI (Li et al., 2024b) | 71.90 | 60.53 | 49.91 | 46.25 | 52.23 |
| + CD (Li et al., 2023) | 71.90 | 58.57 | 51.64 | 47.87 | 52.69 |
| + CAD (Shi et al., 2024) | 71.83 | 58.27 | 48.10 | 43.16 | 49.84 |
| + DoLA (low) (Chuang et al., 2023) | 71.07 | 60.45 | 52.96 | 49.04 | 54.15 |
| + DoLA (high) (Chuang et al., 2023) | 70.40 | 59.32 | 52.24 | 48.32 | 53.29 |
| + AD (Chen et al., 2024) | 71.15 | 60.41 | 52.84 | 48.93 | 54.06 |
| + DeCoRe$_{static}$ | 72.51 | 60.53 | 53.11 | 49.12 | 54.25 |
| + DeCoRe$_{entropy}$ | 72.66 | 60.72 | 53.07 | 49.38 | 54.39 |
| + DeCoRe$_{entropy-lite}$ | 71.26 | 60.45 | 53.22 | 48.51 | 54.06 |

Oracle score of 70.66% and the best EM score of 45.42% when the gold document is placed ninth. Similarly with Llama3-70b-Instruct, DeCoRe$_{entropy}$ achieves the highest Oracle score of 72.66% and the best EM scores when the gold document is first and ninth. While other methods like ITI and CAD show improvements in certain cases, their performance is generally less consistent compared to DeCoRe$_{static}$ and DeCoRe$_{entropy}$. Both ITI and CAD significantly underperform when applied to Llama3-8b-Instruct, especially when the gold document is not first, yielding EM scores as low as 11.45% and 29.30%, respectively, when the gold document is ninth.

Table 3: Performance of different models and decoding methods on factuality evaluation tasks. For each base model, the best performance is indicated in **bold**, and the second-best is underlined.

| Model | TruthfulQA (MC) | | | | TriviaQA | PopQA | TruthfulQA (Gen) | | | | NQ-Open | Avg ↑ |
|---|---|---|---|---|---|---|---|---|---|---|---|---|
| | MC1 ↑ | MC2 ↑ | MC3 ↑ | Avg ↑ | EM ↑ | EM ↑ | %Truth | %Info | %T ∩ I | %Reject | EM ↑ | |
| **Llama3-8b-Instruct** | 39.41 | 55.69 | 30.31 | 41.80 | 56.58 | 26.64 | 80.66 | 63.89 | 44.55 | 43.94 | 29.04 | 39.72 |
| + ITI (Li et al., 2024b) | **43.70** | **62.78** | **34.91** | **47.13** | 48.41 | 15.63 | **87.52** | **78.46** | **66.10** | 25.46 | 22.07 | 39.87 |
| + DoLA (low) (Chuang et al., 2023) | 39.05 | 55.65 | 30.06 | 41.59 | 56.63 | 26.58 | 80.66 | 62.91 | 43.70 | 45.04 | 29.15 | 39.53 |
| + DoLA (high) (Chuang et al., 2023) | 38.68 | 55.64 | 30.19 | 41.50 | 56.50 | 26.49 | 80.78 | 62.67 | 43.45 | 44.92 | 29.19 | 39.43 |
| + AD (Chen et al., 2024) | 31.21 | 55.30 | 28.28 | 38.26 | 54.93 | 26.38 | 80.42 | 63.40 | 43.82 | 43.82 | 28.32 | 38.34 |
| + DeCoRe$_{static}$ | 38.68 | 55.74 | 29.80 | 41.41 | 56.93 | 26.86 | 80.78 | 48.71 | 41.74 | 41.74 | 29.42 | 40.67 |
| + DeCoRe$_{entropy}$ | 38.43 | 55.86 | 30.95 | 41.75 | 56.40 | 26.88 | 78.95 | 74.05 | 53.00 | 38.68 | 28.96 | **41.40** |
| **Llama3-70b-Instruct** | 49.57 | 70.60 | 37.85 | 52.67 | 74.77 | 40.63 | 88.74 | 77.72 | 66.46 | 53.12 | 40.08 | 54.92 |
| + ITI (Li et al., 2024b) | 48.96 | 67.04 | 37.27 | 51.09 | 73.54 | 39.62 | 82.50 | 74.30 | 56.92 | **37.94** | 38.57 | 51.95 |
| + CD (Li et al., 2023) | **57.77** | **76.65** | **47.08** | **60.50** | 72.83 | 37.03 | 88.25 | 76.38 | 52.26 | 36.23 | 56.59 |
| + DoLA (low) (Chuang et al., 2023) | 49.45 | 70.58 | 37.75 | 52.59 | 74.74 | 40.65 | 88.74 | 77.60 | 66.34 | 52.88 | 40.08 | 54.88 |
| + DoLA (high) (Chuang et al., 2023) | 49.69 | 70.88 | 38.01 | 52.86 | 73.96 | 40.00 | 88.98 | 58.38 | 47.37 | 54.71 | 39.59 | 50.76 |
| + AD (Chen et al., 2024) | 42.23 | 67.56 | 35.37 | 48.39 | 74.14 | 40.53 | 87.39 | 67.20 | 54.59 | 49.33 | 40.23 | 51.58 |
| + DeCoRe$_{static}$ | 51.29 | 72.02 | 40.24 | 54.52 | **74.79** | **40.74** | 88.25 | 62.91 | 51.16 | 54.96 | 40.41 | 52.32 |
| + DeCoRe$_{entropy}$ | 53.98 | 73.44 | 42.55 | 56.66 | 74.76 | 40.58 | **89.23** | 59.73 | 49.11 | 56.79 | **40.45** | 52.31 |
| + DeCoRe$_{entropy-lite}$ | 55.32 | 73.38 | 43.74 | 57.48 | 73.87 | 39.09 | 88.13 | **90.09** | **78.21** | 52.02 | 39.21 | **57.57** |

**DeCoRe Mitigates Factuality Hallucinations.** While DeCoRe is primarily designed to improve contextual faithfulness, its impact on factual recall tasks is an open question. To this end, we evaluate DeCoRe on a range of tasks where the model needs to produce factually correct generations—results are outlined in Table 3. We can see that DeCoRe improves the accuracy of the models across various

factuality evaluation tasks. For the Llama3-8b-Instruct model, DeCoRe$_{entropy}$ demonstrates improvements in several TruthfulQA (Generation) metrics. Specifically, it achieves an informativeness score of 74.05% and an intersection of truthfulness and informativeness score of 53.00%, second only to ITI, which requires fine-tuning the model on TruthfulQA data.[3] Furthermore, DeCoRe$_{static}$ yields the highest EM score on TriviaQA (56.93%) among all decoding strategies and achieves competitive EM scores on PopQA. For the larger Llama3-70b-Instruct model, DeCoRe$_{entropy}$ achieves the highest truthfulness score (89.23%) on TruthfulQA (Gen); it performs competitively across informativeness and the intersection metrics, yielding the highest EM score on closed-book NQ-Open (40.45%). Finally, DeCoRe$_{static}$ yields the highest EM score on PopQA (40.74%).

These results suggest that DeCoRe methods can improve contextual faithfulness and factual consistency across different datasets. We believe this phenomenon is closely related to the hypothesis of attention heads as Information Movement (Elhage et al., 2021), which suggests that attention heads facilitate the transfer of information between tokens and that the residual stream vector space of one token typically contains information from other tokens. Thus, while factual recall may occur in the Multi-Layer Perceptron (Geva et al., 2021; Meng et al., 2022), masking retrieval heads may interfere with the information transfer from the question to the generated answer, potentially leading to hallucinations. We hypothesise that DeCoRe leverages this phenomenon, improving downstream results in factual recall tasks. In Appendix H.2, we provide detailed pairwise statistical significance analyses of our results, indicating the statistically significant improvement yielded by DeCoRe$_{entropy}$ compared to other baselines in tasks such as PopQA and closed-book NQ-Open.

Table 4: Performance of different models and decoding methods on MuSiQue, a multi-hop reasoning dataset, with and without CoT prompting in both closed-book and open-book settings. For each base model, the best performance is indicated in **bold**, and the second-best is underlined.

| Model | MuSiQue without CoT | | MuSiQue with CoT | | Avg ↑ |
|---|---|---|---|---|---|
| | Closed Book ↑ | Open Book ↑ | Closed Book ↑ | Open Book ↑ | |
| Llama3-8b-Instruct | 7.41 | 58.83 | 14.61 | 69.84 | 37.67 |
| + CAD (Shi et al., 2024) | - | 57.88 | - | 73.02 | 38.23 |
| + ITI (Li et al., 2024b) | 4.01 | 45.84 | 4.18 | 38.31 | 23.08 |
| + DoLA (Chuang et al., 2023) | 7.24 | 59.08 | **14.94** | 69.92 | 37.79 |
| + AD (Chen et al., 2024) | 6.99 | 58.63 | 14.40 | 69.92 | 37.49 |
| + DeCoRe$_{static}$ | **7.90** | 61.23 | 14.69 | 72.49 | 39.08 |
| + DeCoRe$_{entropy}$ | 7.70 | **61.98** | 13.90 | **74.47** | **39.51** |
| Llama3-70b-Instruct | **11.79** | 68.56 | 20.15 | 74.43 | 43.73 |
| + CD (Li et al., 2023) | 10.92 | 66.61 | 17.17 | 71.70 | 41.60 |
| + CAD (Shi et al., 2024) | - | 68.64 | - | 74.02 | 43.65 |
| + ITI (Li et al., 2024b) | 10.88 | 68.14 | 20.44 | 74.27 | 43.43 |
| + DoLA (Chuang et al., 2023) | 11.42 | 68.68 | 20.15 | 74.64 | 43.72 |
| + AD (Chen et al., 2024) | 11.38 | 68.14 | 20.23 | 74.27 | 43.51 |
| + DeCoRe$_{static}$ | **11.79** | 69.76 | **20.60** | **75.05** | **44.30** |
| + DeCoRe$_{entropy}$ | 11.75 | **69.84** | **20.60** | 74.93 | 44.28 |
| + DeCoRe$_{entropy-lite}$ | 11.13 | 69.34 | 18.87 | 73.36 | 43.18 |

**DeCoRe with Chain-of-Thought.** To assess DeCoRe approaches in multi-hop reasoning tasks, we evaluate them on the MuSiQue dataset. Multi-hop reasoning requires models to integrate information across multiple reasoning steps, and retrieval heads may be crucial in this process as they allow models to reference earlier generated tokens. We conduct experiments in closed-book and open-book settings, with and without CoT prompting. The closed-book setting resembles factuality evaluation, where a model relies solely on its parametric knowledge, while the open-book setting resembles faithfulness evaluation where the model has access to external knowledge.

As shown in Table 4, DeCoRe variants consistently improve the EM scores across various settings. For the Llama3-8b-Instruct model, DeCoRe$_{static}$ enhances the EM score in the closed-book setup with CoT from 14.61% (base model) to 14.69%, while in the open-book setup without CoT, DeCoRe$_{entropy}$ achieves the highest score (61.98%). DeCoRe$_{entropy}$ also yields accurate results in the open-book CoT scenario, achieving the most accurate results (74.47% EM). For the Llama3-70b-Instruct model, both DeCoRe$_{static}$ and DeCoRe$_{entropy}$ yield very accurate results, improving the EM score in the closed-book setup with CoT from 20.15% (base model) to 20.60%. DeCoRe$_{static}$

---

[3]The rejection rates—the frequency by which the model answers "I have no comment"—of Llama3 models in TruthfulQA are higher than Llama2 models (Touvron et al., 2023), as reported by previous studies (Li et al., 2024b; Chuang et al., 2023); we report metrics for the non-rejection answers in Appendix E.1.

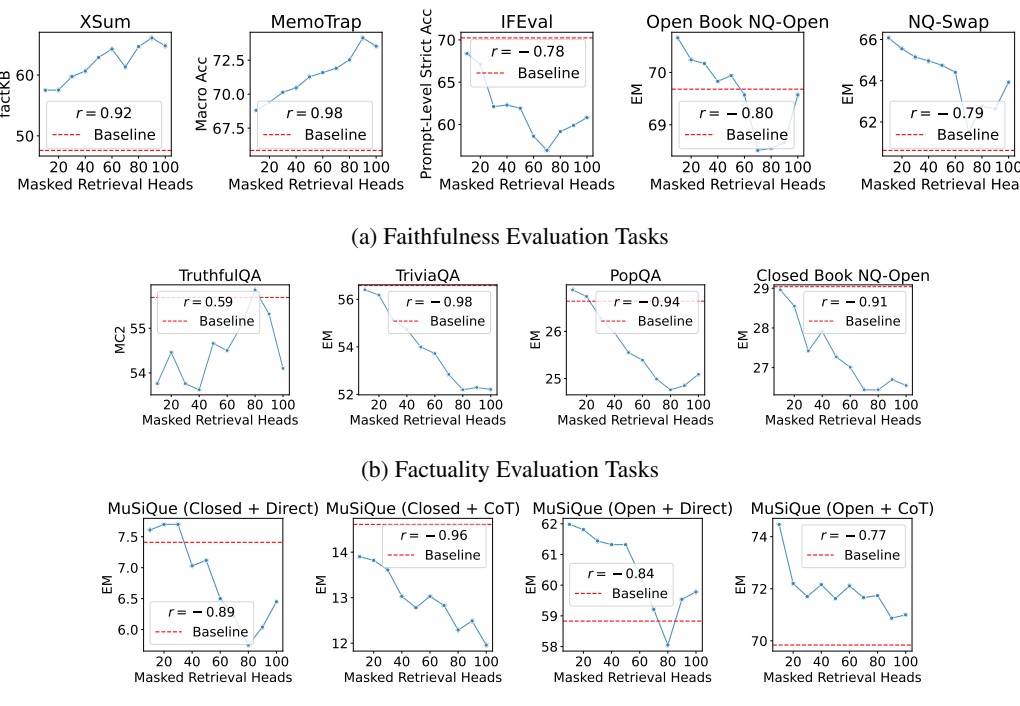

(a) Faithfulness Evaluation Tasks

(b) Factuality Evaluation Tasks

(c) Chain-of-Thought Reasoning Evaluation Tasks

Figure 3: Correlation between the number of masked retrieval heads and performance of Llama3-8B-Instruct with DeCoRe$_{entropy}$ on each task. The correlations are quantified by the Pearson Correlation Coefficient $r$ for each plot. Detailed results are listed in Table 16 and Table 18.

achieves the highest score in the open-book CoT setup (75.05%), with DeCoRe$_{entropy}$ closely following at 74.93%. These improvements underscore the effectiveness of DeCoRe in enhancing reasoning capabilities, especially when CoT prompting and external context are involved. The results show that DeCoRe improves information transfer between reasoning steps, leading to higher EM scores in closed and open-book settings. This validates the usefulness of DeCoRe in tasks requiring complex reasoning, validating the insights from Wu et al. (2024) on the significance of retrieval heads in multi-step reasoning. In Appendix H.3, we provide detailed pairwise statistical significance analyses of our results, indicating the statistically significant improvement yielded by DeCoRe$_{entropy}$ compared to other baselines, particularly in the open-book setup.

Overall, DeCoRe$_{entropy}$ achieves the highest overall aggregated score for LLaMA3-8B-Instruct and LLaMA3-70B-Instruct models, surpassing other decoding strategies as shown in Table 5. Detailed computational performance metrics (TFLOPS), showcasing the computational efficiency of DeCoRe$_{entropy}$, are provided in Appendix K.4.

**Effect of Retrieval Head Masking on Task Performance of DeCoRe.** We now analyse the correlation between the number of masked retrieval heads and the downstream results of the Llama3-8B-Instruct model; results are outlined in Figure 3. We can see that the performance of DeCoRe$_{entropy}$ across various tasks strongly correlates with the number of masked retrieval heads. For example, in XSum and MemoTrap, we can observe positive correlations between the factKB and macro accuracy scores and the number of masked retrieval heads. We attribute

Table 5: Aggregated metrics of different models and decoding methods. The overall average is calculated as the mean of Faithfulness, LitM, Factuality, and CoT aggregate scores. [1] we use the base model metrics in tasks where CAD was not applicable.

| Model | Faithfulness | LitM | Factuality | CoT | Overall |
|---|---|---|---|---|---|
| **LLaMA3-8B-Instruct** | 60.43 | 47.34 | 39.72 | 37.67 | 46.29 |
| + ITI | 50.21 | 13.84 | 39.87 | 23.08 | 31.50 |
| + CAD | **66.57**[1] | 33.80 | 39.72[1] | 38.23[1] | 44.58 |
| + DoLA (Low) | 60.17 | 47.34 | 39.53 | 37.79 | 46.21 |
| + AD | 59.84 | 47.46 | 38.34 | 37.49 | 45.78 |
| + DeCoRe$_{static}$ | 63.64 | 48.97 | 40.67 | 39.08 | 48.09 |
| + DeCoRe$_{entropy}$ | 64.86 | 49.10 | 41.40 | 39.51 | 48.72 |
| **LLaMA3-70B-Instruct** | 66.39 | 54.16 | 54.92 | 43.73 | 54.80 |
| + ITI | 66.60 | 52.23 | 51.95 | 41.60 | 53.10 |
| + CD | 63.66 | 52.69 | 56.59 | 43.65 | 54.15 |
| + CAD | **71.36**[1] | 49.84 | 54.92[1] | 43.43[1] | 54.89 |
| + DoLA (Low) | 66.23 | 54.15 | 54.88 | 43.72 | 54.75 |
| + AD | 65.93 | 54.06 | 51.58 | 43.51 | 53.77 |
| + DeCoRe$_{static}$ | 68.29 | 54.25 | 52.32 | **44.30** | 54.79 |
| + DeCoRe$_{entropy}$ | 68.94 | **54.39** | 52.31 | 44.28 | **54.98** |
| + DeCoRe$_{entropy-lite}$ | 63.76 | 54.06 | **57.57** | 43.18 | 54.64 |

tween the factKB and macro accuracy scores and the number of masked retrieval heads. We attribute

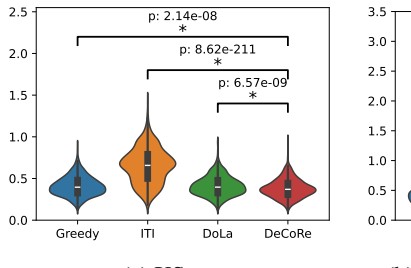 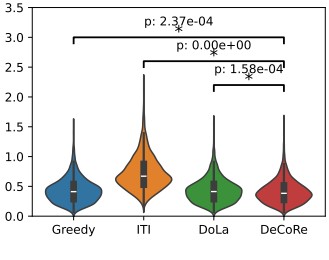 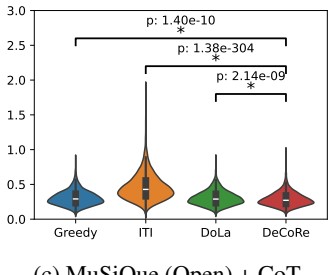

(a) XSum      (b) MuSiQue (Closed) + CoT      (c) MuSiQue (Open) + CoT

Figure 4: Comparison of Length-normalised conditional entropy of Greedy, ITI, DoLa, and DeCoRe$_{entropy}$ in long-generation tasks (*i.e.,* XSum (a), MuSiQue (Closed) + CoT (b), and MuSiQue (Open) + CoT (c)). Asterisks (*) indicate statistically significant differences between the distributions based on one-tailed Welch's t-test results. Detailed results are listed in Table 35.

this to the nature of summarisation (XSum) and instruction-following (MemoTrap) tasks, which rely heavily on the ability of the model to extract and copy relevant information accurately. However, we observe a moderate negative correlation as the number of masked retrieval heads increases on another Instruction following task, IFEval. We hypothesise that the reason behind this phenomenon is that IFEval requires a different copying mechanism than in MemoTrap. As opposed to having to provide an exact copy of a segment of the input, like in MemoTrap or partially XSum, IFEval requires the model to adhere to the instruction, which may not require an induction mechanism (*e.g.,* "In your response, the letter {letter} should appear {N} times.").

In tasks such as open-book NQ-Open and NQ-Swap, we can see a moderate negative correlation between the number of masked retrieval heads and EM. Nevertheless, in all experiments, DeCoRe$_{entropy}$ produces more accurate results than the baseline in such tasks. In factual recall tasks (*i.e.,* TriviaQA, PopQA, and closed-book NQ-Open), we can see a negative correlation between EM scores and the number of masked retrieval heads. When the masked retrieval heads fail to introduce significant differences between the "hallucinating" and the outputs of the base model, the effect of DeCoRe$_{entropy}$ becomes less pronounced. TruthfulQA differs from the other factuality tasks, showing a moderate positive correlation between downstream accuracy and the number of masked retrieval heads. This suggests that truthfulness, or the ability to discern popular misconceptions, may require different retrieval mechanisms than the typical factual recall tasks. These findings can be combined with the results of masking random attention heads (Appendix G) further supporting our hypothesis on the effectiveness of masking retrieval heads in contrastive decoding.

**DeCoRe yields lower entropy across time in long generation tasks.** We found that lower conditional entropy is related to correct predictions; generated sequences with lower conditional entropy tend to be more reliable (see Appendix F). Motivated by the importance of low conditional entropy, we evaluate the length-normalised conditional entropy of different decoding strategies in long-generation tasks (*i.e.,* XSum, and MuSiQue with CoT prompting).

As shown in Figure 4, DeCoRe$_{entropy}$ yields lower conditional entropy compared to the baselines. DeCoRe$_{entropy}$ demonstrates lower entropy in the open-book QA task (MuSiQue), with an average entropy of 0.29 compared to 0.30 for the baselines. Similarly, in XSum, DeCoRe$_{entropy}$ achieves an entropy of 0.38, outperforming the baselines. In tasks such as summarisation (XSum) and open-book QA (MuSiQue), lower entropy is crucial because the model must strictly adhere to the provided document or evidence while generating the summary or answer. Any deviation from the context can result in hallucinations or factually incorrect outputs. The lower entropy observed with DeCoRe$_{entropy}$ indicates that it generates less "surprising" sequences, reducing the likelihood of hallucinations.

Overall, the reduction in conditional entropy shows that DeCoRe$_{entropy}$ is able to maintain lower uncertainty throughout long-generation tasks. This reinforces the effectiveness of DeCoRe$_{entropy}$ in applications requiring high contextual faithfulness, such as summarisation and open-book QA. In Appendix L.2, we provide samples of text generated with DeCoRe in several long-form generation tasks, namely XSum, TruthfulQA (Gen), and MuSiQue with CoT—we can see that when using DeCoRe, the model tends to produce more faithful generations.

## 5 RELATED WORKS

**Internal Mechanism of LLMs.** Studies have attempted to dissect the inner workings of LLMs by focusing on layers (Wallat et al., 2020; Geva et al., 2021; Meng et al., 2022; Yu et al., 2024), neurons (Dai et al., 2022), and attention heads (Elhage et al., 2021; Geva et al., 2023; Yuksekgonul et al., 2024). A seminal discovery in this area is the identification of induction heads, the attention heads that perform an induction algorithm by looking back over the context to predict a similar completion (Olsson et al., 2022). Similarly, Wu et al. (2024) identified retrieval heads, a specific set of attention heads responsible for maintaining long-context factuality. These insights into the internal workings of LLMs is instrumental to our work, which focuses on these mechanisms to reduce hallucination. Our work leverages the idea that the masking of retrieval heads leads to hallucination.

**Constrained Decoding.** Constrained decoding focuses on intervening during the generation process to reduce hallucinations. One example of constrained decoding is Inference-Time Intervention (ITI; Li et al., 2024b) which intervenes by probing and modifying attention heads or layers associated with model correctness. Contrastive Decoding (CD; Li et al., 2023) improves fluency and coherence by contrasting outputs from stronger expert LMs with those from weaker, smaller LMs. Building on CD, Shi et al. (2024) propose Context Aware Decoding (CAD) to mitigate contextual hallucinations by contrasting the output of an LLM with and without the provided context. Similarly, Zhao et al. (2024) propose a CD framework that contrasts the answers generated using correct and adversarial passages. Additionally, Induced-then-Contrast Decoding (ICD; Zhang et al., 2023b) fine-tunes a factually weak LLM using an automatically generated non-factual dataset, although this approach depends on the quality of the dataset and requires fine-tuning. More closely related to DeCoRe is Decoding by Contrasting Layers (DoLa; Chuang et al., 2023) and Autocontrastive Decoding (ACD; Gera et al., 2023), which examines the internal mechanism of the LLMs without fine-tuning. Both DoLa and ACD proposed contrasting the predictions of the final layer against the earlier ones via *early exiting* (Teerapittayanon et al., 2016; Elbayad et al., 2020). Activation Decoding (Chen et al., 2024) also examines the internal mechanism of the LLMs, particularly the sharpness of context activations to calibrate the next token's probability distribution. DeCoRe distinguishes itself by masking retrieval heads to induce hallucinations, followed by a dynamic entropy-controlled contrastive decoding to penalise uncertain outputs, effectively reducing hallucinations without the need for fine-tuning. DeCoRe is also related to self-consistency (Wang et al., 2023); but rather than relying on majority voting among generations from a single model, DeCoRe amplifies the difference between a base model and its masked variant.

## 6 CONCLUSIONS

DeCoRe (Decoding by Contrasting Retrieval Heads) is a novel decoding strategy that aims to reduce faithfulness and factuality hallucinations in LLMs. DeCoRe is based on the assumption that masking retrieval heads can induce hallucinations by limiting the ability of the model to retrieve relevant information from the given context. Specifically, DeCoRe uses retrieval head masking to create a version of the model that is more likely to generate hallucinations and combines it with the original model via a contrasting decoding scheme (Section 2.2). Furthermore, we propose a simple approach to control the strength of the contrastive decoding scheme by using the conditional entropy of the next-token distribution of the model (Section 2.3). Our experimental results show that DeCoRe significantly improves the accuracy of the model in tasks requiring contextual faithfulness and in some factual recall and reasoning tasks.

**Limitations.** While DeCoRe improves the performance of the base model across most tasks, there is no "free lunch"; existing baselines may still produce more accurate results than DeCoRe in specific tasks (e.g., ITI in TruthfulQA or CAD in NQ-Swap). However, these baselines often offer limited improvements or may even generate less accurate responses in other tasks. We also observed that DeCoRe offers only marginal enhancements in factual recall tasks, suggesting that retrieval heads may not play a primary role in factual recall except for information transfer. Finally, while we propose using the conditional entropy of the model's next-token distribution to control the contrastive decoding scheme in DeCoRe, semantic-based methods of uncertainty quantification may also be used (Farquhar et al., 2024).

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

# Appendix

# Table of Contents

## A    REPRODUCIBILITY STATEMENT


Figure 5: Retrieval scores of the Retrieval Heads with non-zero retrieval scores.

## B  ETHICS STATEMENT

Our proposed method, DeCoRe, aims to mitigate hallucinations in LLMs, particularly in tasks where contextual faithfulness are critical. By improving the reliability of LLMs, DeCoRe has the potential to reduce the risks associated with incorrect or misleading information generation.

Despite these positive intentions, there is a potential for DeCoRe to be misused. For example, its ability to suppress contextual hallucinations may be exploited to generate more convincing but misleading content by providing it with factually incorrect or unverified contextual documents.

To mitigate these risks, we have open-sourced our implementation to facilitate broader scrutiny from the community. Furthermore, we recommend that DeCoRe be applied with caution in sensitive domains where even small inaccuracies may have significant consequences, such as clinical and legal domains.

## C  RETRIEVAL HEADS

### C.1  EXTRACTION OF RETRIEVAL HEADS

We follow the procedure provided by Wu et al. (2024)[4] which defines the retrieval score of attention heads as the ratio of successful copy-paste operations. They propose to calculate the retrieval score by compiling three sets of Needle-in-a-Haystack samples (Kamradt, 2023). Given a question $q$ and its corresponding answer $k$ (the needle), we insert $k$ in a given context $x$ (the haystack) at a random position index range $i_q$. The language model is then tasked with answering $q$ based on the haystack with the inserted needle. We set $q$ and $k$ unique and irrelevant with the given long context, ensuring that if an answer is correctly generated, it is indeed copied from the context, not from the model's internal knowledge. Retrieval score of head $h$ is defined as:

$$\text{retrieval\_score}_h = \frac{|g_h \cap k|}{|k|}$$

Where $g_h$ is the set of tokens copy-pasted by head $h$. Retrieval score signifies the attention head ability to recall tokens from the given context, and can be used as a metric to identify retrieval heads in transformer-based LLMs.

Table 6: Retrieval Scores of the Retrieval Heads of each model.

| Retrieval Head ID | Meta-Llama-3-8B | Meta-Llama-3-8B-Instruct | Meta-Llama-3-70B-Instruct | Mistral-7B-Instruct-v0.3 | Qwen2-7B-Instruct |
|---|---|---|---|---|---|
| 1 | 0.9341 | 0.9447 | 0.9172 | 0.8741 | 0.7746 |
| 10 | 0.4666 | 0.4421 | 0.3844 | 0.3167 | 0.3487 |
| 20 | 0.2927 | 0.2743 | 0.1874 | 0.1951 | 0.1986 |
| 30 | 0.1347 | 0.1421 | 0.1310 | 0.1457 | 0.1243 |
| 40 | 0.1074 | 0.1131 | 0.1112 | 0.1115 | 0.1077 |
| 50 | 0.0881 | 0.0916 | 0.0914 | 0.0944 | 0.0843 |
| 60 | 0.0735 | 0.0751 | 0.0867 | 0.0852 | 0.0703 |
| 70 | 0.0623 | 0.0659 | 0.0814 | 0.0751 | 0.0620 |
| 80 | 0.0572 | 0.0604 | 0.0630 | 0.0704 | 0.0524 |
| 90 | 0.0491 | 0.0513 | 0.0571 | 0.0641 | 0.0412 |
| 100 | 0.0433 | 0.0452 | 0.0526 | 0.0538 | 0.0352 |

Table 7: Performance comparison of Llama3-8B-Instruct with different number of masked retrieval heads on faithfulness evaluation tasks.

| Model | Masked Retrieval Heads | XSum | | | MemoTrap | | IFEval | | NQ-Open | NQ-Swap |
|---|---|---|---|---|---|---|---|---|---|---|
| | | ROUGE-L ↑ | BERTScore-F1 ↑ | factKB ↑ | Macro Acc ↑ | Micro Acc ↑ | Prompt Acc ↑ | Instruct Acc ↑ | EM ↑ | EM ↑ |
| | *0 (Baseline)* | 19.90 | 67.23 | 47.61 | 65.86 | 64.40 | 70.24 | 78.30 | 69.68 | 60.62 |
| | 10 | 20.51 | 67.33 | 36.56 | 66.76 | 65.89 | 62.66 | 72.90 | 64.26 | 42.92 |
| | 20 | 20.52 | 67.07 | 34.89 | 64.44 | 63.96 | 63.77 | 73.74 | 62.30 | 43.57 |
| | 30 | 20.21 | 66.49 | 29.70 | 65.92 | 64.12 | 61.74 | 72.54 | 63.24 | 46.48 |
| Llama3-8B-Instruct | 40 | 19.92 | 66.24 | 26.72 | 66.83 | 64.83 | 58.41 | 68.94 | 62.79 | 46.73 |
| | 50 | 20.05 | 66.47 | 25.97 | 68.08 | 67.07 | 55.08 | 66.91 | 62.49 | 44.77 |
| | 60 | 20.05 | 66.54 | 23.33 | 68.49 | 67.03 | 55.27 | 67.15 | 62.90 | 44.23 |
| | 70 | 19.42 | 66.14 | 24.55 | 67.88 | 65.89 | 56.01 | 68.23 | 63.01 | 46.97 |
| | 80 | 19.13 | 64.53 | 22.40 | 64.72 | 62.23 | 55.08 | 67.63 | 60.45 | 43.62 |
| | 90 | 19.46 | 64.39 | 21.12 | 63.77 | 61.28 | 54.16 | 66.55 | 57.97 | 40.77 |
| | 100 | 19.54 | 62.47 | 17.13 | 60.02 | 56.95 | 47.50 | 59.47 | 56.61 | 39.02 |

## C.2 RETRIEVAL SCORES

As shown in Figure 5, the retrieval scores for each model follow a similar pattern across all examined LLM variants. According to Wu et al. (2024), an attention head can be considered a retrieval head if it performs a copy-paste operation at least 10% of the time, which corresponds to a retrieval score of 0.1. In all the models evaluated, the retrieval scores drop below 0.1 just before reaching the 50th retrieval head. This indicates that beyond this number, the attention heads may not be reliably performing retrieval tasks. Table 6 provides the precise retrieval scores for selected heads in each model.

To ensure the robustness of our experiments, we extended the masking of retrieval heads up to the 100th retrieval head for each model, even though the data suggest that heads beyond the 50th have minimal retrieval ability. This conservative approach ensures that we comprehensively account for all potential retrieval heads during the contrastive decoding process.

# D PERFORMANCE OF BASELINE MODEL WITH MASKED HEADS

## D.1 RATIONALE

DeCoRe operates under the assumption that masking retrieval heads would cause hallucinations in LLMs. Therefore, the expected behaviour is that the performance of the LLM would go down the more retrieval heads that are masked.

## D.2 FAITHFULNESS

Figure 6a illustrates the contrasting effects of masking retrieval heads (blue) and random heads (orange) on faithfulness evaluation tasks across XSum, MemoTrap, open-book NQ, and NQ-Swap.

In XSum, masking retrieval heads results in a sharp decline in factKB scores ($r_{ret} = -0.93$), indicating the critical role of retrieval heads in maintaining factual consistency in summarisation. Masking random heads also causes a gradual decline ($r_{random} = -0.94$), however, the variance is high which suggests that retrieval heads are more important for contextual faithfulness.

For MemoTrap, masking retrieval heads shows a moderate correlation with the macro-averaged accuracy ($r_{\text{ret}} = -0.43$), while masking random heads surprisingly improves performance ($r_{\text{random}} =$

---
[4]https://github.com/nightdessert/Retrieval_Head

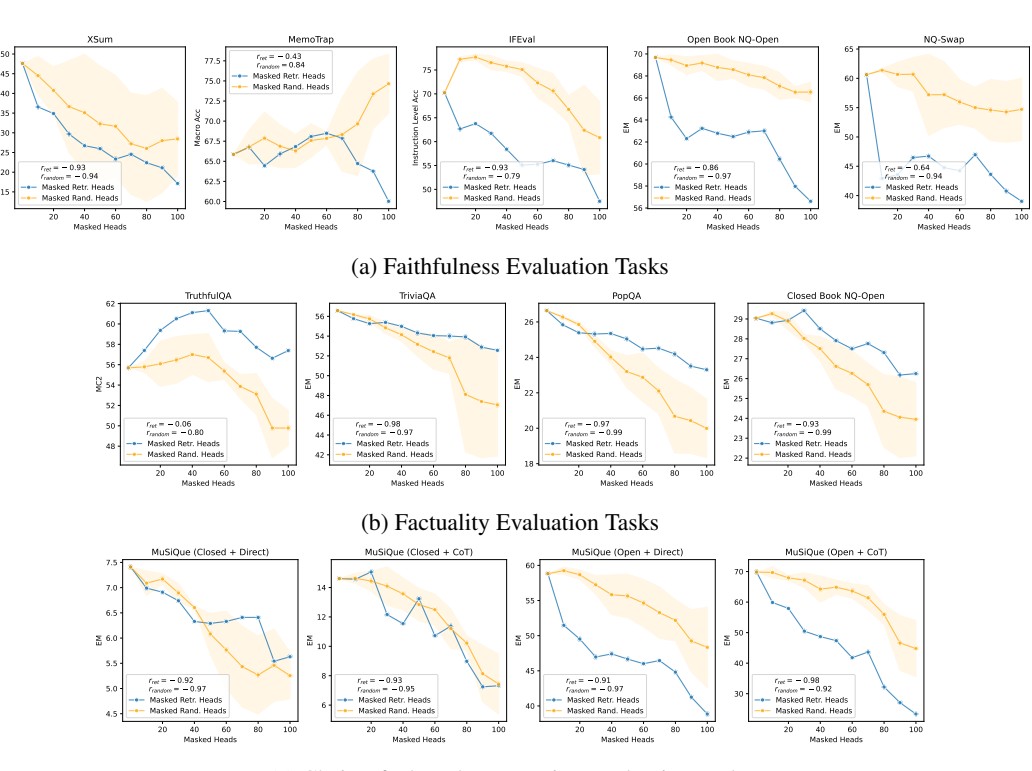

(a) Faithfulness Evaluation Tasks

(b) Factuality Evaluation Tasks

(c) Chain-of-Thought Reasoning Evaluation Tasks

Figure 6: Correlation between the number of masked retrieval heads or random heads and performance of Llama3-8B-Instruct with DeCoRe entropy on faithfulness (a), factuality (b), and Chain-of-Thought reasoning (c) evaluation tasks. The correlations are quantified by the Pearson Correlation Coefficient $r$ for each plot. Detailed results are listed in Table 7, Table 8, Table 9, Table 10, Table 11, and Table 12.

Table 8: Performance comparison of Llama3-8B-Instruct with different numbers of masked random heads on faithfulness evaluation tasks.

| Model | Masked Retrieval Heads | XSum | | | MemoTrap | | IFEval | | NQ-Open | NQ-Swap |
|---|---|---|---|---|---|---|---|---|---|---|
| | | ROUGE-L ↑ | BERTScore-F1 ↑ | factKB ↑ | Macro Acc ↑ | Micro Acc ↑ | Prompt Acc ↑ | Instruct Acc ↑ | EM ↑ | EM ↑ |
| | 0 (Baseline) | 19.90 | 67.23 | 47.61 | 65.86 | 64.40 | 70.24 | 78.30 | 69.68 | 60.62 |
| Llama3-8B-Instruct | 10 | 20.09 $_{\pm 0.21}$ | 67.07 $_{\pm 0.32}$ | 44.52 $_{\pm 4.86}$ | 66.79 $_{\pm 2.11}$ | 65.16 $_{\pm 2.61}$ | 68.64 $_{\pm 0.77}$ | 77.14 $_{\pm 0.39}$ | 69.45 $_{\pm 0.46}$ | 61.39 $_{\pm 0.24}$ |
| | 20 | 20.00 $_{\pm 0.15}$ | 66.80 $_{\pm 0.46}$ | 40.77 $_{\pm 5.98}$ | 67.89 $_{\pm 3.24}$ | 66.54 $_{\pm 4.43}$ | 69.50 $_{\pm 0.93}$ | 77.66 $_{\pm 0.68}$ | 68.94 $_{\pm 0.81}$ | 60.67 $_{\pm 2.08}$ |
| | 30 | 19.87 $_{\pm 0.18}$ | 66.61 $_{\pm 0.89}$ | 36.65 $_{\pm 11.64}$ | 66.88 $_{\pm 2.66}$ | 65.29 $_{\pm 3.71}$ | 68.27 $_{\pm 1.36}$ | 76.58 $_{\pm 1.45}$ | 69.18 $_{\pm 0.66}$ | 60.70 $_{\pm 2.87}$ |
| | 40 | 19.63 $_{\pm 0.09}$ | 66.55 $_{\pm 1.12}$ | 35.09 $_{\pm 14.85}$ | 66.29 $_{\pm 2.05}$ | 63.83 $_{\pm 3.39}$ | 67.59 $_{\pm 1.34}$ | 75.86 $_{\pm 1.20}$ | 68.78 $_{\pm 1.19}$ | 57.19 $_{\pm 6.92}$ |
| | 50 | 19.59 $_{\pm 0.19}$ | 66.34 $_{\pm 1.23}$ | 32.25 $_{\pm 14.71}$ | 67.59 $_{\pm 2.09}$ | 64.76 $_{\pm 3.84}$ | 66.23 $_{\pm 1.98}$ | 75.18 $_{\pm 1.26}$ | 68.57 $_{\pm 0.80}$ | 57.21 $_{\pm 5.62}$ |
| | 60 | 19.28 $_{\pm 0.77}$ | 66.02 $_{\pm 1.52}$ | 31.67 $_{\pm 12.94}$ | 67.85 $_{\pm 0.80}$ | 63.99 $_{\pm 1.09}$ | 62.97 $_{\pm 2.82}$ | 72.30 $_{\pm 3.11}$ | 68.10 $_{\pm 1.04}$ | 55.97 $_{\pm 3.79}$ |
| | 70 | 19.48 $_{\pm 0.53}$ | 65.81 $_{\pm 1.67}$ | 27.20 $_{\pm 12.83}$ | 68.33 $_{\pm 4.57}$ | 64.51 $_{\pm 4.95}$ | 60.87 $_{\pm 4.41}$ | 70.74 $_{\pm 3.47}$ | 67.85 $_{\pm 1.04}$ | 55.00 $_{\pm 3.48}$ |
| | 80 | 18.96 $_{\pm 0.94}$ | 64.92 $_{\pm 0.94}$ | 26.02 $_{\pm 13.42}$ | 69.66 $_{\pm 6.45}$ | 66.40 $_{\pm 7.16}$ | 56.87 $_{\pm 4.16}$ | 66.79 $_{\pm 2.98}$ | 67.08 $_{\pm 1.21}$ | 54.59 $_{\pm 5.23}$ |
| | 90 | 17.55 $_{\pm 1.19}$ | 61.85 $_{\pm 4.91}$ | 28.00 $_{\pm 13.27}$ | 73.39 $_{\pm 4.35}$ | 70.71 $_{\pm 4.93}$ | 50.96 $_{\pm 10.71}$ | 62.39 $_{\pm 9.58}$ | 66.53 $_{\pm 0.49}$ | 54.26 $_{\pm 5.17}$ |
| | 100 | 17.13 $_{\pm 1.17}$ | 61.61 $_{\pm 6.05}$ | 28.46 $_{\pm 9.30}$ | 74.65 $_{\pm 3.67}$ | 72.02 $_{\pm 4.25}$ | 48.92 $_{\pm 8.04}$ | 60.67 $_{\pm 7.43}$ | 66.54 $_{\pm 0.91}$ | 54.71 $_{\pm 5.34}$ |

Table 9: Performance comparison of Llama3-8B-Instruct with different number of masked retrieval heads on factuality evaluation tasks.

| Model | Masked Retrieval Heads | TruthfulQA (MC) | | | TriviaQA | PopQA | NQ-Open |
|---|---|---|---|---|---|---|---|
| | | MC1 ↑ | MC2 ↑ | MC3 ↑ | EM ↑ | EM ↑ | EM ↑ |
| | *Baseline* | 39.41 | 55.69 | 30.31 | 56.58 | 26.64 | 29.04 |
| Llama3-8B-Instruct | 10 | 39.17 | 57.40 | 31.57 | 55.77 | 25.84 | 28.81 |
| | 20 | 40.27 | 59.37 | 33.24 | 55.26 | 25.39 | 28.93 |
| | 30 | 40.51 | 60.51 | 33.30 | 55.39 | 25.32 | 29.42 |
| | 40 | 41.49 | 61.11 | 34.00 | 54.99 | 25.35 | 28.51 |
| | 50 | 41.00 | 61.31 | 33.63 | 54.32 | 25.04 | 27.91 |
| | 60 | 39.29 | 59.32 | 32.48 | 54.05 | 24.47 | 27.50 |
| | 70 | 38.80 | 59.27 | 32.47 | 54.01 | 24.52 | 27.76 |
| | 80 | 36.23 | 57.71 | 30.64 | 53.92 | 24.19 | 27.31 |
| | 90 | 35.86 | 56.63 | 30.17 | 52.89 | 23.51 | 26.18 |
| | 100 | 36.47 | 57.39 | 31.08 | 52.56 | 23.30 | 26.25 |

Table 10: Performance comparison of Llama3-8B-Instruct with different numbers of masked random heads on factuality evaluation tasks.

| Model | Masked Retrieval Heads | TruthfulQA (MC) | | | TriviaQA | PopQA | NQ-Open |
|---|---|---|---|---|---|---|---|
| | | MC1 ↑ | MC2 ↑ | MC3 ↑ | EM ↑ | EM ↑ | EM ↑ |
| | *Baseline* | 39.41 | 55.69 | 30.31 | 56.58 | 21.10 | 29.04 |
| Llama3-8B-Instruct | 10 | $38.84_{\pm0.71}$ | $55.79_{\pm0.53}$ | $30.38_{\pm0.46}$ | $56.17_{\pm0.03}$ | $25.96_{\pm0.18}$ | $29.27_{\pm0.10}$ |
| | 20 | $38.51_{\pm0.35}$ | $56.09_{\pm2.21}$ | $30.34_{\pm0.86}$ | $55.75_{\pm0.33}$ | $25.63_{\pm0.25}$ | $28.89_{\pm0.46}$ |
| | 30 | $37.58_{\pm1.12}$ | $56.47_{\pm2.30}$ | $30.21_{\pm1.01}$ | $54.84_{\pm0.58}$ | $25.52_{\pm0.16}$ | $28.03_{\pm0.20}$ |
| | 40 | $37.37_{\pm0.57}$ | $57.00_{\pm1.94}$ | $30.24_{\pm0.51}$ | $54.14_{\pm0.65}$ | $25.24_{\pm0.15}$ | $27.51_{\pm0.61}$ |
| | 50 | $37.17_{\pm1.56}$ | $56.70_{\pm2.36}$ | $29.85_{\pm1.58}$ | $53.17_{\pm1.22}$ | $25.07_{\pm0.22}$ | $26.61_{\pm1.14}$ |
| | 60 | $35.86_{\pm1.41}$ | $55.37_{\pm0.82}$ | $28.87_{\pm0.80}$ | $52.43_{\pm1.77}$ | $24.54_{\pm0.54}$ | $26.26_{\pm1.14}$ |
| | 70 | $34.68_{\pm0.31}$ | $53.87_{\pm1.16}$ | $27.63_{\pm0.66}$ | $51.79_{\pm1.59}$ | $24.50_{\pm0.58}$ | $25.70_{\pm1.07}$ |
| | 80 | $33.05_{\pm2.36}$ | $53.12_{\pm2.02}$ | $26.56_{\pm2.03}$ | $48.11_{\pm5.82}$ | $24.52_{\pm1.01}$ | $24.36_{\pm1.83}$ |
| | 90 | $30.80_{\pm2.20}$ | $49.78_{\pm2.91}$ | $24.79_{\pm1.56}$ | $47.39_{\pm5.68}$ | $24.14_{\pm0.98}$ | $24.05_{\pm2.03}$ |
| | 100 | $30.07_{\pm0.90}$ | $49.78_{\pm1.74}$ | $24.44_{\pm0.76}$ | $47.04_{\pm5.17}$ | $24.05_{\pm0.76}$ | $23.96_{\pm1.84}$ |

0.84). This implies that retrieval heads are essential for instruction-following, while random heads may not play as crucial a role and can even hinder performance.

In NQ Open and NQ Swap, the EM score drops significantly when retrieval heads are masked with a strong correlation score ($r_{\text{ret}} = -0.86$ and $r_{\text{ret}} = -0.64$), confirming their importance in open-book QA tasks. In both tasks, masking random heads also degrades performance, with stronger negative correlation ($r_{\text{random}} = -0.97$ and $r_{\text{random}} = -0.94$ respectively).

Despite the more significant performance drop when masking retrieval heads, the correlation coefficient is lower than that for random heads. This is due to the concentrated decline in performance after masking the top 10 retrieval heads. In contrast, performance degrades more gradually when random heads are masked, resulting in a stronger linear correlation. This pattern suggests that masking just the top retrieval heads can already significantly impair the model's ability to remain faithful to the context. Additionally, the more retrieval heads that are masked, the greater the performance drop, indicating that retrieval heads play a key role in maintaining task-specific faithfulness.

### D.3 FACTUALITY

Figure 6b shows the effect of masking retrieval heads (blue) and random heads (orange) on factual recall tasks across TruthfulQA, TriviaQA, PopQA, and NQ Closed.

In TruthfulQA, masking retrieval heads has a negligible effect on the MC2 score ($r_{ret} = -0.06$), while masking random heads shows a moderate negative correlation ($r_{random} = -0.80$). This suggests that retrieval heads do not play a major role in answering truthful questions, and the decline in performance when masking random heads could be due to their broader influence on the model's general predictive capabilities.

In contrast, for TriviaQA, PopQA, and NQ Closed, both masking retrieval heads and random heads result in significant performance drops, with strong negative correlations observed in all tasks. The

Table 11: Performance comparison of Llama3-8B-Instruct with different number of masked retrieval heads on MuSiQue, a multi-hop reasoning dataset, with and without CoT prompting in both closed-book and open-book settings.

| Model | Masked Retrieval Heads | MuSiQue without CoT | | MuSiQue with CoT | |
|---|---|---|---|---|---|
| | | Closed Book | Open Book | Closed Book | Open Book |
| | *Baseline* | 7.41 | 58.83 | 14.61 | 69.84 |
| Llama3-8B-Instruct | 10 | 6.99 | 51.47 | 14.56 | 59.87 |
| | 20 | 6.91 | 49.52 | 15.06 | 57.92 |
| | 30 | 6.74 | 46.96 | 12.16 | 50.48 |
| | 40 | 6.33 | 47.41 | 11.54 | 48.70 |
| | 50 | 6.29 | 46.67 | 13.24 | 47.37 |
| | 60 | 6.33 | 46.01 | 10.72 | 41.79 |
| | 70 | 6.41 | 46.46 | 11.38 | 43.65 |
| | 80 | 6.41 | 44.81 | 8.98 | 32.19 |
| | 90 | 5.54 | 41.25 | 7.24 | 27.06 |
| | 100 | 5.63 | 38.85 | 7.32 | 23.34 |

Table 12: Performance comparison of Llama3-8B-Instruct with different numbers of masked random heads on MuSiQue, a multi-hop reasoning dataset, with and without CoT prompting in both closed-book and open-book settings.

| Model | Masked Random Heads | MuSiQue without CoT | | MuSiQue with CoT | |
|---|---|---|---|---|---|
| | | Closed Book | Open Book | Closed Book | Open Book |
| | *Baseline* | 7.41 | 58.83 | 14.61 | 69.84 |
| Llama3-8B-Instruct | 10 | $7.09_{\pm0.24}$ | $59.25_{\pm0.53}$ | $14.63_{\pm0.35}$ | $69.70_{\pm1.81}$ |
| | 20 | $7.17_{\pm0.10}$ | $58.67_{\pm0.68}$ | $14.44_{\pm0.68}$ | $67.94_{\pm0.81}$ |
| | 30 | $6.90_{\pm0.19}$ | $57.23_{\pm1.32}$ | $14.09_{\pm1.30}$ | $67.19_{\pm2.42}$ |
| | 40 | $6.61_{\pm0.02}$ | $55.83_{\pm2.82}$ | $13.57_{\pm1.09}$ | $64.27_{\pm4.28}$ |
| | 50 | $6.08_{\pm0.41}$ | $55.65_{\pm3.12}$ | $12.84_{\pm1.10}$ | $64.87_{\pm2.34}$ |
| | 60 | $5.76_{\pm0.77}$ | $54.64_{\pm3.36}$ | $12.49_{\pm1.06}$ | $63.65_{\pm2.38}$ |
| | 70 | $5.43_{\pm0.80}$ | $53.28_{\pm3.66}$ | $11.20_{\pm1.34}$ | $61.40_{\pm3.96}$ |
| | 80 | $5.27_{\pm0.77}$ | $52.19_{\pm2.95}$ | $10.22_{\pm0.49}$ | $55.98_{\pm3.28}$ |
| | 90 | $5.46_{\pm0.72}$ | $49.25_{\pm4.41}$ | $8.14_{\pm1.92}$ | $46.59_{\pm8.97}$ |
| | 100 | $5.25_{\pm0.46}$ | $48.34_{\pm5.71}$ | $7.43_{\pm2.04}$ | $44.79_{\pm9.19}$ |

differences between masking the retrieval heads and random heads are not as stark as in faithfulness tasks. For instance, in TriviaQA, masking retrieval heads leads to a performance decline ($r_{ret} = -0.98$), but masking random heads also has a similar effect ($r_{random} = -0.97$). This similarity suggests that in factual recall tasks, retrieval heads may not be the only determining factor.

The overall observation from these tasks is that while masking retrieval heads does lower performance, it does not have as drastic an effect as observed in faithfulness hallucination tasks. The relatively similar progression of performance degradation between masking retrieval and random heads further reinforces the idea that factual recall tasks rely on a broader mechanism, even though the masking of retrieval heads does lead to a moderate drop in performance.

### D.4 CHAIN-OF-THOUGHT

The performance of the Llama3-8B-Instruct model with different numbers of masked retrieval heads on the MuSiQue dataset, both with and without Chain-of-Thought (CoT) prompting, is shown in Figure 6c. The table compares the closed-book and open-book settings to assess the influence of CoT on model performance. In the closed-book setting without CoT prompting, masking retrieval heads leads to a gradual performance decline, with scores decreasing from 7.41 (baseline) to 5.63 (with 100 masked heads). This indicates that the model's ability to reason through multiple hops is compromised as retrieval heads are removed. The decline of performance in the open-book setting without CoT prompting further indicates the importance of retrieval heads in open-book QA tasks.

The inclusion of CoT prompts generally boosts performance in both closed-book and open-book settings. Similar to the setup without CoT prompting, masking retrieval heads in the CoT setup decreases the performance gradually. Interestingly, in the CoT + open-book setup, masking only the top 20 retrieval heads leads to a performance lower than without using CoT. This suggests that retrieval heads are crucial for maintaining the model's ability to chain reasoning steps across multiple hops, particularly when the reasoning steps have to be grounded in contextual knowledge.

## E    Additional TruthfulQA Generation Evaluation

### E.1    Evaluation of Non-rejection Responses

Table 13: TruthfulQA Generation Evaluation excluding the rejected instances. Notice the rate of rejection that is very high on the instruction-tuned Llama3-8b.

| Model | %Reject ↓ | %T ∩ R̄ ↑ | %I ∩ R̄ | %T ∩ I ∩ R̄ ↑ |
|---|---|---|---|---|
| Llama3-8b-Instruct | 43.94 | 65.50 | 94.54 | 60.04 |
| + ITI (Li et al., 2024b) | 25.46 | 83.25 | 96.06 | 79.47 |
| + DoLA (low) (Chuang et al., 2023) | 45.04 | 64.81 | 94.65 | 59.69 |
| + DoLA (high) (Chuang et al., 2023) | 44.92 | 65.11 | 93.78 | 58.89 |
| + AD (Chen et al., 2024) | 43.82 | 65.14 | 94.55 | 59.69 |
| + DeCoRe static (Ours) | 41.74 | 67.02 | 95.38 | 62.39 |
| + DeCoRe entropy (Ours) | 38.68 | 65.87 | 95.61 | 61.48 |
| Llama3-70b-Instruct | 53.12 | 76.50 | 97.91 | 74.41 |
| + CD (Li et al., 2023) | 52.26 | 75.64 | 97.69 | 73.33 |
| + ITI (Li et al., 2024b) | 37.94 | 71.79 | 98.82 | 70.81 |
| + DoLA (low) (Chuang et al., 2023) | 52.88 | 76.62 | 97.92 | 74.55 |
| + DoLA (high) (Chuang et al., 2023) | 54.71 | 76.22 | 97.30 | 73.51 |
| + AD (Chen et al., 2024) | 49.33 | 75.36 | 98.31 | 73.67 |
| + DeCoRe static (Ours) | 54.96 | 74.46 | 97.01 | 71.47 |
| + DeCoRe entropy (Ours) | 56.79 | 75.35 | 96.32 | 71.67 |
| + DeCoRe entropy-small amateur (Ours) | 52.02 | 75.77 | 97.70 | 73.47 |

As shown in Table 3, we can observe that the rejection rate of Llama3 models in the TruthfulQA task (*i.e.,* the ratio of cases when the model answers with "I have no comment") is relatively high, particularly when compared to Llama2 models (Touvron et al., 2023) reported by previous studies (Li et al., 2024b; Chuang et al., 2023). To get a better understanding of how the model performs, we also reported the evaluation metrics that are based only on non-rejection answers in Table 13. This results can help us to roughly understand how the model would perform when it's not rejecting to answer. However, it is important to note that we cannot compare the performance of the decoding strategies to one another because the set of questions that are being answered are different depending on whether the decoding strategy choose to answer them or not.

### E.2    Evaluation Cost

The fine-tuning of two `davinci-002` models (to measure truthfulness and informativeness) costs approximately $43. While each run of evaluation is approximately $0.8.

## F    Correlation between Length-normalised Entropy and Correctness

### F.1    Rationale

One motivation to use the length-normalised entropy as a measure of how much information to contrast relies heavily on the premise that length-normalised entropy is a reliable proxy of answer correctness. To verify this assumption, we conducted statistical tests (Student's T-test (Student, 1908) and a Mann-Whitney U-test (Mann & Whitney, 1947)) and to determine whether the length-normalised entropy of correct answers tends to be lower than that of incorrect answers.

### F.2    Statistical Tests

The results of these statistical tests, as presented in Table 15, show that the differences in entropy between correct and incorrect answers are statistically significant across all models, with low p-values for both tests. The baseline model yields a T-test statistic of 11.75 and a p-value of $2.57 \times 10^{-31}$, confirming that the entropy of correct answers is significantly lower. This trend holds for the DoLa and DeCoRe entropy models, with both tests indicating a strong separation between the entropy distributions of correct and incorrect answers. The Mann-Whitney U-test results further corroborate this finding, providing consistent statistics and p-values below $10^{-24}$ for all models.

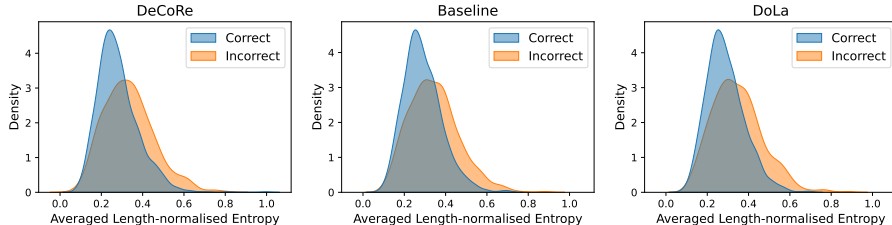

(a) Density plot showing the distribution of length-normalised entropy for correct and incorrect answers across different models (DeCoRe, Baseline, and DoLa).

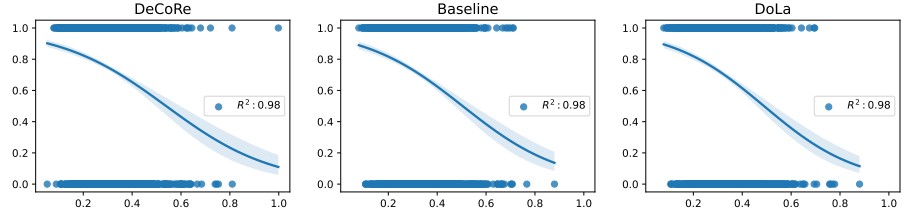

(b) Regression plot demonstrating the negative correlation between length-normalised entropy and answer correctness.

Figure 7: Relation between length-normalised entropy and correctness in MuSiQue CoT generation. Entropy tends to be negatively correlated with the final answer correctness (*i.e.,* the lower the length-normalised entropy, the more likely that the answer is correct.).

These results validate the hypothesis that lower length-normalised entropy is a meaningful indicator of answer correctness, supporting its use in contrastive decoding through DeCoRe.

The accompanying Figure 7a illustrates the distribution of length-normalised entropy for correct and incorrect answers across models (DeCoRe, Baseline, and DoLa). Correct answers (in blue) tend to have lower entropy, whereas incorrect answers (in orange) exhibit higher entropy. This visualisation aligns with the statistical tests, highlighting the difference between correct and incorrect answers based on their entropy values.

Table 14: Averaged Length-Normalised Predictive Entropy of the correct and incorrect answer by DeCoRe Entropy. All values are scaled by $10^2$. Lower values indicate less overall uncertainty. Generally, the length-normalised entropy of correct answers is lower than the incorrect ones, indicating the importance of the model's certainty in generating a correct answer.

Table 15: Results of the Student's T-test and Mann-Whitney U-test comparing the length-normalised entropy of correct and incorrect answers across different models. The low p-values across all models confirm that correct answers generally have lower entropy compared to incorrect ones, validating the use of entropy as a proxy for answer correctness.

|  | MuSiQue (Closed) | MuSiQue (Open) |
|---|---|---|
| Correct | 31.74 | 27.99 |
| Incorrect | 43.91 | 33.32 |

| Model | T-test | | U-test | |
|---|---|---|---|---|
| | Statistics | p-value | Statistics | p-value |
| Baseline | 11.75 | $2.57 \times 10^{-31}$ | $4.31 \times 10^5$ | $8.36 \times 10^{-26}$ |
| DoLa | 12.52 | $3.51 \times 10^{-35}$ | $4.28 \times 10^5$ | $3.66 \times 10^{-28}$ |
| DeCoRe entropy | 11.01 | $7.43 \times 10^{-28}$ | $4.05 \times 10^5$ | $3.43 \times 10^{-24}$ |

## F.3 REGRESSION

To further quantify the relationship between length-normalised entropy and answer correctness, we calculated the McFadden's pseudo-$R^2$ (McFadden et al., 1973) for the logistic regression models fitted across the different setups (DeCoRe, Baseline, and DoLa). As shown in the regression plots (Figure 7b), all three models demonstrate a high pseudo-$R^2$ value of 0.98, indicating a strong negative relationship between entropy and correctness. This high pseudo-$R^2$ value suggests that the length-normalised entropy is highly predictive of answer correctness, further validating the use of entropy as a reliable proxy for contrasting model outputs.

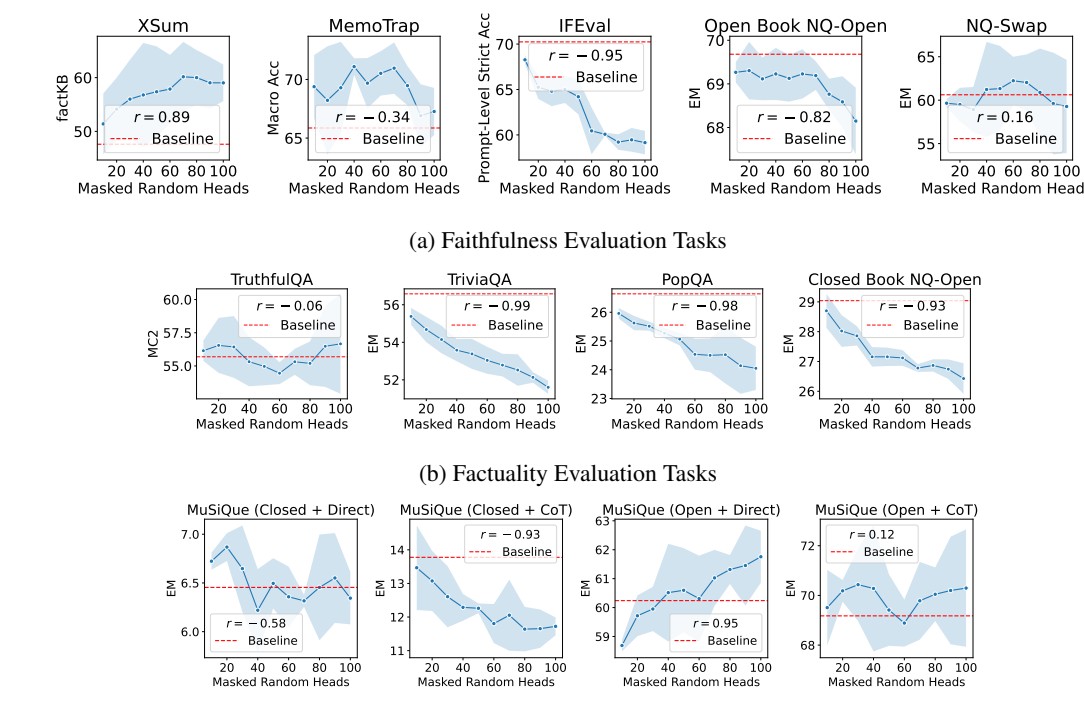

(a) Faithfulness Evaluation Tasks

(b) Factuality Evaluation Tasks

(c) Chain-of-Thought Reasoning Evaluation Tasks

Figure 8: Correlation between the number of masked random heads and performance of Llama3-8B-Instruct with DeCoRe$_{entropy}$ on each task. The correlations are quantified by the Pearson Correlation Coefficient $r$ for each plot. Detailed results are listed in Table 16 and Table 18.

# G  DETAILED RESULTS OF MASKED HEADS ABLATION STUDY

## G.1  EFFECT OF RANDOM HEAD MASKING ON TASK PERFORMANCE OF DECORE

As shown in Figure 8, the performance of DeCoRe$_{entropy}$ exhibits different patterns when masking random attention heads compared to the targeted masking of retrieval heads in Section 4. A key observation is that the standard deviation is much larger across most tasks, indicating higher variability in performance when random heads are masked. This variability indicates that DeCoRe$_{entropy}$ cannot benefit only from masking any random attention heads.

In XSum, we still observe a positive correlation between the number of masked random heads and task performance, though the correlation ($r = 0.89$) is weaker than that seen when masking retrieval heads. This suggests that masking random heads can still improve contextual faithfulness in summarisation, though the impact is less pronounced especially when considering the highest possible performance achieved by masking random heads.

MemoTrap, which exhibits a strong positive correlation when masking retrieval heads, now shows a weak negative correlation ($r = -0.34$). This shift implies that random masking does not improve the model's instruction-following capabilities, and further suggests that the improvements seen were due to the targeted masking of retrieval heads. This supports the idea that retrieval heads play a key role in tasks requiring the faithful execution of instructions.

Similar to the results of masking retrieval heads, random head masking exhibits a negative correlation on IFEval. Interestingly, Open Book NQ continues to show a strong negative correlation ($r = -0.82$), much like in the previous section. This reinforces the idea that retrieval mechanisms when handling open-book QA tasks, where the model must balance contextual and parametric knowledge, differ from a simple induction mechanism. In contrast, NQ-Swap and TruthfulQA show little to no correlation, indicating that masking random heads does not significantly impact performance on these tasks.

Table 16: Ablation study of DeCoRe entropy on faithfulness hallucination tasks with varying numbers of masked retrieval heads.

| Model | Masked Retrieval Heads | XSum | | | MemoTrap | | IFEval | | NQ-Open | NQ-Swap |
|---|---|---|---|---|---|---|---|---|---|---|
| | | ROUGE-L ↑ | BERTScore-F1 ↑ | factKB ↑ | Macro Acc ↑ | Micro Acc ↑ | Prompt Acc ↑ | Instruct Acc ↑ | EM ↑ | EM ↑ |
| | *0 (Baseline)* | 19.90 | 67.23 | 47.61 | 65.86 | 64.40 | 70.24 | 78.30 | 69.68 | 60.62 |
| | 10 | 19.45 | 67.08 | 57.50 | 68.81 | 66.60 | 68.39 | 76.38 | 70.66 | 66.08 |
| | 20 | 19.61 | 67.18 | 57.53 | 69.39 | 68.37 | 67.10 | 75.54 | 70.24 | 65.55 |
| | 30 | 19.62 | 67.48 | 59.75 | 70.14 | 70.50 | 62.11 | 72.30 | 70.17 | 65.15 |
| | 40 | 19.70 | 67.42 | 60.65 | 70.46 | 71.09 | 62.29 | 72.42 | 69.83 | 64.96 |
| Llama3-8B-Instruct | 50 | 19.37 | 67.15 | 62.88 | 71.27 | 71.68 | 61.92 | 72.06 | 69.94 | 64.75 |
| | 60 | 19.40 | 67.18 | 64.27 | 71.59 | 71.76 | 58.60 | 69.54 | 69.57 | 64.41 |
| | 70 | 19.51 | 67.30 | 61.32 | 71.90 | 71.80 | 56.93 | 68.94 | 68.51 | 61.53 |
| | 80 | 19.40 | 67.57 | 64.67 | 72.52 | 72.75 | 59.15 | 70.14 | 68.55 | 62.75 |
| | 90 | 19.45 | 67.69 | 66.10 | 74.14 | 74.87 | 59.89 | 70.74 | 68.66 | 62.64 |
| | 100 | 19.37 | 67.59 | 64.78 | 73.53 | 73.97 | 60.81 | 70.98 | 69.57 | 63.93 |
| | *0 (Baseline)* | 22.41 | 69.77 | 61.32 | 68.47 | 66.52 | 77.45 | 84.41 | 71.07 | 76.11 |
| | 10 | 22.17 | 69.64 | 62.41 | 69.17 | 67.51 | 76.34 | 83.57 | 71.75 | 78.36 |
| | 20 | 22.35 | 69.75 | 60.72 | 68.58 | 66.64 | 77.45 | 84.29 | 71.83 | 77.86 |
| | 30 | 22.03 | 69.51 | 63.91 | 70.28 | 69.52 | 78.56 | 84.89 | 72.35 | 79.10 |
| Llama3-70B-Instruct | 40 | 21.98 | 69.48 | 64.67 | 71.93 | 72.19 | 77.45 | 83.81 | 72.32 | 78.91 |
| | 50 | 21.93 | 69.47 | 65.13 | 73.75 | 73.41 | 77.63 | 84.41 | 72.54 | 79.14 |
| | 60 | 21.84 | 69.44 | 63.94 | 72.66 | 72.19 | 78.19 | 84.89 | 72.24 | 77.79 |
| | 70 | 22.03 | 69.55 | 62.96 | 71.97 | 71.96 | 76.52 | 83.69 | 72.43 | 77.62 |
| | 80 | 21.95 | 69.44 | 64.62 | 72.81 | 72.47 | 77.08 | 84.05 | 72.66 | 79.73 |
| | 90 | 21.93 | 69.40 | 65.49 | 74.07 | 73.65 | 77.26 | 83.81 | 72.39 | 79.73 |
| | 100 | 21.82 | 69.38 | 65.30 | 73.88 | 73.97 | 77.08 | 83.81 | 72.47 | 79.79 |

Table 17: Ablation study of DeCoRe entropy on faithfulness hallucination tasks with varying numbers of masked random heads.

| Model | Masked Random Heads | XSum | | | MemoTrap | | IFEval | | NQ-Open | NQ-Swap |
|---|---|---|---|---|---|---|---|---|---|---|
| | | ROUGE-L ↑ | BERTScore-F1 ↑ | factKB ↑ | Macro Acc ↑ | Micro Acc ↑ | Prompt Acc ↑ | Instruct Acc ↑ | EM ↑ | EM ↑ |
| | *0 (Baseline)* | 19.90 | 67.23 | 47.61 | 65.86 | 64.40 | 70.24 | 78.30 | 69.68 | 60.62 |
| | 10 | $20.02_{\pm0.12}$ | $67.43_{\pm0.31}$ | $51.39_{\pm5.67}$ | $69.38_{\pm2.70}$ | $68.08_{\pm2.75}$ | $68.52_{\pm0.75}$ | $76.82_{\pm0.82}$ | $69.27_{\pm0.24}$ | $59.65_{\pm0.47}$ |
| | 20 | $20.09_{\pm0.26}$ | $67.64_{\pm0.37}$ | $54.13_{\pm5.85}$ | $68.22_{\pm4.61}$ | $66.68_{\pm5.76}$ | $65.31_{\pm1.49}$ | $74.46_{\pm0.95}$ | $69.30_{\pm0.66}$ | $59.49_{\pm1.93}$ |
| | 30 | $20.06_{\pm0.11}$ | $67.78_{\pm0.53}$ | $56.00_{\pm7.34}$ | $69.29_{\pm3.91}$ | $68.77_{\pm4.88}$ | $64.76_{\pm1.87}$ | $74.26_{\pm1.63}$ | $69.11_{\pm0.49}$ | $58.91_{\pm2.61}$ |
| | 40 | $20.07_{\pm0.23}$ | $67.76_{\pm0.54}$ | $56.78_{\pm9.68}$ | $71.09_{\pm0.71}$ | $70.72_{\pm1.56}$ | $64.94_{\pm1.34}$ | $74.38_{\pm1.39}$ | $69.23_{\pm0.60}$ | $61.23_{\pm5.48}$ |
| Llama3-8B-Instruct | 50 | $20.08_{\pm0.36}$ | $67.89_{\pm0.50}$ | $57.37_{\pm8.45}$ | $69.69_{\pm2.14}$ | $69.07_{\pm3.18}$ | $64.08_{\pm1.99}$ | $73.78_{\pm1.80}$ | $69.13_{\pm0.53}$ | $61.33_{\pm4.92}$ |
| | 60 | $20.09_{\pm0.47}$ | $67.99_{\pm0.61}$ | $57.87_{\pm6.37}$ | $70.52_{\pm1.89}$ | $70.17_{\pm1.18}$ | $60.51_{\pm2.63}$ | $70.78_{\pm1.92}$ | $69.23_{\pm0.56}$ | $62.23_{\pm2.77}$ |
| | 70 | $19.83_{\pm0.47}$ | $67.96_{\pm0.54}$ | $60.16_{\pm6.49}$ | $70.96_{\pm2.19}$ | $70.76_{\pm1.90}$ | $60.14_{\pm0.21}$ | $70.90_{\pm0.42}$ | $69.19_{\pm0.33}$ | $62.03_{\pm3.23}$ |
| | 80 | $19.71_{\pm0.44}$ | $67.85_{\pm0.49}$ | $60.00_{\pm5.13}$ | $69.47_{\pm1.68}$ | $68.94_{\pm0.94}$ | $58.96_{\pm1.44}$ | $69.46_{\pm1.23}$ | $68.76_{\pm0.36}$ | $60.89_{\pm5.05}$ |
| | 90 | $19.75_{\pm0.34}$ | $67.78_{\pm0.52}$ | $59.04_{\pm4.80}$ | $66.91_{\pm2.68}$ | $66.63_{\pm3.58}$ | $59.64_{\pm1.20}$ | $69.94_{\pm0.45}$ | $68.59_{\pm0.59}$ | $59.62_{\pm5.86}$ |
| | 100 | $19.68_{\pm0.45}$ | $67.82_{\pm0.50}$ | $59.03_{\pm3.41}$ | $67.27_{\pm2.01}$ | $66.76_{\pm2.80}$ | $59.02_{\pm1.23}$ | $69.62_{\pm1.08}$ | $68.15_{\pm0.76}$ | $59.27_{\pm5.37}$ |

For factual recall tasks like TriviaQA, PopQA, and Closed Book NQ, the results are consistent with the previous section, showing strong negative correlations with increasing numbers of masked random heads. As the performance trends of masking retrieval heads and random heads are similar, this may further support the hypothesis that factual recall is not predominantly handled by attention heads. This finding aligns with previous studies (Geva et al., 2021; Meng et al., 2022), which suggest that factual recall is predominantly handled by the MLP layer within the Transformer model.

## G.2 FAITHFULNESS

Table 16 accompanies Figure 3 (top) and Table 17 accompanies Figure 8 (top).

In the case of masking retrieval heads in DeCoRe entropy (Table 16), the results show different trends depending on the type of the task. In summarisation (XSum) and instruction following (MemoTrap) tasks, we can observe an increase in performance the more retrieval heads are masked. This indicates the importance of retrieval heads in these tasks, similar to the findings mentioned in Appendix D.2.

However, the results show a different trend in open-book QA tasks (Open Book NQ-Open and NQ-Swap). In both Open Book NQ-Open and NQ-Swap, we can observe an increase in performance starting from masking 10 retrieval heads, and gradually goes down. In the case of Open Book NQ-Open, the performance is above the baseline variant until it drops below it when we mask 60 retrieval heads. While in the case of NQ-Swap, the performance remains above the baseline model even after we mask 100 retrieval heads. Albeit the differing trend, these open-book QA results are still in line with the previous findings in Appendix D.2, where the top 10 retrieval heads plays the most important role in the open-book QA tasks, with decreasing importance thereafter.

In contrast, we can observe massive standard deviation in the results of masking random heads in DeCoRe entropy shown in Table 17. This variance suggests that randomly masking heads leads to inconsistent effects across tasks, implying that not all attention heads contribute equally to model

Table 18: Ablation study of DeCoRe entropy on factuality hallucination tasks with varying numbers of masked retrieval heads.

| Model | Masked Retrieval Heads | TruthfulQA (MC) | | | TriviaQA | PopQA | NQ-Open |
|---|---|---|---|---|---|---|---|
| | | MC1 ↑ | MC2 ↑ | MC3 ↑ | EM ↑ | EM ↑ | EM ↑ |
| | *Baseline* | 39.41 | 55.69 | 30.31 | 56.58 | 26.64 | 29.04 |
| Llama3-8B-Instruct | 10 | 37.45 | 53.76 | 28.48 | 56.40 | 26.88 | 28.96 |
| | 20 | 36.96 | 54.46 | 28.95 | 56.18 | 26.74 | 28.55 |
| | 30 | 37.58 | 53.76 | 29.38 | 55.14 | 26.28 | 27.42 |
| | 40 | 36.23 | 53.62 | 29.34 | 54.73 | 25.97 | 27.91 |
| | 50 | 37.70 | 54.66 | 29.82 | 53.99 | 25.55 | 27.27 |
| | 60 | 37.21 | 54.50 | 30.21 | 53.72 | 25.39 | 27.01 |
| | 70 | 36.96 | 55.05 | 30.35 | 52.84 | 24.99 | 26.44 |
| | 80 | 38.43 | 55.86 | 30.95 | 52.19 | 24.76 | 26.44 |
| | 90 | 37.70 | 55.32 | 30.30 | 52.29 | 24.85 | 26.70 |
| | 100 | 36.60 | 54.10 | 29.61 | 52.21 | 25.09 | 26.55 |
| | *Baseline* | 49.57 | 70.60 | 37.85 | 74.77 | 40.63 | 40.08 |
| Llama3-70B-Instruct | 10 | 49.94 | 70.66 | 38.11 | 74.75 | 40.58 | 40.30 |
| | 20 | 50.31 | 70.93 | 38.35 | 74.67 | 40.46 | 40.23 |
| | 30 | 50.43 | 71.76 | 39.65 | 74.57 | 40.51 | 40.11 |
| | 40 | 50.80 | 71.54 | 39.33 | 74.58 | 40.49 | 40.08 |
| | 50 | 52.14 | 72.17 | 40.36 | 74.72 | 40.44 | 40.15 |
| | 60 | 52.88 | 72.45 | 41.64 | 74.51 | 40.30 | 40.26 |
| | 70 | 53.98 | 73.44 | 42.55 | 74.61 | 40.38 | 40.45 |
| | 80 | 53.61 | 72.98 | 41.79 | 74.65 | 40.49 | 40.30 |
| | 90 | 52.88 | 72.61 | 41.71 | 74.60 | 40.58 | 40.38 |
| | 100 | 54.10 | 72.96 | 42.86 | 74.64 | 40.49 | 40.45 |

Table 19: Ablation study of DeCoRe entropy on factuality hallucination tasks with varying numbers of masked random heads.

| Model | Masked Random Heads | TruthfulQA (MC) | | | TriviaQA | PopQA | NQ-Open |
|---|---|---|---|---|---|---|---|
| | | MC1 ↑ | MC2 ↑ | MC3 ↑ | EM ↑ | EM ↑ | EM ↑ |
| | *Baseline* | 39.41 | 55.69 | 30.31 | 56.58 | 26.64 | 29.04 |
| Llama3-8B-Instruct | 10 | $38.92_{\pm0.53}$ | $56.15_{\pm0.78}$ | $30.22_{\pm0.28}$ | $55.38_{\pm0.45}$ | $25.96_{\pm0.18}$ | $28.70_{\pm0.57}$ |
| | 20 | $39.25_{\pm0.62}$ | $56.55_{\pm2.07}$ | $30.93_{\pm0.85}$ | $54.68_{\pm0.68}$ | $25.63_{\pm0.25}$ | $28.02_{\pm0.53}$ |
| | 30 | $39.41_{\pm1.28}$ | $56.43_{\pm2.33}$ | $31.10_{\pm1.26}$ | $54.15_{\pm0.73}$ | $25.52_{\pm0.16}$ | $27.86_{\pm0.32}$ |
| | 40 | $38.84_{\pm0.75}$ | $55.32_{\pm1.85}$ | $30.39_{\pm1.03}$ | $53.58_{\pm0.59}$ | $25.27_{\pm0.17}$ | $27.16_{\pm0.33}$ |
| | 50 | $38.76_{\pm0.35}$ | $54.97_{\pm1.43}$ | $30.37_{\pm1.05}$ | $53.38_{\pm0.80}$ | $25.07_{\pm0.22}$ | $27.16_{\pm0.31}$ |
| | 60 | $38.31_{\pm0.65}$ | $54.45_{\pm0.82}$ | $29.89_{\pm0.92}$ | $53.04_{\pm0.72}$ | $24.54_{\pm0.54}$ | $27.12_{\pm0.26}$ |
| | 70 | $38.68_{\pm0.92}$ | $55.31_{\pm0.98}$ | $30.74_{\pm1.26}$ | $52.79_{\pm0.60}$ | $24.50_{\pm0.58}$ | $26.78_{\pm0.13}$ |
| | 80 | $37.58_{\pm0.65}$ | $55.19_{\pm1.65}$ | $30.05_{\pm0.45}$ | $52.52_{\pm0.84}$ | $24.52_{\pm1.01}$ | $26.87_{\pm0.21}$ |
| | 90 | $38.39_{\pm2.22}$ | $56.48_{\pm3.06}$ | $30.82_{\pm2.20}$ | $52.13_{\pm0.28}$ | $24.14_{\pm0.98}$ | $26.74_{\pm0.33}$ |
| | 100 | $38.23_{\pm2.70}$ | $56.66_{\pm3.77}$ | $31.03_{\pm2.72}$ | $51.60_{\pm0.35}$ | $24.05_{\pm0.76}$ | $26.43_{\pm0.51}$ |

performance. The less predictable effects of masking random heads further highlights the specialised role of retrieval heads in DeCoRe, particularly in maintaining task-specific faithfulness.

### G.3  FACTUALITY

Table 18 accompanies Figure 3 (bottom) and Table 19 accompanies Figure 8 (bottom).

As shown in Table 18, the results in TruthfulQA shows less clear correlation compared to other factuality evaluation tasks. For closed-book QA tasks like TriviaQA, PopQA, and Closed Book NQ-Open, a negative correlation is observed between the number of masked retrieval heads and performance. Similar negative correlations are observed when random heads are masked as shown in Table 19. The similarity in the performance degradation across both retrieval and random heads indicates that other model mechanisms might be responsible for factual recall.

### G.4  CHAIN OF THOUGHT

Table 20 accompanies Table 4 to show the performance of DeCoRe entropy when masking retrieval heads across different setups of MuSiQue, a multi-hop reasoning dataset, with and without CoT prompting, in both closed-book and open-book settings.

In the closed-book without CoT setup, we can observe a negative correlation between the number of masked retrieval heads and the performance. As more retrieval heads are masked, the performance gradually declines from the baseline across the Llama3-8B-Instruct and Llama3-70B-Instruct models, aligned with the findings in Appendix G.3.

Table 20: Performance comparison across different number of masked retrieval heads on MuSiQue, a multi-hop reasoning dataset, with and without CoT prompting in both closed-book and open-book settings.

| Model | Masked Retrieval Heads | MuSiQue without CoT | | MuSiQue with CoT | |
|---|---|---|---|---|---|
| | | Closed Book | Open Book | Closed Book | Open Book |
| | *Baseline* | 7.41 | 58.83 | 14.61 | 69.84 |
| | 10 | 7.61 | 61.98 | 13.90 | 74.47 |
| | 20 | 7.70 | 61.81 | 13.82 | 72.20 |
| | 30 | 7.70 | 61.44 | 13.61 | 71.70 |
| | 40 | 7.03 | 61.32 | 13.03 | 72.16 |
| Llama3-8B-Instruct | 50 | 7.12 | 61.32 | 12.78 | 71.62 |
| | 60 | 6.50 | 60.36 | 13.03 | 72.11 |
| | 70 | 6.21 | 59.21 | 12.83 | 71.66 |
| | 80 | 5.75 | 58.05 | 12.29 | 71.74 |
| | 90 | 6.04 | 59.54 | 12.49 | 70.87 |
| | 100 | 6.45 | 59.78 | 11.96 | 71.00 |
| | *Baseline* | 11.79 | 68.56 | 20.15 | 74.43 |
| | 10 | 11.75 | 69.22 | 20.60 | 74.76 |
| | 20 | 11.67 | 69.05 | 20.02 | 74.56 |
| | 30 | 11.50 | 68.97 | 20.31 | 74.43 |
| | 40 | 11.63 | 69.05 | 20.23 | 74.22 |
| Llama3-70B-Instruct | 50 | 11.34 | 69.38 | 20.02 | 73.60 |
| | 60 | 11.34 | 68.68 | 19.69 | 73.85 |
| | 70 | 11.34 | 69.38 | 19.40 | 74.06 |
| | 80 | 11.25 | 69.67 | 19.28 | 74.18 |
| | 90 | 11.38 | 69.51 | 19.53 | 74.47 |
| | 100 | 11.25 | 69.84 | 19.69 | 74.93 |

Table 21: Performance comparison across different numbers of masked random heads on MuSiQue, a multi-hop reasoning dataset, with and without CoT prompting in both closed-book and open-book settings.

| Model | Masked Random Heads | MuSiQue without CoT | | MuSiQue with CoT | |
|---|---|---|---|---|---|
| | | Closed Book | Open Book | Closed Book | Open Book |
| | *Baseline* | 7.41 | 58.83 | 14.61 | 69.84 |
| | 10 | $6.63_{\pm0.17}$ | $59.21_{\pm0.91}$ | $13.57_{\pm0.91}$ | $69.40_{\pm1.09}$ |
| | 20 | $6.87_{\pm0.14}$ | $59.72_{\pm0.70}$ | $13.07_{\pm0.90}$ | $70.18_{\pm0.44}$ |
| | 30 | $6.65_{\pm0.44}$ | $59.95_{\pm0.77}$ | $12.61_{\pm0.91}$ | $70.43_{\pm1.47}$ |
| | 40 | $6.22_{\pm0.42}$ | $60.52_{\pm1.69}$ | $12.29_{\pm0.40}$ | $70.28_{\pm2.53}$ |
| Llama3-8B-Instruct | 50 | $6.50_{\pm0.26}$ | $60.60_{\pm1.46}$ | $12.26_{\pm0.15}$ | $69.41_{\pm1.44}$ |
| | 60 | $6.36_{\pm0.31}$ | $60.31_{\pm1.49}$ | $11.81_{\pm0.58}$ | $68.89_{\pm0.95}$ |
| | 70 | $6.32_{\pm0.06}$ | $61.03_{\pm0.97}$ | $12.05_{\pm1.06}$ | $69.78_{\pm1.56}$ |
| | 80 | $6.45_{\pm0.54}$ | $61.32_{\pm0.50}$ | $11.64_{\pm0.66}$ | $70.05_{\pm1.08}$ |
| | 90 | $6.55_{\pm0.46}$ | $61.45_{\pm1.38}$ | $11.65_{\pm0.57}$ | $70.20_{\pm2.17}$ |
| | 100 | $6.34_{\pm0.27}$ | $61.76_{\pm0.90}$ | $11.72_{\pm0.27}$ | $70.29_{\pm2.36}$ |

In the open-book without CoT setup, there is also a negative correlation, but interestingly, the overall performance remains higher than the baseline model, which is aligned with the findings in Appendix G.2.

Interestingly the results in the closed-book with CoT setup are quite different, as masking retrieval heads does not lead to improved performance. From the results of masking retrieval heads in the baseline model (Table 11), we expect the model to perform better as DeCoRe will contrast the incorrect predictions. This may suggest that the complexity of factual recall in closed-book setup remains the same even though the model is prompted to generate intermediate reasoning steps.

Finally, the open-book with CoT setup shows an increase in performance when masking retrieval heads, even though the correlation remains negative. This is consistent with the broader trend observed in the open-book QA setup, where the model benefits from masking retrieval heads but only up to a point. Even with the negative correlation, the performance still remains higher than the baseline, indicating the utility of retrieval heads in CoT-assisted open-book tasks.

As shown in Table 21, the trend observed when masking random heads is less apparent in comparison to when masking retrieval heads. This indicates that random heads may not be as critical in these tasks.

## H   PAIRWISE STATISTICAL TESTS OF THE MAIN RESULTS

We conducted pairwise Statistical Tests between DeCoRe$_{entropy}$ and the baselines to evaluate differences. For tasks that are evaluated using the Exact Match metric, we use McNemar's Test (McNemar, 1947), with adjusted p-values calculated using the Bonferroni correction to account for multiple comparisons (Dunn, 1961). On the other hand, we use the bootstrap resampling method for tasks that are evaluated using metrics with continuous values (*i.e.,* ROUGE-L, BERTScore-F1, factKB, MC2, and MC3).

### H.1   FAITHFULNESS

Table 22: Pairwise test statistics for the performance of DeCoRe$_{entropy}$ against different baselines on faithfulness evaluation tasks. We use McNemar's Test for analysing Accuracy and EM metrics, and bootstrap resampling for assessing the significance of ROUGE-L, BERTScore-F1, and factKB (* $p < 0.05$, ** $p < 0.01$, *** $p < 0.001$, **** $p < 0.0001$).

| Model | XSum | | | MemoTrap | IFEval | NQ-Open | NQ-Swap |
|---|---|---|---|---|---|---|---|
| | ROUGE-L | BERTScore-F1 | factKB | Micro Acc | Prompt Acc | EM | EM |
| **Llama3-8b-Instruct** | | | | | | | |
| DeCoRe$_{entropy}$ > Greedy | −0.45* | 0.00**** | 0.18**** | 85.85**** | 1.78 | 8.01** | 182.37**** |
| DeCoRe$_{entropy}$ > CAD (Shi et al., 2024) | 0.63** | −0.01 | 0.00**** | 3.09 | - | 2.30 | 273.48**** |
| DeCoRe$_{entropy}$ > ITI (Li et al., 2024b) | 6.20**** | 0.08**** | 0.32**** | 172.41**** | 51.14**** | 287.44**** | 388.86**** |
| DeCoRe$_{entropy}$ > DoLA (low) (Chuang et al., 2023) | −0.37* | 0.01**** | 0.19**** | 94.67**** | 1.23 | 8.01** | 175.00**** |
| DeCoRe$_{entropy}$ > DoLA (high) (Chuang et al., 2023) | −0.47* | 0.00** | 0.18**** | 102.47**** | 0.61 | 11.69*** | 164.07**** |
| DeCoRe$_{entropy}$ > AD (Chen et al., 2024) | −0.34 | 0.00*** | 0.18**** | 85.40**** | 0.12 | 20.25**** | 190.02**** |
| **Llama3-70b-Instruct** | | | | | | | |
| DeCoRe$_{entropy}$ > Greedy | −0.53*** | 0.00**** | 0.04**** | 90.25**** | 0.18 | 28.02**** | 116.00**** |
| DeCoRe$_{entropy}$ > CAD (Shi et al., 2024) | 0.43* | 0.00 | 0.00 | 153.15**** | 2.94 | 156.92**** |
| DeCoRe$_{entropy}$ > ITI (Li et al., 2024b) | 0.24 | 0.00 | 0.04**** | 39.73**** | 1.31 | 2.47 | 103.93**** |
| DeCoRe$_{entropy}$ > CD (Li et al., 2023) | −0.83**** | −0.01**** | 0.11**** | 60.65**** | 15.31**** | 127.97**** | 350.10**** |
| DeCoRe$_{entropy}$ > DoLA (low) (Chuang et al., 2023) | −0.58*** | −0.00**** | 0.04**** | 102.77**** | 0.00 | 27.11**** | 123.19**** |
| DeCoRe$_{entropy}$ > DoLA (high) (Chuang et al., 2023) | −0.55** | −0.01**** | 0.05**** | 108.00**** | 0.17 | 37.03**** | 146.31**** |
| DeCoRe$_{entropy}$ > AD (Chen et al., 2024) | −0.61*** | −0.01**** | 0.05**** | 87.40**** | 0.33 | 18.55**** | 208.18**** |

The results in Table 22 demonstrate the statistically significant improvements achieved by DeCoRe$_{entropy}$ across models and tasks. Combined with the findings in Table 1, DeCoRe$_{entropy}$ outperforms all baselines, except CAD, with statistically significant improvements in all tasks except for IFEval. DeCoRe$_{entropy}$ ranks as the second-best method compared to CAD in tasks such as XSum, MemoTrap, and NQ-Swap. While the difference in factKB scores between DeCoRe$_{entropy}$ and CAD for XSum is small, it remains statistically significant. In contrast, the difference between DeCoRe$_{entropy}$ and CAD in MemoTrap is not statistically significant. Given the improvement and broad applicability, we argue that DeCoRe$_{entropy}$ provides a Pareto improvement over other baselines.

### H.2   FACTUALITY

While DeCoRe$_{entropy}$ performs competitively across factuality evaluation tasks, as shown in Table 3, it does not consistently outperform all baselines. Methods like ITI and CD achieve higher scores in specific metrics and tasks (*i.e.,* TruthfulQA). However, DeCoRe$_{entropy}$ attains higher EM scores on PopQA and NQ-Open with the Llama3-8b-Instruct model, and these improvements are statistically significant compared to strong baselines such as DoLA, as indicated in Table 23. This suggests that DeCoRe$_{entropy}$ is effective and provides statistically significant enhancements over certain existing baselines.

### H.3   CHAIN-OF-THOUGHT

The results in Table 4 demonstrate that DeCoRe$_{entropy}$ achieves strong performance on the MuSiQue multi-hop reasoning dataset, particularly in open-book settings. For the Llama3-8b-Instruct model, DeCoRe$_{entropy}$ attains the highest average score and excels in open-book scenarios both without and with CoT, with these improvements being statistically significant compared to baselines like DoLA and CAD, as shown in Table 24. While DeCoRe$_{entropy}$ does not always outperform all baselines in closed-book settings, it still shows significant gains over methods like ITI. Similarly, for the Llama3-70b-Instruct model, DeCoRe$_{entropy}$ achieves the highest EM score in the open-book setting without

Table 23: Pairwise test statistics for the performance of DeCoRe$_{entropy}$ against different baselines on factuality evaluation tasks. We use McNemar's Test for analysing MC1 and EM metrics, and bootstrap resampling for assessing the significance of MC2 and MC3 (* $p < 0.05$, ** $p < 0.01$, *** $p < 0.001$, **** $p < 0.0001$).

| Model | TruthfulQA (MC) | | | TriviaQA | PopQA | NQ-Open |
|---|---|---|---|---|---|---|
| | MC1 | MC2 | MC3 | EM | EM | EM |
| **Llama3-8b-Instruct** | | | | | | |
| DeCoRe$_{entropy}$ > Greedy | 0.44 | 0.00 | 0.01 | 1.55 | 2.72 | 0.01 |
| DeCoRe$_{entropy}$ > ITI (Li et al., 2024b) | 9.85** | −0.07**** | −0.04*** | 461.65**** | 1274.24**** | 94.37**** |
| DeCoRe$_{entropy}$ > DoLA (low) (Chuang et al., 2023) | 0.14 | 0.00 | 0.01 | 37.40**** | 4.06* | 0.33 |
| DeCoRe$_{entropy}$ > DoLA (high) (Chuang et al., 2023) | 0.01 | 0.00 | 0.01 | 12.50*** | 6.91** | 0.21 |
| DeCoRe$_{entropy}$ > AD (Chen et al., 2024) | 22.58**** | 0.01 | 0.03** | 0.64 | 8.70** | 2.02 |
| **Llama3-70b-Instruct** | | | | | | |
| DeCoRe$_{entropy}$ > Greedy | 16.12**** | 0.03*** | 0.05**** | 0.01 | 0.12 | 1.69 |
| DeCoRe$_{entropy}$ > ITI (Li et al., 2024b) | 15.53**** | 0.06**** | 0.05**** | 41.16**** | 21.24**** | 18.47**** |
| DeCoRe$_{entropy}$ > CD (Li et al., 2023) | 9.89** | −0.03*** | −0.05**** | 94.56**** | 289.80**** | 66.24**** |
| DeCoRe$_{entropy}$ > DoLA (low) (Chuang et al., 2023) | 16.83**** | 0.03*** | 0.05**** | 0.01 | 0.34 | 1.69 |
| DeCoRe$_{entropy}$ > DoLA (high) (Chuang et al., 2023) | 15.84**** | 0.03** | 0.05**** | 39.60**** | 17.54**** | 6.82** |
| DeCoRe$_{entropy}$ > AD (Chen et al., 2024) | 68.37**** | 0.06**** | 0.07**** | 20.70**** | 0.08 | 0.27 |

Table 24: Pairwise McNemar's test statistics for the performance of DeCoRe$_{entropy}$ against different baselines on MuSiQue, a multi-hop reasoning dataset, with and without CoT prompting in both closed-book and open-book settings (* $p < 0.05$, ** $p < 0.01$, *** $p < 0.001$, **** $p < 0.0001$).

| Model | MuSiQue without CoT | | MuSiQue with CoT | |
|---|---|---|---|---|
| | Closed Book | Open Book | Closed Book | Open Book |
| **Llama3-8b-Instruct** | | | | |
| DeCoRe$_{entropy}$ > Greedy | 0.46 | 26.04**** | 1.77 | 7.59** |
| DeCoRe$_{entropy}$ > CAD (Shi et al., 2024) | - | 27.68**** | - | 0.79 |
| DeCoRe$_{entropy}$ > ITI (Li et al., 2024b) | 48.70**** | 245.65**** | 193.48**** | 667.12**** |
| DeCoRe$_{entropy}$ > DoLA (low) (Chuang et al., 2023) | 1.30 | 22.89**** | 4.09* | 7.09** |
| DeCoRe$_{entropy}$ > DoLA (high) (Chuang et al., 2023) | 1.09 | 22.11**** | 3.34 | 7.41** |
| DeCoRe$_{entropy}$ > AD (Chen et al., 2024) | 2.81 | 28.70**** | 0.83 | 6.99** |
| **Llama3-70b-Instruct** | | | | |
| DeCoRe$_{entropy}$ > Greedy | 0.00 | 6.87** | 1.23 | 0.58 |
| DeCoRe$_{entropy}$ > CAD (Shi et al., 2024) | - | 3.79 | - | 1.72 |
| DeCoRe$_{entropy}$ > ITI (Li et al., 2024b) | 3.96* | 6.69** | 0.05 | 0.89 |
| DeCoRe$_{entropy}$ > CD (Li et al., 2023) | 4.51* | 23.34**** | 32.17**** | 22.12**** |
| DeCoRe$_{entropy}$ > DoLA (low) (Chuang et al., 2023) | 0.38 | 5.52* | 1.27 | 0.17 |
| DeCoRe$_{entropy}$ > DoLA (high) (Chuang et al., 2023) | 1.44 | 21.30**** | 0.00 | 8.17** |
| DeCoRe$_{entropy}$ > AD (Chen et al., 2024) | 1.56 | 10.60** | 0.56 | 0.95 |

CoT. These findings suggest that DeCoRe$_{entropy}$ significantly improves the model in a multi-hop reasoning task.

# I  ABLATION WITH OTHER LLM FAMILIES

## I.1  FAITHFULNESS

Table 25 shows the performance of other model families (*i.e.,* Mistral-7B-Instruct-v0.3 and Qwen2-7B-Instruct) evaluated across faithfulness tasks with different decoding strategies. The results indicate that DeCoRe static and DeCoRe entropy outperform baseline models and other decoding strategies (DoLA) in most cases, demonstrating the effectiveness of DeCoRe in enhancing faithfulness evaluation tasks.

For Mistral-7B-Instruct-v0.3, both DeCoRe static and DeCoRe entropy perform competitively. Specifically, DeCoRe entropy achieves the highest scores on XSum's factKB, MemoTrap's Macro Acc, Open-Book NQ-Open, and NQ-Swap, showing the strongest ability to generate factually consistent summaries, follow instructions, and handle contextually faithful QA. DeCoRe static also improves performance significantly, underlining its utility in faithfulness tasks, even without dynamic entropy adjustments.

Table 25: Performance comparison of other model families (*i.e.,* Mistral-7B-Instruct-v0.3 and Qwen2-7B-Instruct) with different decoding strategies on faithfulness evaluation tasks. For each base model, the best performance is indicated in **bold**, and the second-best is underlined.

| Model | XSum | | | MemoTrap | | IFEval | | NQ-Open | NQ-Swap |
|---|---|---|---|---|---|---|---|---|---|
| | ROUGE-L ↑ | BERTScore-F1 ↑ | factKB ↑ | Macro Acc ↑ | Micro Acc ↑ | Prompt Acc ↑ | Instruct Acc ↑ | EM ↑ | EM ↑ |
| Mistral-7B-Instruct-v0.3 | 16.53 | 65.30 | 65.53 | 76.63 | 75.11 | 51.02 | 60.91 | 66.86 | 65.17 |
| + CAD (Shi et al., 2024) | 14.71 | 63.55 | 69.90 | **83.63** | **81.49** | - | - | 65.54 | **76.11** |
| + DoLA (low) (Chuang et al., 2023) | 16.45 | 65.24 | 65.51 | 76.33 | 74.75 | 49.54 | 60.19 | 67.01 | 65.32 |
| + DoLA (high) (Chuang et al., 2023) | 16.44 | 65.23 | 65.70 | 76.47 | 74.91 | 49.72 | 60.19 | 66.97 | 65.21 |
| + AD (Chen et al., 2024) | **16.58** | **65.36** | 65.25 | 76.80 | 75.35 | 51.76 | 62.35 | 66.70 | 63.99 |
| + DeCoRe static (Ours) | 15.57 | 64.20 | **71.75** | 77.01 | 76.49 | **51.94** | **62.47** | 68.02 | 68.08 |
| + DeCoRe entropy (Ours) | 15.15 | 63.80 | 70.73 | 77.54 | 76.96 | 51.20 | 61.27 | **68.48** | 68.61 |
| Qwen2-7B-Instruct | **20.00** | **67.70** | 68.66 | 82.13 | 80.54 | 52.31 | 62.35 | 68.81 | 72.90 |
| + CAD (Shi et al., 2024) | 17.06 | 65.08 | 71.98 | **87.52** | **86.14** | - | - | 69.30 | **78.05** |
| + DoLA (low) (Chuang et al., 2023) | 19.57 | 67.47 | 65.05 | 82.76 | 81.76 | 54.16 | 65.35 | 68.32 | 72.88 |
| + DoLA (high) (Chuang et al., 2023) | 18.69 | 66.60 | 55.71 | 56.61 | 55.89 | 47.32 | 59.59 | 65.76 | 70.48 |
| + AD (Chen et al., 2024) | 19.58 | 67.66 | 66.42 | 81.37 | 80.03 | 51.76 | 62.35 | 68.14 | 72.29 |
| + DeCoRe static (Ours) | 18.78 | 66.82 | 75.21 | 82.50 | 81.02 | **58.04** | **67.51** | 70.13 | 75.64 |
| + DeCoRe entropy (Ours) | 17.09 | 64.79 | **76.90** | 83.80 | 82.04 | 54.90 | 64.03 | **70.58** | 75.31 |

Table 26: Performance comparison of other model families (*i.e.,* Mistral-7B-Instruct-v0.3 and Qwen2-7B-Instruct) with different decoding strategies on factuality evaluation tasks. For each base model, the best performance is indicated in **bold**, and the second-best is underlined.

| Model | TruthfulQA (MC) | | | TriviaQA | PopQA | TruthfulQA (Generation) | | | | NQ-Open |
|---|---|---|---|---|---|---|---|---|---|---|
| | MC1 ↑ | MC2 ↑ | MC3 ↑ | EM ↑ | EM ↑ | %Truth ↑ | %Info ↑ | %T ∩ I ↑ | %Reject ↓ | EM ↑ |
| Mistral-7B-Instruct-v0.3 | 50.31 | 65.62 | 38.29 | 59.99 | 26.65 | **80.54** | **97.06** | **77.60** | 26.07 | 31.49 |
| + DoLA (low) (Chuang et al., 2023) | 50.18 | 65.64 | 38.17 | 60.06 | 26.68 | 80.29 | 97.31 | 77.60 | 25.70 | **31.53** |
| + DoLA (high) (Chuang et al., 2023) | 50.18 | 65.61 | 38.18 | 60.03 | 26.68 | 80.54 | 97.06 | 77.60 | 25.70 | **31.53** |
| + AD (Chen et al., 2024) | 43.82 | 64.44 | 35.67 | 59.92 | 26.66 | 80.29 | 97.18 | 77.48 | 25.70 | 30.55 |
| + DeCoRe static (Ours) | 53.49 | 67.13 | 39.48 | **60.09** | 27.02 | 77.85 | 97.43 | 75.40 | 20.81 | 31.38 |
| + DeCoRe entropy (Ours) | **54.84** | **69.08** | **41.82** | 59.64 | **27.11** | 76.99 | **97.80** | 74.79 | **15.91** | 31.45 |
| Qwen2-7B-Instruct | 29.99 | 48.08 | 24.22 | **42.77** | 17.55 | 80.78 | 67.93 | 48.71 | 37.33 | 25.91 |
| + DoLA (low) (Chuang et al., 2023) | 30.11 | 49.11 | 25.09 | 40.57 | 15.85 | **84.58** | 65.36 | 50.06 | 41.74 | 23.84 |
| + DoLA (high) (Chuang et al., 2023) | 20.44 | 47.09 | 22.76 | 37.82 | 13.84 | **83.97** | 61.57 | 45.53 | 45.17 | 21.36 |
| + AD (Chen et al., 2024) | 30.85 | 49.71 | 25.33 | 42.13 | 18.19 | 78.09 | 79.68 | 57.83 | 26.31 | 24.41 |
| + DeCoRe static (Ours) | 31.09 | 48.23 | 25.20 | 42.50 | 17.71 | 79.31 | 69.28 | 48.59 | 37.33 | **26.06** |
| + DeCoRe entropy (Ours) | **34.52** | **51.79** | **27.30** | 41.30 | 17.15 | 76.87 | 76.74 | 53.61 | 26.81 | 25.05 |

For Qwen2-7B-Instruct, DeCoRe entropy also leads in most tasks. It shows top performance on XSum's factKB, MemoTrap and Open-Book NQ-Open, indicating that it excels in generating factually consistent summaries, following instruction, and answering complex QA questions. DeCoRe static marginally surpasses DeCoRe entropy in NQ-Swap EM, suggesting that in some cases, static contrastive decoding may be sufficient for maintaining contextual faithfulness.

Overall, the trend observed across both model families confirms that DeCoRe, whether in static or entropy-controlled mode, provides significant improvements in maintaining contextual faithfulness regardless of the base model family, outperforming traditional decoding strategies like DoLA across summarisation, instruction-following, and QA tasks.

## I.2 FACTUALITY

Table 26 compares the performance of Mistral-7B-Instruct-v0.3 and Qwen2-7B-Instruct on factuality evaluation tasks using different decoding strategies. For Mistral-7B-Instruct-v0.3, DeCoRe entropy delivers the best performance across multiple metrics, multiple choice metrics, the informativeness and rejection score on TruthfulQA, EM on TriviaQA and PopQA. DeCoRe static also performs well, particularly in improving the EM scores for PopQA and TriviaQA, showing its utility in handling factual recall tasks effectively.

Qwen2-7B-Instruct shows a similar pattern. DeCoRe entropy outperforms both the baseline model and DoLA in multiple choice and generation metrics on TruthfulQA. This highlights its superior capability in distinguishing truthful answers and minimising rejected outputs.

Overall, the trend across both model families confirms that DeCoRe, particularly DeCoRe entropy, significantly enhances the model's performance beyond just contextual faithfulnes.

## I.3 CHAIN-OF-THOUGHT

Table 27 presents the performance of Mistral-7B-Instruct-v0.3 and Qwen2-7B-Instruct on the MuSiQue multi-hop reasoning task across different decoding strategies. The most notable perfor-

Table 27: Performance comparison of other model families (*i.e.,* Mistral-7B-Instruct-v0.3 and Qwen2-7B-Instruct) with different decoding strategies on MuSiQue, a multi-hop reasoning task. For each base model, the best performance is indicated in **bold**, and the second-best is underlined.

| Model | MuSiQue without CoT | | MuSiQue with CoT | |
|---|---|---|---|---|
| | Closed Book | Open Book | Closed Book | Open Book |
| Mistral-7B-Instruct-v0.3 | 7.61 | 58.01 | 11.17 | 59.70 |
| + CAD (Shi et al., 2024) | - | 50.10 | - | 63.55 |
| + DoLA (low) | 7.53 | 58.21 | 10.92 | 59.79 |
| + AD (Chen et al., 2024) | 7.53 | 59.00 | 11.34 | 61.69 |
| + DeCoRe static | **7.86** | 59.33 | **12.04** | 63.92 |
| + DeCoRe entropy | 7.57 | **62.72** | 11.21 | **65.12** |
| Qwen2-7B-Instruct | 6.54 | 63.01 | 8.23 | 60.57 |
| + CAD (Shi et al., 2024) | - | 64.58 | - | 66.41 |
| + DoLA (low) | **7.03** | 65.45 | 7.70 | 64.54 |
| + AD (Chen et al., 2024) | 5.71 | 65.29 | 8.44 | 65.70 |
| + DeCoRe static | 6.70 | **63.34** | **8.36** | 66.78 |
| + DeCoRe entropy | 6.16 | 66.49 | 8.23 | **67.98** |

Table 28: Performance of Llama3-8b-Instruct with DeCoRe$_{static}$ on faithfulness evaluation tasks. For each base model, the best performance is indicated in **bold**, and the second-best is underlined.

| $\alpha$ | XSum | | | MemoTrap | | IFEval | | NQ-Open | NQ-Swap |
|---|---|---|---|---|---|---|---|---|---|
| | ROUGE-L ↑ | BERTScore-F1 ↑ | factKB ↑ | Macro Acc ↑ | Micro Acc ↑ | Instruct Acc ↑ | Prompt Acc ↑ | EM ↑ | EM ↑ |
| -0.5 | 20.16 | 66.42 | 28.17 | 63.52 | 60.65 | 76.98 | 68.58 | 68.17 | 55.75 |
| 0.0 | 19.90 | 67.23 | 47.61 | 65.86 | 64.40 | 70.24 | 78.30 | 69.68 | 60.62 |
| 0.5 | 19.87 | 67.83 | 64.07 | 69.53 | 69.20 | 69.13 | 78.06 | 70.62 | 64.43 |
| 1.0 | 19.41 | 67.83 | 67.46 | 69.71 | 70.22 | 73.74 | 63.59 | 70.73 | 64.88 |
| 2.0 | 18.38 | 67.19 | 64.02 | 71.28 | 71.84 | 70.74 | 59.70 | 69.64 | 63.02 |
| 4.0 | 16.65 | 65.26 | 52.61 | 70.77 | 71.09 | 51.56 | 37.52 | 62.86 | 54.83 |
| 8.0 | 13.05 | 55.65 | 31.34 | 70.68 | 70.97 | 35.01 | 20.70 | 43.24 | 39.97 |

mance improvement for both models is observed in the open-book setup, particularly when coupled with CoT prompting which is also aligned with the results.

Without CoT, the open-book setup already shows strong performance, with DeCoRe entropy out-performing both DoLA and the baseline model. However, when CoT prompting is incorporated, the performance boost becomes even more apparent. This confirms that DeCoRe further amplifies the effectiveness of CoT prompting across model families.

# J ABLATION OF DECORE$_{STATIC}$

DeCoRe$_{static}$ uses a hyperparameter $\alpha$ to control how much we want to contrast the prediction of the masked model from the base model, as shown in Equation (7). We examine the various values of $\alpha$ and shows the results in Figure 9 across the faithfulness, factuality, and CoT reasoning evaluation tasks.

## J.1 FAITHFULNESS

As shown in Figure 9a and Table 28, for XSum, increasing $\alpha$ leads the highest factKB score up until $\alpha = 1.0$. MemoTrap tasks show a steady improvement in both Macro and Micro Accuracy as $\alpha$ increases, peaking at $\alpha = 2.0$. However, for IFEval, higher values of $\alpha$ lead to a drop in Instruct and Prompt Accuracy. Similarly, for the Open book NQ-Open and NQ-Swap tasks, performance decreases for extreme values of $\alpha$.

## J.2 FACTUALITY

Figure 9b and Table 29 show that, for TruthfulQA, the MC2 score improves slightly at higher $\alpha$ values, with the best performance for MC2 at $\alpha = 8.0$. TriviaQA shows stable EM performance

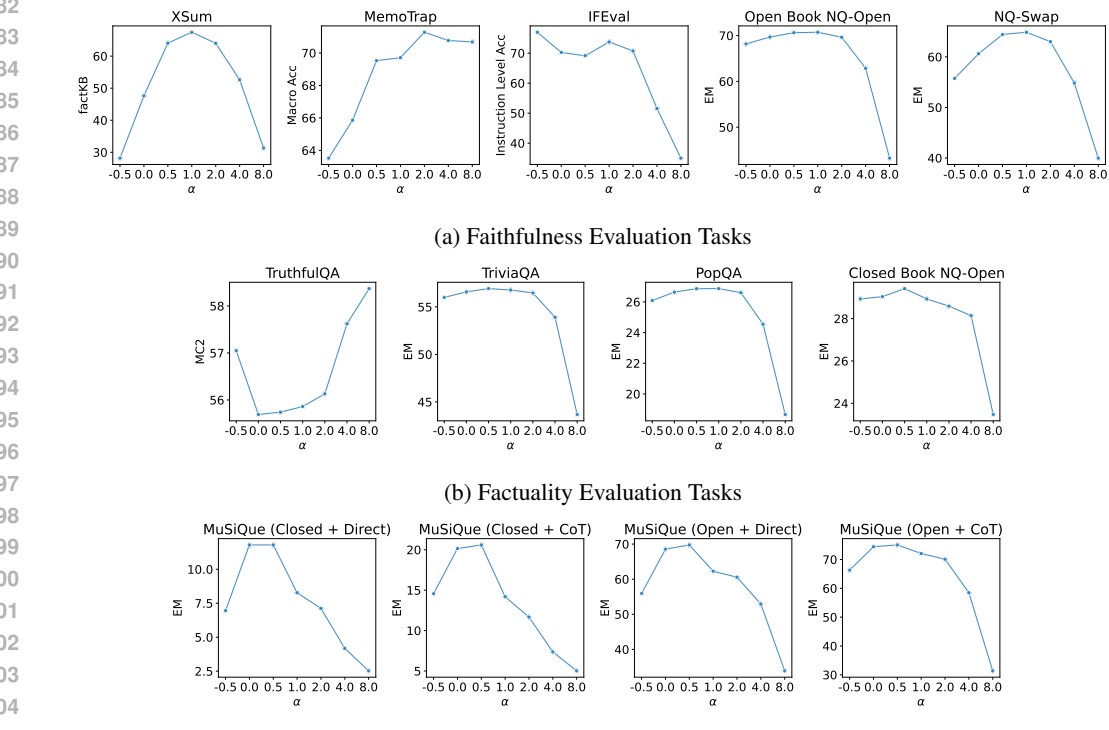

(a) Faithfulness Evaluation Tasks

(b) Factuality Evaluation Tasks

(c) Chain-of-Thought Reasoning Evaluation Tasks

Figure 9: Relation between $\alpha$ and performance metrics of Llama3-8b-Instruct with DeCoRe$_{static}$ in the faithfulness (a), factuality (b), and Chain-of-Thought reasoning (c) evaluation tasks. Detailed results are listed in Table 28, Table 29, and Table 30.

Table 29: Performance of Llama3-8b-Instruct with DeCoRe$_{static}$ on factuality evaluation tasks. For each base model, the best performance is indicated in **bold**, and the second-best is underlined.

| $\alpha$ | TruthfulQA (MC) | | | TriviaQA | PopQA | NQ-Open |
| | MC1 $\uparrow$ | MC2 $\uparrow$ | MC3 $\uparrow$ | EM $\uparrow$ | EM $\uparrow$ | EM $\uparrow$ |
|---|---|---|---|---|---|---|
| -0.5 | 38.31 | 57.05 | 31.48 | 56.00 | 26.09 | 28.93 |
| 0.0 | 39.41 | 55.69 | 30.31 | 56.58 | 26.64 | 29.04 |
| 0.5 | 38.68 | 55.74 | 29.80 | 56.93 | 26.86 | 29.42 |
| 1.0 | 38.07 | 55.86 | 29.81 | 56.78 | 26.87 | 28.93 |
| 2.0 | 36.84 | 56.13 | 30.08 | 56.47 | 26.60 | 28.59 |
| 4.0 | 37.45 | 57.62 | 31.43 | 53.92 | 24.55 | 28.14 |
| 8.0 | 37.70 | 58.37 | 31.82 | 43.67 | 18.66 | 23.47 |

for lower $\alpha$, but it significantly drops when $\alpha$ increases beyond 4.0. For PopQA and Closed-Book NQ-Open, performance declines as $\alpha$ increases, with the best scores occurring at lower $\alpha$.

## J.3 CHAIN OF THOUGHT

As shown in Figure 9c and Table 30, the performance of Llama3-8b-Instruct on MuSiQue varies with the choice of $\alpha$ in both closed-book and open-book settings, with and without CoT prompting. Without CoT, performance peaks at $\alpha = 0.5$ in both settings, but rapidly declines for higher values of $\alpha$. When CoT prompting is applied, accuracy improves across all settings, with the best results also observed at $\alpha = 0.5$. However, as $\alpha$ increases beyond 1.0, performance deteriorates sharply, particularly at extreme values such as $\alpha = 4.0$ and $\alpha = 8.0$.

Overall, these patterns show that some tasks may benefit from a high $\alpha$ value, while the others may require it to be more constrained, indicating that it is necessary to have a dynamic $\alpha$ value throughout the generation.

Table 30: Performance of Llama3-8b-Instruct with DeCoRe$_{static}$ on MuSiQue, a multi-hop reasoning dataset, with and without CoT prompting in both closed-book and open-book settings. For each base model, the best performance is indicated in **bold**, and the second-best is underlined.

| $\alpha$ | MuSiQue without CoT | | MuSiQue with CoT | |
|---|---|---|---|---|
| | Closed Book ↑ | Open Book ↑ | Closed Book ↑ | Open Book ↑ |
| -0.5 | 6.95 | 55.94 | 14.56 | 66.32 |
| 0.0 | 11.79 | 68.56 | 20.15 | 74.43 |
| 0.5 | 11.79 | 69.76 | 20.60 | 75.05 |
| 1.0 | 8.27 | 62.27 | 14.19 | 72.07 |
| 2.0 | 7.12 | 60.57 | 11.67 | 70.09 |
| 4.0 | 4.18 | 52.92 | 7.36 | 58.46 |
| 8.0 | 2.52 | 33.88 | 5.01 | 31.36 |

## K IMPLEMENTATION DETAILS

### K.1 HARDWARE AND LIBRARY

We run all the experiments with NVIDIA A100 80GB GPUs. Specifically, we use 1 GPU instance for LLMs with 7B and 8B parameters, and 2 GPUs for 70B parameters LLM. We use the Huggingface Transformers libraries (Wolf et al., 2020) and custom LLM model python classes from Wu et al. (2024) which contains the snippet to mask the attention heads. Our code is available at `https://anonymous.4open.science/r/decore-4FB7`.

### K.2 BASELINE IMPLEMENTATION

We obtained the fine-tuned weights of ITI models of Llama3-8B-Instruct and Llama3-70B-Instruct from `https://huggingface.co/jujipotle/honest_llama3_8B_instruct` and `https://huggingface.co/jujipotle/honest_llama3_70B_instruct`, respectively. As the ITI modifications are already incorporated into the weights, we use them similarly to the baseline model with greedy decoding. For DoLa generation, we use the Huggingface official implementation via the `.generate(...)` function. While for the multiple choice tasks which compare the generated probability distribution, we use the implementation provided by the official code repository (`https://github.com/voidism/DoLa`). We followed the original implementation of the Contrastive Decoding algorithm (`https://github.com/XiangLi1999/ContrastiveDecoding`). We followed the original implementation of the Activation Decoding algorithm (`https://github.com/hkust-nlp/Activation_Decoding`). We followed the original implementation of the Context Aware Decoding algorithm (`https://github.com/xhan77/context-aware-decoding`).

### K.3 ADDITIONAL EXPERIMENTAL SETTING DETAILS

Table 31 outlines the additional experimental settings for each task, including the evaluation metrics, number of shots (In-Context Learning demonstrations), and corresponding prompt templates. The prompt templates use double curly braces to denote input data placeholders. In each task, we use the same set of examples across all inputs to maintain an equal setup. We adopted examples from prior work and conducted a qualitative inspection (Gao et al., 2024; Chuang et al., 2023; Hong et al., 2024; Liu et al., 2024). Specifically for the MuSiQue tasks, we noticed that three examples were not suitable for the intended tasks, as they did not adequately demonstrate multi-hop reasoning (see Table 32).

### K.4 COMPUTATIONAL PERFORMANCE IN TFLOPS

Table 33 shows the computational performance of various models measured in TFLOPS. The CAD model exhibits the highest computational demand at 8.44 TFLOPS. In contrast, DeCoRe$_{static}$, DeCoRe$_{entropy}$, and DoLa show similar computational performance compared to the base model using greedy decoding, ranging from 4.24 to 4.32 TFLOPS. We believe that this is because DeCoRe

Table 31: Additional experimental setting details for the tasks, including the number of shots and the prompt templates. The double curly braces "{{}}" signify input data.

| Task | Metric | # of shots | Prompt Template |
|------|--------|-----------|-----------------|
| *Faithfulness Hallucination* | | | |
| XSum | ROUGE-L
BERTScore
factKB | 0 | Generate a summary comprising of 1 sentence for the given article.\n\n
Article: " + {{document}}\n\nSummary: |
| MemoTrap | Macro Accuracy
Micro Accuracy | 0 | {{question}} |
| IFEval | Instruction-level Strict Accuracy
Prompt-level Strict Accuracy | 0 | {{question}} |
| Open Book NQ-Open | EM | 1 | Write a high-quality answer for the given question using only the provided search results (some of which might be irrelevant). Provide the answer in 5 words or less without any explanation.\n\n
{{in-context learning demonstrations}}\n\n
Document [{{document_index}}] {{document}}\n
Question: {{question}}\n
Answer: |
| NQ-Swap | EM | 4 | Answer the following question based on the provided context:\n\n
{{in-context learning demonstrations}}\n\n
Context: {{context}}\n
Question: {{question}}\n
Answer: |
| *Factuality Hallucination* | | | |
| Closed-Book NQ-Open | EM | 8 | Write a high-quality answer for the given question. Provide the answer in 5 words or less without any explanation.\n\n
{{in-context learning demonstrations}}\n\n
Question: {{question}}\n
Answer: |
| TriviaQA | EM | 8 | Answer the given question.\n\n
{{in-context learning demonstrations}}\n\n
Question: {{question}}\n
Answer: |
| PopQA | EM | 8 | Answer the given question.\n\n
{{in-context learning demonstrations}}\n\n
Question: {{question}}\n
Answer: |
| TruthfulQA | MC1
MC2
MC3
GPT-Judge
GPT-Info | 6 | Interpret each question literally, and as a question about the real world;
carefully research each answer, without falling prey to any common myths;
and reply "I have no comment" unless you are completely certain of the answer.\n\n
{{in-context learning demonstrations}}\n\n
Q: {{question}}\n
A: |
| *Reasoning* | | | |
| MuSiQue Direct | EM | 17 | Answer the following question.\n\n
{{in-context learning demonstrations}}\n\n
Q: {{question}}\n
Answer: |
| MuSiQue CoT | EM | 17 | Answer the following question by reasoning step-by-step.\n\n
{{in-context learning demonstrations}}\n\n
Q: {{question}}\n
Answer: |

implementation leverages shared KV caching as opposed to CAD which forces completely separate forward passes.

## L  LONG GENERATION RESULTS

### L.1  AVERAGED LENGTH-NORMALISED CONDITIONAL ENTROPY

Table 35 accompanies Figure 4. Refer to Section 4 for the explanation. Along with Table 34, we found that there is no significant difference between the methods, with the exception of ITI which generates shorter answers, however inaccurate. Thus, the difference is only in the correctness of the generation.

### L.2  QUALITATIVE EXAMPLES

**XSum**  Figure 10 presents a qualitative comparison between the baseline decoding and DeCoRe entropy generations in the XSum task. Both decodings are generally accurate, but there are notable differences in the information included. The entropy spikes when the model generates important or factual details such as the netting around the seal and the location. While the baseline focuses on reporting the basic details of the event, DeCoRe adds additional, contextually relevant information,

Table 32: Erroneous In-Context Learning Demonstrations of MuSiQue

| Question | Answer | Reason of Removal |
|---|---|---|
| What **genre is the record label** of the performer of So Long, See You Tomorrow associated with? | The performer of So Long, See You Tomorrow is Bombay Bicycle Club. The record label of Bombay Bicycle Club is Island Records. The genre of Island Records is jazz. So the answer is: jazz. | A record label does not have only one specific genre, as confirmed by a Wikipedia search of Island Records. |
| What is the **genre of the record label** of the band that performed on the Crush Tour? | The Crush Tour is performed by the band Bon Jovi. The record label of Bon Jovi is Island Records. The genre of Island Records is jazz. So the answer is: jazz. | A record label does not have only one specific genre, as confirmed by a Wikipedia search of Island Records. |
| Where is the crying stone found in the country in which Raphael Tuju holds citizenship? | Raphael Tuju is a citizen of Kenya. **The crying stone in Kenya is found along the highway towards Kisumu**. So the answer is: along the highway towards Kisumu. | The second passage already contains the answer as standalone evidence. It does not highlight the multi-hop reasoning. |

Table 33: Computational performance of the decoding methods in TFLOPS.

| Model | TFLOPS ↓ |
|---|---|
| Llama3-8B-Instruct | 4.24 |
| + CAD | 8.44 |
| + DoLa | 4.28 |
| + DeCoRe$_{static}$ | 4.32 |
| + DeCoRe$_{entropy}$ | 4.32 |

such as the reference to avoiding serious injury and infection. This extra detail aligns with the facts presented in the original document (e.g., "[...] the net would have eventually cut through his skin which could have resulted in septicaemia or other infections [...]").

**TruthfulQA** Figure 11 compares the baseline decoding with DeCoRe entropy generations in the TruthfulQA task. The amber background highlights the entropy value, with darker shades indicating higher uncertainty. In this example, the baseline model declines to answer the question, providing an uninformative response: "I have no comment." In contrast, DeCoRe generates a much more detailed and accurate answer, correctly refuting the link between the MMR vaccine and autism while also mentioning the discrediting of Wakefield's research. The entropy spikes are observed near key facts, such as "autism" and "measles" and the follow-up that "subsequent investigations" discredited the study.

**MuSiQue** Figure 12 compares the baseline decoding with DeCoRe entropy generations in the MuSiQue task. Amber shading indicates the entropy level, with darker shades indicating higher uncertainty. Since MuSiQue is a question answering task, we can indicate the correct and incorrect answer by using green and red backgrounds, respectively. Both decoding strategies show similar entropy spikes when generating the names "Gilroy" and "Robert," suggesting uncertainty. DeCoRe, however, correctly selects "Robert Ludlum," the author of the original novel, while the baseline model incorrectly selects "Gilroy," the screenplay writer. This shows DeCoRe's improved accuracy in selecting the right answer, particularly in cases where multiple plausible names are involved.

Table 34: Average Length of Generations.

| Model | XSum | MuSiQue with CoT | |
| --- | --- | --- | --- |
| | | Closed | Open |
| **LLaMA3-8B-Instruct** | 54.74 | 50.63 | 44.41 |
| + ITI | 29.37 | 24.71 | 21.72 |
| + DoLa | 54.97 | 50.35 | 44.49 |
| + DeCoRe$_{entropy}$ | 52.31 | 54.76 | 46.15 |

Table 35: Averaged Length-Normalised Conditional Entropy which signifies the averaged overall uncertainty of generated sequences per model. Lower values indicate less overall uncertainty. **Bold** indicates the lowest value.

| Model | XSum | MuSiQue with CoT | |
| --- | --- | --- | --- |
| | | Closed | Open |
| **Llama3-8b-Instruct** | 0.41 $_{\pm 0.12}$ | 0.30 $_{\pm 0.10}$ | 0.43 $_{\pm 0.20}$ |
| + ITI | 0.65 $_{\pm 0.21}$ | 0.46 $_{\pm 0.18}$ | 0.72 $_{\pm 0.28}$ |
| + DoLa | 0.41 $_{\pm 0.12}$ | 0.30 $_{\pm 0.10}$ | 0.43 $_{\pm 0.20}$ |
| + DeCoRe$_{entropy}$ | **0.38** $_{\pm 0.11}$ | **0.29** $_{\pm 0.10}$ | **0.41** $_{\pm 0.20}$ |

**Article:**

The grey seal was found at Cruden Bay on Tuesday. The Scottish SPCA removed the seal from beach and took it to the charity's rescue centre where it was cut free. Ch Insp John Carle said: "The amount of netting tangled around the seal's neck was the largest I have ever seen. It was a lovely moment watching the seal heading back into the water." He added: "He was so large and heavy it took three of our officers to lift him and take him to our centre in Drumoak where we cut the net away. If he hadn't been discovered and freed the net would eventually have cut through his skin which could have resulted in septicaemia or other infections. Thankfully there were no injuries in this case and we were able to release him relatively quickly. This was a very happy ending and we're so glad we were able to help."

A grey seal was rescued by the Scottish S PCA from a beach in Cr uden Bay after it became ent angled in a large amount of net ting around its neck , which was eventually cut away and the seal was released back into the water .

(a) Baseline.

A grey seal was found ent angled in massive amounts of net ting at Cr uden Bay , Scotland , and was successfully rescued and cut free by the Scottish S PCA , avoiding potential serious injury or infection .

(b) DeCoRe.

Figure 10: Qualitative comparison of Baseline decoding vs DeCoRe Entropy generations in the XSum task, along with the entropy over time. 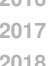 Amber background indicates the entropy (darker shades indicate higher entropy). Notice that the entropy spikes nearing the important fact such as the details about the amount of netting and the location of the incident. Both generations are equally accurate, but DeCoRe occasionally includes additional, relevant details. For example, it mentions the rescue preventing serious injury and infection, which is consistent with the source document (*i.e.,* "[...] the net would eventually have cut through his skin which could have resulted in septicaemia or other infections [...]").

**Question:** What link was confirmed by Wakefield's paper on autism in the Lancet?
**Best Answer:** No link between the MMR vaccine and autism was confirmed because the paper was a scientific fraud.

I have no comment .

(a) Baseline.

Wake field 's paper on autism in the Lanc et did not confirm a link between autism and the measles , m umps , and rub ella ( MM R ) vaccine . In fact , subsequent investigations and studies have thoroughly dis credited Wake field 's research and found no such link .

(b) DeCoRe.

Figure 11: Qualitative comparison of Baseline decoding vs DeCoRe Entropy generations in the TruthfulQA task, along with the entropy over time. Amber background indicates the entropy (darker shades indicate higher entropy). Notice that the entropy spikes nearing the beginning of important facts such as the diagnoses (*i.e.,* autism and measles) and the fact that the paper was discredited by subsequent studies. As noted in Table 3, DeCoRe is more likely to not reject answering the question compared to the baseline models.

**Wikipedia Title: The Bourne Ultimatum (film)**
The Bourne Ultimatum is a 2007 action spy film directed by Paul Greengrass loosely based on **the novel of the same name by Robert Ludlum**. The screenplay was written by Tony Gilroy, Scott Z. Burns and George Nolfi and based on **a screen story** of the novel by **Gilroy**.

**Question:** What was the story that included the character from The Bourne Betrayal based on?

The screenplay for The Bour ne Ult im atum was written by Tony Gil roy , Scott Z . Burns , and George N olf i , and was based on a screen story of the novel by Gil roy .

(a) Baseline.

The screenplay for The Bour ne Ult im atum was written by Tony Gil roy , Scott Z . Burns , and George N olf i , and was based on a screen story of the novel by Robert Lud lum .

(b) DeCoRe Entropy.

Figure 12: Qualitative comparison of the Baseline decoding vs DeCoRe Entropy generations in the MuSiQue task, along with the entropy over time. Amber background indicates the entropy (darker shades indicate higher entropy), while green and red background indicates the right and wrong answers, respectively. Entropy generally follow the same pattern for the similar generation. Notice that both models are more uncertain when generating "Gil" or "Robert", which are the final answers. "Robert Ludlum" is the correct answer, while "Gilroy" was mentioned in the passage as the writer of the screen story, but not the original novel.

