# OpenReview forum: "DeCoRe: Decoding by Contrasting Retrieval Heads to Mitigate Hallucinations"
_ICLR.cc/2025/Conference — Submitted to ICLR 2025_

### Official Review · Reviewer_1dQs · 2024-11-02

**Soundness:** 3
**Presentation:** 3
**Contribution:** 3
**Rating:** 6
**Confidence:** 3

**Summary:**

The paper proposes a new variant of contrastive decoding for reducing hallucinations in language models (LMs). At high level, contrastive decoding uses the difference between two LMs’ likelihood as the final decoding criteria. Typically, there is a baseLLM that generates a basic sequence of tokens which might contain hallucinations. To reduce the amount of hallucination, the probabilities of each generated token is then offset by the corresponding token probabilities from another weaker or corrupted LM which might contain more hallucinations. By using the offset to calibrate the sequence likelihood, the resulting decoding can potentially avoid undesirable failures from the second LM.

In this paper, the authors focus on building the corrupted LM leveraging the so-called retrieval attention heads found by recent work. Specifically, by masking those retrieval attention heads, the resulting corrupted LM is more likely to ignore context information, leading to more faithfulness issues (hallucinations).

Thus, the hypothesis is that using the token probability difference between the base LM and the corrupted LM can lead to better grounded responses.

Experiments are conducted on several representative datasets across different open source LMs (Llama 3 8B & 70B in main text), including long-form (summarization) and short-form (question answering) generation tasks.
Overall, the proposed method is found to be effective in making LMs more sensitive to provided context while not interfering with the LM’s parametric knowledge.

**Strengths:**

The paper is easy-to-follow with necessary background.

Given the wide range of LM applications, developing efficient and reliable hallucination mitigation methods is a very important research direction.

The proposed method is a nice extension of contrastive decoding with recent attention head function insights.

The experiment design are mostly reasonable with several recent baselines to validate the effectiveness of the proposed method

**Weaknesses:**

Although it is good to build upon recent attention head insights for contrastive decoding, the generalizability of those identified retrieval attention seems very limited.

The improvement over baselines is at odds and not pronounced enough. For example, the proposed method does not provide consistent improvements over the base LM and CAD on fluency and coherence (lower ROUGE-L and BERTScore-F1). It is good to analyze the response difference to drive deeper understanding, e.g., does the proposed method lead to shorter and more concise generation? Typically, shorter answers are preferred by question answering datasets, e.g., NQ, TriviaQA.

As reported by the authors in Fig 3, masking more heads is not necessarily helpful. It is good to explore alternative attention selection strategies, e.g., randomly masking, selecting heads based on the required task solving capabilities (profiling using question answering datasets), or layer-based selection (e.g., balanced masking out across different layers).
That said, it might be interesting to see whether certain attention heads are more related to shorter vs longer context retrieval.

There are some further improvements can be done for the experiment part:
Most of the datasets studied are of shorter input context. It is good to explore some datasets with longer context (e.g., retrieving more passages) and compare the method there. In addition to longer context, it is also good to explore more involving contexts, e.g., contextual conflicts [1].

It is good to report the robustness of those chosen hyperparameters. For example, can the proposed method benefit from different sampling temperatures and varying number of heads to be masked. As in-context examples are used for certain tasks, it is good to study the robustness of that decision, e..g, different set of shot and less or more examples.

[1] Wang et al., Resolving Knowledge Conflicts in Large Language Models

**Questions:**

Is there any difference in terms of generations from the proposed method vs others, e.g., fluency, length?

What is the smaller LM used for DeCoRe lite?

What is the alpha value for DeCoRe static? How that is determined?

What is the computational cost for all compared methods, e.g., tflops?

---

> ### Author Response · Authors · 2024-11-15
> **Response to Reviewer 1dQs**
>
> Thank you for the review!
>
> > Although it is good to build upon recent attention head insights for contrastive decoding, the generalizability of those identified retrieval attention seems very limited.
>
> We agree that retrieval head insights are relatively recent. Our work represents an initial exploration in leveraging these insights in hallucination mitigation. Our experiments indicate that the approach of leveraging retrieval heads using contrastive decoding can improve contextual faithfulness across diverse tasks.
>
> > The improvement over baselines is at odds and not pronounced enough. [...] It is good to analyze the response difference to drive deeper understanding.
>
> DeCoRe yields more accurate results than the majority of the baselines in most of the tasks. As shown in Tables 1, 2, and 3, DeCoRe achieves first place in six out of 11 tasks, second place in four, and third place in one, specifically:
>
> - 1st place: MemoTrap, NQ-Open (Open Book), TriviaQA, PopQA, NQ-Open (Closed Book), MuSiQue
> - 2nd place: XSum (factKB), NQ-Swap, TruthfulQA (MC), TruthfulQA (Gen)
> - 3rd place: IFEval
>
> In our Conclusions section (“Limitations” paragraph), we mention that, in some specific tasks, there may be baselines that achieve better results than DeCoRe. However, these baselines often offer limited improvements or yield significantly less accurate results in other tasks:
>
> - ITI: 1st place in TruthfulQA (MC) and TruthfulQA (Gen); Significant drops in all other tasks.
> - CAD: 1st place in XSum (factKB), NQ-Swap, MemoTrap; Not applicable to other tasks.
>
> On the other hand, we show that DeCoRe can yield state-of-the-art or competitive results on a wide array of tasks.
>
> We believe that the reason why ITI performs very well on TruthfulQA but not the others is because it is fine-tuned on TruthfulQA, which might have harmed the generalisation capability of the model. We believe that CAD yields significantly more accurate predictions on NQ-Swap and MemoTrap because it is designed to penalise the parametric knowledge of the model, thus making it very suitable for tasks that require total faithfulness to the context.
>
> We hope that these explanations clarify our position and look forward to any further feedback on these points.
>
> > It is good to explore alternative attention selection strategies, [...] it might be interesting to see whether certain attention heads are more related to shorter vs longer context retrieval.
>
> We have experiments with random selections in Appendix G. We referred to Appendix G.1 in the “Effect of Retrieval Head Masking on Task Performance of DeCoRe” paragraph. However, we will amend this to refer to the entire Appendix G for clarity.
> Regarding the idea of examining whether certain attention heads are more related to shorter vs. longer context retrieval, we agree that this is an intriguing direction, thank you for the suggestion. While it may fall outside the current scope of this paper, we believe it is a promising avenue for future work, and we plan to explore it in subsequent research.
>
> > It is good to explore some datasets with longer context (e.g., retrieving more passages) and compare the method there
>
> We have experiments using MuSiQue which requires a longer retrieved context (i.e., multi-hop reasoning using multiple documents). Additionally, we just conducted an experiment on Lost-in-the-Middle setups which may resemble retrieving more passages as per your suggestion:
>
> | Model               | Oracle | Gold at 0 | Gold at 4 | Gold at 9 |
> | ------------------- | ------ | --------- | --------- | --------- |
> | LLaMA3-8B-Instruct  | 69.68  | 52.92     | 45.61     | 44.48     |
> | + CAD               | 69.83  | 40.57     | 31.53     | 29.30     |
> | + DoLa (low)        | 69.68  | 52.88     | 45.76     | 44.37     |
> | + DoLa (high)       | 69.49  | 52.28     | 45.39     | 44.14     |
> | + DeCoRe_entropy    | 70.66  | 54.39     | 47.50     | 45.42     |
> | ------------------- | ------ | --------- | --------- | --------- |
> | LLaMA3-70B-Instruct | 71.07  | 60.49     | 52.99     | 49.00     |
> | + CAD               | 71.83  | 58.27     | 48.10     | 43.16     |
> | + DoLa (low)        | 71.07  | 60.45     | 52.96     | 49.04     |
> | + DoLa (high)       | 70.40  | 59.32     | 52.24     | 48.32     |
> | + DeCoRe_entropy    | 72.66  | 60.72     | 53.07     | 49.38     |
>
> We found that DeCoRe yields the most accurate responses, while the strongest contextual faithfulness baseline (i.e., CAD) suffers in this setup, which reinforces our previous hypothesis that CAD only performs well when the model has to be completely faithful to all given contexts.
>
> We will continue experimenting with all baselines if this addresses your concern.

---

> > ### Author Response · Authors · 2024-11-15
> > **Response to Reviewer 1dQs (cont.)**
> >
> > > It is good to report the robustness of those chosen hyperparameters
> >
> > We included an analysis of the impact of varying the number of masked retrieval heads in the Results section paragraph “Effect of Retrieval Head Masking on Task Performance of DeCoRe”. We also accompanied this analyses with detailed tables in Appendix G2-G4.
> >
> > Regarding in-context examples, we followed standard settings from previous studies to ensure consistency and comparability with existing benchmarks. However, if the reviewer would like to see additional robustness studies, we will aim to incorporate them in the revision before the rebuttal deadline.
> >
> > > Is there any difference in terms of generations from the proposed method vs others, e.g., fluency, length?
> >
> > We included several examples illustrating the fluency and coherence of the generated outputs of DeCoRe in Appendix K.2; however, this was not mentioned in the main text, which may have caused some oversight. Following your comment, we will expand more on the text quality generated by DeCoRe and refer to Appendix K at the end of Section 4 in the revised version of the paper.
> >
> > Regarding the length (i.e., number of generated tokens), there is no significant difference between the methods (with the exception of ITI which generates shorter answers, however inaccurate):
> >
> > | Model              | TruthfulQA | XSum  | MuSiQue CoT Open Book | MuSiQue CoT Closed Book |
> > | ------------------ | ---------- | ----- | --------------------- | ----------------------- |
> > | LLaMA3-8B-Instruct | 25.50      | 54.74 | 50.63                 | 44.41                   |
> > | ITI                | 19.06      | 29.37 | 24.71                 | 21.72                   |
> > | DoLa               | 25.12      | 54.97 | 50.35                 | 44.49                   |
> > | DeCoRe_entropy     | 30.79      | 52.31 | 54.76                 | 46.15                   |
> >
> > The difference is only in the correctness of the generation.
> >
> > We will incorporate this into the revised version of the paper. We look forward to any further analysis suggestions and aim to incorporate them before the rebuttal deadline.
> >
> > > What is the smaller LM used for DeCoRe lite?
> >
> > We use LLama3-70B-Instruct and LLama3-8B-Instruct as the base and masked LLMs, respectively. Thank you for spotting this overlook. We will rearrange the Experiment Setup chapter and add this change to the part mentioning the experimented DeCoRe variants.
> >
> > > What is the alpha value for DeCoRe static? How that is determined?
> >
> > We chose $\alpha = 0.5$ because it worked well in preliminary experiments across all tasks. We also analyse the downstream accuracy as a function of $\alpha$ in Appendix I. However, this was not referred to in the main text, which may have caused some oversight. We will add a reference to it in the revised version of the paper.
> >
> > > What is the computational cost for all compared methods, e.g., tflops?
> >
> > We have calculated the computational cost of the greedy decoding, CAD, DoLa, and DeCoRe, as detailed here:
> >
> > | Model              | TFLOPS |
> > | ------------------ | ------ |
> > | LLaMA3-8B-Instruct | 4.24   |
> > | CAD                | 8.44   |
> > | DoLa               | 4.28   |
> > | DeCoRe_static      | 4.32   |
> > | DeCoRe_entropy     | 4.32   |
> >
> > DeCoRe TFLOPS are not very different from greedy decoding. We believe that this is because our DeCoRe implementation leverages shared KV caching as opposed to CAD which forces completely separate forward passes. We will incorporate this result into the revised version of the paper.

---

> ### Comment · Reviewer_1dQs · 2024-11-19
>
> Thanks the authors for providing more details and results.
>
> There is no further question from my side.

---

> > ### Author Response · Authors · 2024-11-22
> > **Response to Reviewer 1dQs**
> >
> > We deeply appreciate your questions and comments, which helped us improve the quality of our paper. We are glad that our response addresses the points you raised. We would like to know if you would consider adjusting your review rating and confidence. We would be happy to address anything, otherwise.

---

### Official Review · Reviewer_6H8B · 2024-11-02

**Soundness:** 2
**Presentation:** 3
**Contribution:** 3
**Rating:** 6
**Confidence:** 4

**Summary:**

The paper introduces DeCoRe, a decoding strategy for reducing hallucinations in LLMs. DeCoRe works by contrasting between a LLM (that can be augmented with additional context), and the same LLM where the attention heads that are responsible for extracting information from the context (i.e., retrieval heads) are masked, assuming masking these heads induces hallucinations and thus contrasting between the two distributions increase factuality. The weight given to each distribution can be controlled by a hyperparameter that estimates the model’s uncertainty using the entropy of the model. Experimental results show that DeCoRe significantly improves contextual faithfulness on multiple tasks including summarization and open-domain QA.

**Strengths:**

- The methodology of contrasting between a variant where retrieval heads are masked is interesting and has benefits over previous methods, for example in applicability.
- The experimental setting is thorough, comparing against six strong baselines on three tasks (reading comprehension, summarization, and instruction-following) with multiple datasets, including XSum, MemoTrap, IFEval NQ, PopQA, and TruthfulQA.
- The advancements on MemoTrap in comparison to the baselines are significant.
- The paper is overall well-written and easy to follow.

**Weaknesses:**

- It seems that in several experiments, the improvements from DeCoRe are either marginal or it even performs significantly worse than other methods. For example, as presented in Tab.1 CAD significantly outperforms DeCoRe on NQ-Swap whilst performing similarly on other benchmarks (within a one point margin). While I agree with the paper that DeCoRe has higher applicability I am not entirely sure that CAD cannot be used for MemoTrap (see questions).

- The paper also claims that DeCoRe can increase factuality (Tab.2) and multi-hop reasoning with CoT (Tab.3) but from looking at these tables, in most cases that DeCoRe is best-performing it is within a one-point margin to the second-best performing model (as stated in the paper, there are also cases where the baselines perform better). Additionally, in all experiments three DeCoRe variants are compared against the baselines, which can be a bit misleading. I believe the paper might benefit from an aggregated score for each method and error bars or statistical tests for the main claims. This can simplify understanding the significance of the main results.

- I believe the paper might benefit from a qualitative analysis that supports the main claims. For example, when is using conditional entropy beneficial and when does it hurt performance, and when does DeCoRe outperform the baselines?

**Questions:**

Please see weaknesses, I am willing to reconsider my score.

Regarding CAD and MemoTrap, maybe I am missing something, but perhaps it’s possible to use the task as the additional context, e.g., for the task - “Write a quote that ends in the word "talk": Don't bark if you can't”, the additional context can be “Write a quote that ends in the word "talk".

Again, maybe I am missing something, but lines 330-331 state that DeCoRe has ‘competitive performance’ on NQ-Swap, but CAD outperforms DeCoRe by 8+ points. What gains do you consider substantial in the context of the experiments and the datasets in the paper?

---

> ### Author Response · Authors · 2024-11-15
> **Response to Reviewer 6H8B**
>
> Thank you for the review!
>
> > It seems that in several experiments, the improvements from DeCoRe are either marginal or it even performs significantly worse than other methods.
>
> DeCoRe yields more accurate results than the majority of the baselines in most of the tasks. As shown in Tables 1, 2, and 3, DeCoRe achieves first place in six out of 11 tasks, second place in four, and third place in one, specifically:
>
> - 1st place: MemoTrap, NQ-Open (Open Book), TriviaQA, PopQA, NQ-Open (Closed Book), MuSiQue
> - 2nd place: XSum (factKB), NQ-Swap, TruthfulQA (MC), TruthfulQA (Gen)
> - 3rd place: IFEval
>
> In our Conclusions section (“Limitations” paragraph), we mention that, in some specific tasks, there may be baselines that achieve better results than DeCoRe. However, these baselines often offer limited improvements or yield significantly less accurate results in other tasks:
>
> - ITI: 1st place in TruthfulQA (MC) and TruthfulQA (Gen); Significant drops in all other tasks.
> - CAD: 1st place in XSum (factKB), NQ-Swap, MemoTrap; Not applicable to other tasks.
>
> On the other hand, we show that DeCoRe can yield state-of-the-art or competitive results on a wide array of tasks.
>
> We believe that the reason why ITI performs very well on TruthfulQA but not the others is because it is fine-tuned on TruthfulQA, which might have harmed the generalisation capability of the model. We believe that CAD yields significantly more accurate predictions on NQ-Swap and MemoTrap because it is designed to penalise the parametric knowledge of the model, thus very suitable for these tasks that require total faithfulness to the context.
>
> Additionally, in our response to reviewer 1dQs, we also added a new experiment using multi-documents retrieval where only one document contains the answer with the Lost-in-the-Middle setup. We found that DeCoRe yields the most accurate responses, while the strongest contextual faithfulness baseline (CAD) suffers in this setup, which reinforce our previous hypothesis that CAD only performs well when the model have to be completely faithful to all given contexts:
>
> | Model               | Oracle | Gold at 0 | Gold at 4 | Gold at 9 |
> | ------------------- | ------ | --------- | --------- | --------- |
> | LLaMA3-8B-Instruct  | 69.68  | 52.92     | 45.61     | 44.48     |
> | + CAD               | 69.83  | 40.57     | 31.53     | 29.30     |
> | + DoLa (low)        | 69.68  | 52.88     | 45.76     | 44.37     |
> | + DoLa (high)       | 69.49  | 52.28     | 45.39     | 44.14     |
> | + DeCoRe_entropy    | 70.66  | 54.39     | 47.50     | 45.42     |
> | ------------------- | ------ | --------- | --------- | --------- |
> | LLaMA3-70B-Instruct | 71.07  | 60.49     | 52.99     | 49.00     |
> | + CAD               | 71.83  | 58.27     | 48.10     | 43.16     |
> | + DoLa (low)        | 71.07  | 60.45     | 52.96     | 49.04     |
> | + DoLa (high)       | 70.40  | 59.32     | 52.24     | 48.32     |
> | + DeCoRe_entropy    | 72.66  | 60.72     | 53.07     | 49.38     |
>
> We hope that these explanations clarify our position and look forward to any further feedback on these points.

---

> ### Author Response · Authors · 2024-11-15
> **Response to Reviewer 6H8B (cont.)**
>
> > I believe the paper might benefit from an aggregated score for each method and error bars or statistical tests for the main claims. This can simplify understanding the significance of the main results.
>
> Thank you for the suggestion. Here is the aggregated score per table, and the combined average:
>
> | Model                     | Faithfulness ↑ | Factuality ↑ | CoT ↑   | **Overall Avg ↑** |
> |---------------------------|----------------|--------------|---------|-------------------|
> | **LLaMA3-8B-Instruct**    | 60.43          | 39.72        | 37.67   | 45.94             |
> | + ITI                     | 50.21          | 39.87        | 23.08   | 37.72             |
> | + CAD                     | 66.57¹         | 39.72¹       | 38.23   | _48.17_           |
> | + DoLA (Low)              | 60.17          | 39.53        | 37.79   | 45.83             |
> | + DoLA (High)             | 60.35          | 39.43        | 37.67¹  | 45.82             |
> | + AD                      | 59.84          | 38.34        | 37.49   | 45.22             |
> | + **DeCoRe Static**       | 63.64          | 40.67        | 39.08   | 47.79             |
> | + **DeCoRe Entropy**      | 64.86          | 41.40        | 39.51   | **48.59**         |
>
> ¹ Missing values filled with base model scores.
>
> We calculated the aggregated scores as such:
>
> - Faithfulness Score is an average of XSum (ROUGE-L, BERTScore-F1, factKB), MemoTrap (Macro Accuracy, Micro Accuracy), IFEval (Prompt Accuracy, Instruct Accuracy), NQ-Open EM, NQ-Swap EM.
> - Factuality Score is an average of Averaged MC scores, TriviaQA EM, PopQA EM, TruthfulQA Generation (% T ∩ I), NQ-Open EM.
> - CoT score: MuSiQue Closed Book without CoT EM, MuSiQue Open Book without CoT EM, MuSiQue Closed Book with CoT EM, MuSiQue Open Book with CoT EM.
>
> From the table, we observe that the DeCoRe Entropy method achieves the highest overall average score of 48.59, indicating a consistent improvement across all evaluation metrics compared to the base model and other decoding strategies. The CAD method also demonstrates strong performance with an overall average of 48.17.
>
> Regarding Statistical Tests, we appreciate your suggestion to include error bars or statistical tests to simplify understanding the significance of the main results. However, since our evaluations are conducted using a greedy decoding setup, thus running multiple trials would yield identical results, rendering standard statistical tests inapplicable in this context.
>
> Nevertheless, we acknowledge the importance of statistical validation and are open to exploring alternative methods that could provide insights into the significance of our results. If you have specific statistical tests or methodologies in mind that are suitable for deterministic outputs, we would be eager to consider them in our analysis.
>
> > I believe the paper might benefit from a qualitative analysis that supports the main claims. For example, when is using conditional entropy beneficial and when does it hurt performance, and when does DeCoRe outperform the baselines?
>
> Thank you for the suggestion! In this study, using conditional entropy to control the contrastive penalty (DeCoRe_entropy) is generally beneficial compared to using a static penalty (DeCoRe_static). In cases when DeCoRe_static outperforms DeCoRe_entropy, the difference is negligible. We look forward to further feedback on this point and aim to incorporate any further suggested analyses before the rebuttal deadline.
>
> > Regarding CAD and MemoTrap, maybe I am missing something, but perhaps it’s possible to use the task as the additional context, e.g., for the task - “Write a quote that ends in the word "talk": Don't bark if you can't”, the additional context can be “Write a quote that ends in the word "talk".
>
> Thank you for the suggestion! We have added this experiment to the manuscript. CAD outperforms DeCoRe on MemoTrap.
>
> | Model                        | Method | Macro Acc | Micro Acc |
> |------------------------------|--------|-----------|-----------|
> | **LLaMA3-8b-Instruct**       | CAD    | 76.58     | 76.76     |
> |                              | DeCoRe | 74.14     | 74.87     |
> |------------------------------|--------|-----------|-----------|
> | **LLaMA3-70b-Instruct**      | CAD    | 83.58     | 83.89     |
> |                              | DeCoRe | 74.07     | 73.65     |
> |------------------------------|--------|-----------|-----------|
> | **Mistral-7b-Instruct-v0.3** | CAD    | 83.63     | 81.49     |
> |                              | DeCoRe | 77.54     | 76.96     |
> |------------------------------|--------|-----------|-----------|
> | **Qwen2-7b-Instruct**        | CAD    | 87.52     | 86.14     |
> |                              | DeCoRe | 83.80     | 82.04     |
>
> Despite being second best in several tasks, as you stated, the flexibility of DeCoRe allows it to be applicable to more tasks.

---

> > ### Author Response · Authors · 2024-11-15
> > **Response to Reviewer 6H8B (cont.)**
> >
> > > lines 330-331 state that DeCoRe has ‘competitive performance’ on NQ-Swap, but CAD outperforms DeCoRe by 8+ points. What gains do you consider substantial in the context of the experiments and the datasets in the paper?
> >
> > We agree with your comment – we rephrased “competitive performance” to “second-best results”. What we intended to say is that DeCoRe achieves significantly higher accuracy when compared to other baselines, except for CAD.

---

> ### Comment · Reviewer_6H8B · 2024-11-19
> **Response to authors**
>
> Thank you for the detailed response! I have some follow-up questions.
>
> - In the response, you mentioned that the manuscript was updated with the CAD results for MemoTrap, but I could not find these in the paper, am I missing something? I believe the paper could also benefit from the aggregated results.
>
> - Regarding statistical significance, I recommend checking [1] or standard methods to calculate variance. Do you think any of these methods is applicable to your setting?
>
> I agree that the paper has merits and I could be missing something, but my main concern is that due to the small differences between CAD, DeCoRe-Static, and DeCoRe-Entropy (<1 point on aggregate), and similarly small deltas in Tab.2 and Tab.3, some of the main claims in the paper may need to be slighly softened (specifically the improvement in accuracy of DeCoRe over baselines and advantages of entropy as a guide and especially if trends are not statistically significant).
>
> [1] The Hitchhiker’s Guide to Testing Statistical Significance in Natural Language Processing

---

> > ### Author Response · Authors · 2024-11-22
> > **Response to Reviewer 6H8B**
> >
> > Thank you very much for your reply and very good suggestions!
> >
> > > In the response, you mentioned that the manuscript was updated with the CAD results for MemoTrap, but I could not find these in the paper, am I missing something? I believe the paper could also benefit from the aggregated results.
> >
> > Thanks for noticing! We uploaded the revised version of the paper including the CAD and the aggregated results. We also followed your suggestions to tone down the claims in the main text. We remain available if you would like us to correct anything.
> >
> > > Regarding statistical significance, I recommend checking [1] or standard methods to calculate variance. Do you think any of these methods is applicable to your setting?
> >
> > Thank you for the suggestion! We agree that the methods in the referred paper can be used in our case. We have added pairwise statistical significance analyses in Appendix H of the revised paper (referred to in the Results section). Specifically, we have applied the following methods:
> >
> > - McNemar's Test: We used McNemar's Test to analyse categorical outcomes such as Accuracy and Exact Match (EM) metrics.
> > - Bootstrap Resampling: For continuous metrics like ROUGE-L, BERTScore-F1, and factKB, we employed bootstrap resampling techniques to estimate variances and compute confidence intervals.
> >
> > The analysis confirms that many of the improvements observed with DeCoRe are statistically significant, particularly in contextual faithfulness setup. We appreciate your insightful suggestion, which has strengthened the methodological soundness of our work. We remain available if you would like us to add more analyses.
> >
> > > Qualitative analysis
> >
> > You also previously mentioned the point about qualitative analyses. In response, we have added an additional qualitative analysis on the length of generations, which was also requested by another reviewer (1dQs).
> >
> > We found that there is no significant difference in terms of generation length between the methods (with the exception of ITI which generates shorter answers, however inaccurate). The difference is only in the correctness of the generation.
> >
> > | Model              | TruthfulQA | XSum  | MuSiQue CoT Open Book | MuSiQue CoT Closed Book |
> > | ------------------ | ---------- | ----- | --------------------- | ----------------------- |
> > | LLaMA3-8B-Instruct | 25.50      | 54.74 | 50.63                 | 44.41                   |
> > | ITI                | 19.06      | 29.37 | 24.71                 | 21.72                   |
> > | DoLa               | 25.12      | 54.97 | 50.35                 | 44.49                   |
> > | DeCoRe_entropy     | 30.79      | 52.31 | 54.76                 | 46.15                   |
> >
> > We added these results in the revised version of the paper too.

---

> > > ### Comment · Reviewer_6H8B · 2024-11-24
> > > **Response to authors**
> > >
> > > Thank you for your response and for uploading a new version of the paper! I see that some of the text is colored blue (e.g., Appendix H parts of Sec.4), is this intended?

---

> > > > ### Author Response · Authors · 2024-11-24
> > > >
> > > > Yes, they are intended for the rebuttal phase. The blue texts in the revised draft indicate the changes, making it easier to spot them. We will change them back to black in camera-ready.

---

> > > > > ### Comment · Reviewer_6H8B · 2024-11-25
> > > > >
> > > > > Thank you for the clarification. I have no follow-up questions at this time.
> > > > >
> > > > > To summarize, while the competitive performance of some baselines (especially CAD) and the small deltas between the DeCoRe variants are limitations, the method is novel, and the authors made significant efforts to address these limitations during the discussion period (by adding experiments, introducing statistical tests, and softening the main claims). As a result, I have adjusted my score.

---

> > > > > > ### Author Response · Authors · 2024-11-25
> > > > > > **Thank you!**
> > > > > >
> > > > > > Thank you for your thoughtful feedback and for recognising our work and efforts to address the concerns raised. We greatly appreciate your constructive input!

---

### Official Review · Reviewer_yWX4 · 2024-11-04

**Soundness:** 3
**Presentation:** 3
**Contribution:** 3
**Rating:** 5
**Confidence:** 4

**Summary:**

This paper introduces a training-free decoding approach named DeCoRe for reducing hallucinations in LLMs. By identifying specific retrieval heads that influence the generation of grounded content, DeCoRe employs a masking strategy to induce hallucinations, then applies contrastive decoding to distinguish accurate from hallucinatory outputs. Experiments show improvements in contextual faithfulness on datasets like XSum and factual accuracy on tasks such as TriviaQA.

**Strengths:**

1. The paper is clearly structured, making complex ideas accessible and easy to follow.

2. The paper introduces a novel decoding approach that contrasts outputs from the original and masked LLM, effectively reducing ungrounded content.

3. The paper  incorporates an entropy-controlled mechanism, allowing the method to adjust the level of contrast applied depending on the model’s uncertainty.

**Weaknesses:**

1. The model's performance improvements are inconsistent across different datasets, and it fails to outperform baseline models in many settings (e.g., XSum, IFEval). This is inconsistent with the summary conclusions presented in the abstract.

2. The paper only compares a limited number of baselines, primarily focusing on contrastive methods, without evaluating against many related models.

3. The approach in this paper appears to be related to self-consistency and should better include a discussion on this topic, along with relevant baseline comparisons.

**Questions:**

n/a

---

> ### Author Response · Authors · 2024-11-15
> **Response to Reviewer yWX4**
>
> Thank you for the review!
>
> > The model's performance improvements are inconsistent across different datasets, and it fails to outperform baseline models in many settings (e.g., XSum, IFEval). This is inconsistent with the summary conclusions presented in the abstract.
>
> DeCoRe yields more accurate results than the majority of the baselines in most of the tasks. As shown in Tables 1, 2, and 3, DeCoRe achieves first place in six out of 11 tasks, second place in four, and third place in one, specifically:
>
> - 1st place: MemoTrap, NQ-Open (Open Book), TriviaQA, PopQA, NQ-Open (Closed Book), MuSiQue
> - 2nd place: XSum (factKB), NQ-Swap, TruthfulQA (MC), TruthfulQA (Gen)
> - 3rd place: IFEval
>
> In our Conclusions section (“Limitations” paragraph), we mention that, in some specific tasks, there may be baselines that achieve better results than DeCoRe. However, these baselines often offer limited improvements or yield significantly less accurate results in other tasks:
>
> - ITI: 1st place in TruthfulQA (MC) and TruthfulQA (Gen); Significant drops in all other tasks.
> - CAD: 1st place in XSum (factKB), NQ-Swap, MemoTrap; Not applicable to other tasks.
>
> On the other hand, we show that DeCoRe can yield state-of-the-art or competitive results on a wide array of tasks.
>
> We believe that the reason why ITI performs very well on TruthfulQA but not the others is because it is fine-tuned on TruthfulQA, which might have harmed the generalisation capability of the model. We believe that CAD yields significantly more accurate predictions on NQ-Swap and MemoTrap because it is designed to penalise the parametric knowledge of the model, thus making it very suitable for these tasks that require total faithfulness to the context.
>
> Additionally, in our response to reviewer 1dQs, we also added a new experiment using multi-document retrieval where only one document contains the answer with the Lost-in-the-Middle setup. We found that DeCoRe yields the most accurate responses, while the strongest contextual faithfulness baseline (CAD) suffers in this setup, which reinforces our previous hypothesis that CAD only performs well when the model has to be completely faithful to all given contexts:
>
> | Model               | Oracle | Gold at 0 | Gold at 4 | Gold at 9 |
> | ------------------- | ------ | --------- | --------- | --------- |
> | LLaMA3-8B-Instruct  | 69.68  | 52.92     | 45.61     | 44.48     |
> | + CAD               | 69.83  | 40.57     | 31.53     | 29.30     |
> | + DoLa (low)        | 69.68  | 52.88     | 45.76     | 44.37     |
> | + DoLa (high)       | 69.49  | 52.28     | 45.39     | 44.14     |
> | + DeCoRe_entropy    | 70.66  | 54.39     | 47.50     | 45.42     |
> | ------------------- | ------ | --------- | --------- | --------- |
> | LLaMA3-70B-Instruct | 71.07  | 60.49     | 52.99     | 49.00     |
> | + CAD               | 71.83  | 58.27     | 48.10     | 43.16     |
> | + DoLa (low)        | 71.07  | 60.45     | 52.96     | 49.04     |
> | + DoLa (high)       | 70.40  | 59.32     | 52.24     | 48.32     |
> | + DeCoRe_entropy    | 72.66  | 60.72     | 53.07     | 49.38     |
>
> We hope that these explanations clarify our position and look forward to any further feedback on these points.
>
> > The paper only compares a limited number of baselines, primarily focusing on contrastive methods, without evaluating against many related models.
>
> We compare DeCoRe against five strong baselines on top of the greedy decoding baseline:
>
> - 3 Contrastive decoding methods (CAD, CD, DoLa). Despite being not applicable to some tasks, We included CAD as it is a very strong baseline.
> - 2 Non-contastive decoding methods: ITI and Activation Decoding. ITI specifically requires fine-tuning, as opposed to the other selected baselines.
>
> We believe this is sufficient; however, if the reviewer would like to specify additional baselines, we will aim to incorporate them in the revision before the rebuttal deadline.
>
> > The approach in this paper appears to be related to self-consistency and should better include a discussion on this topic, along with relevant baseline comparisons.
>
> Thank you for this suggestion – indeed, DeCoRe can be seen as related to self-consistency as it involves combining multiple predictions; however, while self-consistency mainly relies on majority voting among generations from a single model, DeCoRe attempts to amplify the difference between a base model and a masked model. We will add a discussion on this link between self-consistency and DeCoRe in the Related Work section of the revised version of the paper. Specifically: “DeCoRe can also be viewed as an asymmetric version of self-consistency (Wang et al., 2023); rather than relying on majority voting among generations from a single model, DeCoRe amplifies the difference between a base model and its masked variant.”
>
> We hope that this answers your concerns and we look forward to any further feedback.

---

> ### Author Response · Authors · 2024-11-22
> **Feedback on our rebuttal**
>
> Dear Reviewer,
>
> Since the deadline for the rebuttal phase is approaching, we kindly ask you to confirm whether we have sufficiently addressed your comments or if there are any remaining concerns.
>
> Thank you!
>
> Authors

---

> > ### Author Response · Authors · 2024-11-25
> > **Any remaining concerns?**
> >
> > Thank you for your feedback! With the discussion period closing in less than 48 hours, we want to ensure we have addressed your concerns and that our additional experiments meet your expectations. We’d appreciate your feedback on our responses!

---

### Official Review · Reviewer_9XYH · 2024-11-04

**Soundness:** 3
**Presentation:** 3
**Contribution:** 2
**Rating:** 5
**Confidence:** 3

**Summary:**

The paper addresses the issue of hallucinations in Large Language Models (LLMs), where outputs may be unfaithful or factually incorrect. It identifies specific attention heads, called retrieval heads, within the Transformer architecture that are crucial for extracting relevant information. The authors hypothesize that masking these heads can induce hallucinations. To tackle this, they propose a novel decoding strategy named Decoding by Contrasting Retrieval Heads (DeCoRe), which does not require additional training. DeCoRe works by comparing outputs from the base LLM and a masked version to reduce hallucinations, guided by conditional entropy. Experiments show that DeCoRe significantly enhances performance in tasks demanding high contextual accuracy, such as summarization, instruction following, and open-book question answering, with notable improvements in various benchmarks.

**Strengths:**

1.	The paper is well-written.
2.	The research includes thorough experiments that effectively demonstrate the capability of DeCoRe in mitigating hallucinations across various tasks.
3.	The proposed method, DeCoRe, is training-free and useful for hallucination mitigation.

**Weaknesses:**

1.	The paper only provides evaluations on the effectiveness of mitigating hallucination. However, as a decoding method, it is also important to show the generated texts are coherent and fluent. Therefore, it is required to provide evidence to show this decoding method can generate text with high quality with low hallucination.
2.	It seems that in the experiments, many baselines can achieve better results than DeCoRe.
3.	The paper seems a combination of the existing findings, which applies the retrieval heads (Wu et al., 2024) to the Contrastive Decoding (Li et al., 2023) method, potentially limiting the originality of the contribution.

**Questions:**

See weaknesses

---

> ### Author Response · Authors · 2024-11-15
> **Response to Reviewer 9XYH**
>
> Thank you for the review!
>
> > it is required to provide evidence to show this decoding method can generate text with high quality with low hallucination.
>
> Thank you for spotting this – we included several examples illustrating the fluency and coherence of the generated outputs of DeCoRe in Appendix K.2; however, this was not mentioned in the main text, which may have caused some oversight. Following your comment, we will expand more on the text quality generated by DeCoRe and refer to Appendix K at the end of Section 4 in the revised version of the paper.
>
> > It seems that in the experiments, many baselines can achieve better results than DeCoRe.
>
> DeCoRe yields more accurate results than the majority of the baselines in most of the tasks. As shown in Tables 1, 2, and 3, DeCoRe achieves first place in six out of 11 tasks, second place in four, and third place in one, specifically:
>
> - 1st place: MemoTrap, NQ-Open (Open Book), TriviaQA, PopQA, NQ-Open (Closed Book), MuSiQue
> - 2nd place: XSum (factKB), NQ-Swap, TruthfulQA (MC), TruthfulQA (Gen)
> - 3rd place: IFEval
>
> In our Conclusions section (“Limitations” paragraph), we mention that, in some specific tasks, there may be baselines that achieve better results than DeCoRe. However, these baselines often offer limited improvements or yield significantly less accurate results in other tasks:
>
> - ITI: 1st place in TruthfulQA (MC) and TruthfulQA (Gen); Significant drops in all other tasks.
> - CAD: 1st place in XSum (factKB), NQ-Swap, MemoTrap; Not applicable to other tasks.
>
> On the other hand, we show that DeCoRe can yield state-of-the-art or competitive results on a wide array of tasks.
>
> We believe that the reason why ITI performs very well on TruthfulQA but not the others is because it is fine-tuned on TruthfulQA, which might have harmed the generalisation capability of the model. We believe that CAD yields significantly more accurate predictions on NQ-Swap and MemoTrap because it is designed to penalise the parametric knowledge of the model, thus very suitable for these tasks that require total faithfulness to the context.
>
> Additionally, in our response to reviewer 1dQs, we also added a new experiment using multi-documents retrieval where only one document contains the answer with the Lost-in-the-Middle setup. We found that DeCoRe yields the most accurate responses, while the strongest contextual faithfulness baseline (CAD) suffers in this setup, which reinforce our previous hypothesis that CAD only performs well when the model have to be completely faithful to all given contexts:
>
> | Model               | Oracle | Gold at 0 | Gold at 4 | Gold at 9 |
> | ------------------- | ------ | --------- | --------- | --------- |
> | LLaMA3-8B-Instruct  | 69.68  | 52.92     | 45.61     | 44.48     |
> | + CAD               | 69.83  | 40.57     | 31.53     | 29.30     |
> | + DoLa (low)        | 69.68  | 52.88     | 45.76     | 44.37     |
> | + DoLa (high)       | 69.49  | 52.28     | 45.39     | 44.14     |
> | + DeCoRe_entropy    | 70.66  | 54.39     | 47.50     | 45.42     |
> | ------------------- | ------ | --------- | --------- | --------- |
> | LLaMA3-70B-Instruct | 71.07  | 60.49     | 52.99     | 49.00     |
> | + CAD               | 71.83  | 58.27     | 48.10     | 43.16     |
> | + DoLa (low)        | 71.07  | 60.45     | 52.96     | 49.04     |
> | + DoLa (high)       | 70.40  | 59.32     | 52.24     | 48.32     |
> | + DeCoRe_entropy    | 72.66  | 60.72     | 53.07     | 49.38     |
>
> We hope that these explanations clarify our position and look forward to any further feedback on these points.
>
> > The paper seems a combination of the existing findings, which applies the retrieval heads (Wu et al., 2024) to the Contrastive Decoding (Li et al., 2023) method, potentially limiting the originality of the contribution.
>
> Indeed, we agree that DeCoRe is grounded on intuitions from the retrieval heads and contrastive decoding literature; however, our work is the first to integrate retrieval head masking and contrastive decoding for hallucination mitigation. Furthermore, we propose a simple method for automatically selecting the DeCoRe hyper-parameters. We clarify this in the revised version.

---

> > ### Author Response · Authors · 2024-11-22
> > **Feedback on our rebuttal**
> >
> > Dear Reviewer,
> >
> > Since the deadline for the rebuttal phase is approaching, we kindly ask you to confirm whether we have sufficiently addressed your comments or if there are any remaining concerns.
> >
> > Thank you!
> >
> > Authors

---

> > > ### Comment · Reviewer_9XYH · 2024-11-25
> > >
> > > Thanks. I prefer to keep the score

---

### Author Response · Authors · 2024-11-22
**General response**

We thank the reviewers for their time and detailed reviews.

## Strengths

We appreciate that the reviewers recognise the strengths of our work:

| Category | Comment | Reviewer |
|---|---|---|
| **Importance and Novelty** | "Given the wide range of LM applications, developing efficient and reliable hallucination mitigation methods is a very important research direction." | 1dQs |
| | "The methodology of contrasting between a variant where retrieval heads are masked is interesting and has benefits over previous methods, for example in applicability." | 6H8B |
| | "The proposed method, DeCoRe, is training-free and useful for hallucination mitigation." | 9XYH |
| | "The proposed method is a nice extension of contrastive decoding with recent attention head function insights." | 1dQs |
| | "Introduces a novel decoding approach that contrasts outputs from the original and masked LLM, effectively reducing ungrounded content." | yWX4 |
| | "Incorporates an entropy-controlled mechanism, allowing the method to adjust the level of contrast applied depending on the model’s uncertainty." | yWX4 |
| **Experimental Validation and Results** | "The research includes thorough experiments that effectively demonstrate the capability of DeCoRe in mitigating hallucinations across various tasks." | 9XYH |
| | "The experimental setting is thorough, comparing against six strong baselines on three tasks (reading comprehension, summarization, and instruction-following) with multiple datasets, including XSum, MemoTrap, IFEval NQ, PopQA, and TruthfulQA." | 6H8B |
| | "The experiment designs are mostly reasonable with several recent baselines to validate the effectiveness of the proposed method." | 1dQs |
| | "The advancements on MemoTrap in comparison to the baselines are significant." | 6H8B |
| **Clarity/Presentation** | "The paper is easy to follow with the necessary background." | 1dQs |
| | "The paper is well-written." | 9XYH |
| | "The paper is clearly structured, making complex ideas accessible and easy to follow." | yWX4 |
| | "The paper is overall well-written and easy to follow." | 6H8B |

## Amendments

We deeply appreciate their feedback, and to that end, we made several amendments to the manuscript, which we believe significantly improved its quality.

| **Category** | **Amendment** | **Reviewer** |
|---|---|---|
| **Experimental Results** | Added Lost-in-the-Middle experiments in the Results section (in the “DeCoRe Mitigates Faithfulness Hallucinations amidst Distractor Documents” paragraph and in Table 2). | 1dQs |
| | Added average performance in Tables [1](https://anonymous.4open.science/r/decore-4FB7/docs/assets/decore_rebuttals/table_1.png), [2](https://anonymous.4open.science/r/decore-4FB7/docs/assets/decore_rebuttals/table_2_and_explanation.png), [3](https://anonymous.4open.science/r/decore-4FB7/docs/assets/decore_rebuttals/table_3.png), [4](https://anonymous.4open.science/r/decore-4FB7/docs/assets/decore_rebuttals/table_4.png), and overall average performance in [Table 5](https://anonymous.4open.science/r/decore-4FB7/docs/assets/decore_rebuttals/table_5.png). | 6H8B |
| **Statistical Analysis** | Added statistical test analyses in Appendices [H.1](https://anonymous.4open.science/r/decore-4FB7/docs/assets/decore_rebuttals/appendix_h1_table_22.png), [H.2](https://anonymous.4open.science/r/decore-4FB7/docs/assets/decore_rebuttals/appendix_h2_table_23.png), and [H.3](https://anonymous.4open.science/r/decore-4FB7/docs/assets/decore_rebuttals/appendix_h3_table_24.png) and toned down claims of significant improvements when they were not statistically significant in several factuality evaluation tasks. | 6H8B |
| **Computational Efficiency** | Added a section in [Appendix K.4](https://anonymous.4open.science/r/decore-4FB7/docs/assets/decore_rebuttals/appendix_k4_table_33.png) to show the computational efficiency (TFLOPS) of DeCoRe, referred to in the Results section. | 1dQs |
| **Qualitative Evaluation** | Added a reference to Appendix L.2 at the end of the Results section; Appendix L.2 shows generation examples for qualitative evaluation purposes. | 9XYH, 6H8B, 1dQs |
| | Added an analysis of the average length of generations in Appendix [L.1](https://anonymous.4open.science/r/decore-4FB7/docs/assets/decore_rebuttals/appendix_l1_table_34.png). | 1dQs |
| **Additional Clarification** | Restructured the Experimental Setup section (3.4) to include details about DeCoRe_entropy-lite. | 1dQs |
| | Added a discussion on the relationship between DeCoRe and self-consistency in the Related Works section. | yWX4 |

All changes are highlighted in blue in the revised draft.

---

> ### Author Response · Authors · 2024-11-26
> **Summary of Rebuttals**
>
> ## Main comments addressed in the response
>
> ### **All Reviewers**
>
> **Inconsistent improvements across tasks**: This is the main comment mentioned by all reviewers. In the revised version, we have addressed this by:
>
> - **Adding Additional Lost-in-the-Middle Experiments** (Table 2): These experiments further validate DeCoRe's effectiveness in contexts with distractor documents.
> - **Including Statistical Analyses** (Appendix H): We performed statistical significance tests to support our claims and soften our claims where necessary.
> - **Providing Aggregate Scores** in Table 1, 2, 3, 4, and 5: We included average performance metrics to give a clearer picture of DeCoRe's overall effectiveness across tasks.
>
> **Reviewer 6H8B** acknowledged our efforts, stated that we have made *“significant efforts to address these limitations during the discussion period (by adding experiments, introducing statistical tests, and softening the main claims)”*. All changes are currently written in blue in the revised manuscript.
>
> In summary, **DeCoRe achieves more accurate results compared to most baselines in the majority of tasks**:
>
> - **1st place** in **10 out of 17 tasks** (NQ-Open (Oracle), NQ-Open (Gold at 1), NQ-Open (Gold at 4th), NQ-Open (Gold at 9th), NQ-Open (Closed book), TriviaQA, PopQA, MuSiQue (without CoT, Closed Book), MuSiQue (without CoT, Open Book), MuSiQue (with CoT, Open Book));
> - **2nd place** in **6 out of 17 tasks** (XSum, MemoTrap, NQ-Swap, TruthfulQA (MC), TruthfulQA (Gen), MuSiQue (with CoT, Closed Book));
> - **<2nd place** in **1 out of 17 task** (IFEval).
>
> In cases where a baseline outperforms DeCoRe on specific tasks, these improvements are often marginal or come at the cost of significantly worse performance on other tasks. For instance:
> - **ITI**: 1st place in TruthfulQA, while yielding **significant drops in all other tasks**;
> - **CAD**: 1st place in XSum (factKB), NQ-Swap, MemoTrap, while yielding **significant drops in Lost-in-the-Middle experiments** and **not applicable to factuality evaluation tasks**.
>
> DeCoRe consistently yields **state-of-the-art or competitive performance across a wide variety of tasks**, making it a **broadly applicable** and **reliable solution**.
>
> ### **Specific Reviewers**
>
> #### **Reviewer yWX4**
>
> **Reviewer yWX4** assigned a borderline rating of **5**, while acknowledging several strengths of our work. We have addressed their comments as follows:
>
> - **Weakness 1 (Inconsistent Improvements)**: Addressed as per the actions taken for all reviewers. Additionally, **Reviewers 6H8B and 1dQs** have acknowledged our efforts in addressing this limitation.
> - **Weakness 2 (Limited Baseline Comparisons)**: In our response, we clarified that DeCoRe has been compared against **five strong baselines in addition to the greedy decoding baseline**. These include **three contrastive decoding methods (CAD, CD, DoLa)** and **two non-contrastive methods (ITI, Activation Decoding)**. **Reviewers 9XYH, 6H8B, and 1dQs** have acknowledged the thoroughness of our experiments.
> - **Weakness 3 (Relation to Self-Consistency)**: We added **a discussion in the Related Work** section on the relationship between DeCoRe and self-consistency methods.
>
> #### **Reviewer 9XYH**
>
> **Reviewer 9XYH** assigned a borderline rating of **5**, while acknowledging the comprehensive experiments that demonstrates the capability of DeCoRe in mitigating hallucinations. We have addressed their comments as follows:
>
> - **Weakness 1 (Qualitative Examples of Coherent Text)**: We added a reference to **Appendix L.2**, which includes examples illustrating the fluency and coherence of outputs generated by DeCoRe.
> - **Weakness 2 (Baselines Achieving Better Results)**: Addressed as per the actions taken for all reviewers. **Reviewers 6H8B and 1dQs** have acknowledged our efforts in addressing this limitation.
> - **Weakness 3 (Novelty of the Work)**: In our rebuttal, we clarified that our work is **the first to integrate retrieval head masking and contrastive decoding** for hallucination mitigation. Additionally, we introduced **a simple method for automatically selecting the DeCoRe hyperparameters**. **Reviewers yWX4, 6H8B, and 1dQs** have acknowledged the novelty of our approach.

---

### Author Response · Authors · 2024-12-03
**Summary of Contributions**

We thank the reviewers for their insightful comments. Here is a **summary of our contributions**:

- **A Novel Contrastive Decoding Mechanism**: We introduce **DeCoRe**, which **contrasts** outputs from the base LLM and a variant where its **retrieval heads are masked**, effectively reducing hallucinations without requiring fine-tuning or significant computational overhead;
- **Conditional Entropy Guidance**: DeCoRe utilises **conditional entropy** --a predictor of potential hallucinations-- to dynamically modulate the contrastive weight hyperparameter during the decoding process;
- **Accuracy Boost in Most Tasks**: DeCoRe achieves more accurate results compared to most baselines in the majority of tasks (**10 out of 17 tasks**).

**Limitations and Future Work**:

- While **DeCoRe may not outperform specific baselines** on every task (e.g., ITI in TruthfulQA or CAD in MemoTrap and NQ-Swap), these **baselines often experience significant performance drops** on other tasks. This variability is a common limitation in the field, as methods may be tailored to excel in particular areas. In contrast, **DeCoRe offers consistently strong performance with very minimal drops** across tasks.
- DeCoRe uses token-level conditional entropy to modulate the contrastive weight. Future work may consider **other hallucination detection methods** beyond token-level conditional entropy such as semantic entropy.

---

### Meta-Review · Area_Chair_WUBy · 2024-12-21

**Metareview:**

The paper introduces a variant of contrastive decoding to mitigate hallucinations in language models. The proposed DeCoRe identifies specific retrieval heads that influence the generation of grounded content and uses a masking strategy to induce hallucinations. It then applies contrastive decoding to differentiate accurate outputs from hallucinatory ones. Experiments demonstrate improved contextual faithfulness on datasets like XSum and enhanced factual accuracy on tasks such as TriviaQA.

Pros:
1. The paper is clearly structured, making complex ideas accessible and easy to follow.
2. Experimental results demonstrate the capability of DeCoRe in mitigating hallucinations across various tasks.

Cons:
1. Limited Novelty. The paper seems a combination of the existing findings, which applies the retrieval heads (Wu et al., 2024) to the Contrastive Decoding (Li et al., 2023) method, potentially limiting the originality of the contribution.
2. Effectiveness. The improvement over baselines is at odds and not pronounced enough. It seems that in the experiments, many baselines can achieve better results than DeCoRe. The model's performance improvements are inconsistent across different datasets.
3. Insufficient evaluation. The paper only provides evaluations on the effectiveness of mitigating hallucination. However, as a decoding method, it is also important to show the generated texts are coherent and fluent. Therefore, it is required to provide evidence to show this decoding method can generate text with high quality with low hallucination.

While the reviewers agree this paper is fairly well written and easy to follow, there are several major drawbacks that are not fully addressed. Therefore, I believe this paper needs another round of major revision before it can be published.

**Additional Comments On Reviewer Discussion:**

While the reviewers agree this paper is fairly well written and easy to follow, and appreciate the efforts of providing additional experimental results, there are still several major drawbacks that are not fully addressed during the rebuttal including the inconsistent improvement across different datasets. Therefore, I believe this paper needs another round of major revision before it can be published.

---

### Decision · Program_Chairs · 2025-01-22

Reject